# A sandcastle worm-inspired strategy to functionalize wet hydrogels

Donghui Zhang [1,5], Jingjing Liu[1,5], Qi Chen [2], Weinan Jiang[2], Yibing Wang[1,3], Jiayang Xie[2], Kaiqian Ma[2], Chao Shi[2], Haodong Zhang [2], Minzhang Chen[2], Jianglin Wan[2], Pengcheng Ma [2], Jingcheng Zou [2], Wenjing Zhang [2], Feng Zhou[4] & Runhui Liu [1,2✉]

Hydrogels have been extensively used in many fields. Current synthesis of functional hydrogels requires incorporation of functional molecules either before or during gelation via the pre-organized reactive site along the polymer chains within hydrogels, which is tedious for polymer synthesis and not flexible for different types of hydrogels. Inspired by sandcastle worm, we develop a simple one-step strategy to functionalize wet hydrogels using molecules bearing an adhesive dibutylamine-DOPA-lysine-DOPA tripeptide. This tripeptide can be easily modified with various functional groups to initiate diverse types of polymerizations and provide functional polymers with a terminal adhesive tripeptide. Such functional molecules enable direct modification of wet hydrogels to acquire biological functions such as anti-microbial, cell adhesion and wound repair. The strategy has a tunable functionalization degree and a stable attachment of functional molecules, which provides a tool for direct and convenient modification of wet hydrogels to provide them with diverse functions and applications.

[1] State Key Laboratory of Bioreactor Engineering, East China University of Science and Technology, Shanghai 200237, China. [2] Key Laboratory for Ultrafine Materials of Ministry of Education, Frontiers Science Center for Materiobiology and Dynamic Chemistry, Research Center for Biomedical Materials of Ministry of Education, School of Materials Science and Engineering, East China University of Science and Technology, Shanghai 200237, China. [3] Biomedical Nanotechnology Center, School of Biotechnology, East China University of Science and Technology, Shanghai 200237, China. [4] State Key Laboratory of Solid Lubrication, Lanzhou Institute of Chemical Physics, Chinese Academy of Sciences, Lanzhou 730000, China. [5] These authors contributed equally: Donghui Zhang, Jingjing Liu. ✉email: rliu@ecust.edu.cn

Hydrogels have been extensively utilized as the scaffold for controlled drug release and the synthetic mimics of extra cellular matrix (ECM) for tissue engineering[1–15]. Hydrogels can be prepared from a large variety of hydrophilic polymers including but not limited to polysaccharides (chitosan[16], sodium alginate (ALG)[17] etc.), poly(ethylene glycol) (PEG)[18], poly(vinyl alcohol) (PVA)[19], poly(2-hydroxyethyl methacrylate) (PHEMA)[20], poly(sulfobetaine methacrylate) (PSBMA)[21], and related polymers[22–25]. Hydrophilic polymers form highly hydrated three-dimensional networks and many of them provide low fouling properties, hence require chemical modification with functional molecules to exert a variety of functions such as cell adhesion[26–30]. These functionalized hydrogels can be prepared via two routes: chemically modifying the polymers with functional molecules before gelation[27,28] (Fig. 1a); modification of functional molecules or the pre-organized reactive sites during gelation[29,31] (Fig. 1a). Nevertheless, current methods to prepare functional hydrogels typically require incorporating reactive groups, such as azide, into the substrates and then tethering functional molecules to them via the reactive groups. However, for a substrate incorporated with a particular type of reactive group (such as azide), only functional molecules bearing specific reactive groups (such as alkyne) can be used to undergo the further functionalization step[32]. Therefore, both the polymers pre-modified with specific reactive/binding sites and the functional molecules bearing the matching reactive/binding sites need to be synthesized to prepare a particular type of functional hydrogel, which limits the flexibility to develop functional hydrogels (Fig. 1a). Therefore, it is an urgent need but also an exceptional challenge to establish a common strategy in preparing variable types of functional hydrogels without having to modify the polymers individually with either functional groups or pre-organized reactive groups. In recent years, adhesive materials have been extensively studied especially the mussel-inspired adhesive materials[33–43]. Many of these studies use molecules that are terminally functionalized with 3,4-dihydroxy-L-phenylalanine (DOPA) or DOPA-lysine combinations to realize surface modifications[44–49]. The function of these molecules can be cell adhesive[50,51], antimicrobial[52,53], and antifouling[54–58]. Nevertheless, it is unknown if these types of adhesive materials can be used to modify hydrogels directly to obtain diverse functions.

We notice that sandcastle worms can secrete adhesive proteins to tightly glue the wet mineral particles (sand, shell, etc.) together and form a tubular house[59,60]. Similarly we are using a biomimetic approach to functionalize hydrogels. Pc-1 is a key adhesion protein in the secretions of sandcastle worms and has 1:1 molar ratio of DOPA and lysine[61,62] (Fig. 1b), which inspired us to design an adhesive short peptide with 1:1 ratio of DOPA: lysine. Because carboxylate group in the C-terminal lysine was redundant to the wet adhesion function of the peptide[63,64], we simplified the C-terminal lysine to dibutylamine (Dba) and designed the adhesive peptide as dibutylamine-DOPA-lysine-DOPA (DbaYKY), a short tripeptide.

Here, we show the DbaYKY tripeptide can tether functional molecules to various types of hydrogels that have no specific groups along the polymer chains (Fig. 1c). The DbaYKY peptide enables direct modification of functional molecules to the surface and interior of hydrogels in an aqueous environment, providing hydrogels with diverse functions such as anti-bacteria, cell adhesion for tissue engineering, and wound healing (Fig. 1c).

## Results

### Synthesis of DbaYKY terminated initiators for variable polymerizations.
An amine terminated DbaYKY peptide can be easily synthesized in gram scale (5.3 g, 43% yield over 4 steps from Fmoc-DOPA(acetonide)-OH) and work as the initiator for polymerization of α-amino acid N-carboxyanhydride (NCA) (Fig. 2a, Supplementary Figs. 1–2). We also demonstrated a convenient synthesis of this amine terminated DbaYKY through a solid-phase method (Supplementary Fig. 3), which enables other research groups and companies to obtain DbaYKY easily, thereby greatly expanding its applications. One step reaction easily converted the amine terminated DbaYKY peptide to three other initiators, respectively, for atom transfer radical polymerization (ATRP), reversible addition-fragmentation chain transfer polymerization (RAFT), and anionic ring-opening polymerization of β-lactams (AROP) (Fig. 2a, Supplementary Figs. 4–5). Four initiators, initiator 1–4, successfully initiated four classes of classical polymerizations (NCA, RAFT, ATRP, and AROP) and provided corresponding polymers (Pol-1 to Pol-8) with diverse functions and a terminal adhesive DbaYKY moiety for attachment (Fig. 2b–e, Supplementary Figs. 6–13, Supplementary Table 1). These polymerization demonstrations showed the convenience and wide applicability of introducing adhesive DbaYKY moiety into polymer chains, using a variety of common polymerization methods.

### Adhesion strength measurements.
We modified DbaYKY moiety to the tip of atomic force microscope (AFM) cantilevers (Supplementary Figs. 14–15) and quantified the adhesion strength of DbaYKY moiety to PEG hydrogels using the single-molecule force spectroscopy (SMFS) approach[65–67] (Fig. 3a). A median rupture force of 223 pN was measured with a wide distribution from 40 to 760 pN, as showed in the representative force–distance (F–D) curves and histograms of rupture force distribution (Fig. 3b). Rupture events that the DbaYKY is not in contact with PEG hydrogel and a few events that DbaYKY has multiple interactions with PEG hydrogel were discarded in the data analysis (Supplementary Fig. 16). To analyze the importance of lysine and Dba group within DbaYKY moiety for adhesion to hydrogels, we prepared peptides YY and YKY through solid phase synthesis (Supplementary Fig. 17) and measured their adhesion to PEG hydrogels (Fig. 3c–d). We found that median values of rupture force of YY without any amine group (~82 pN, Fig. 3c) is significantly lower than that of DbaYKY (223 pN), and that the rupture force of YKY (~164 pN, Fig. 3d) with the removal of Dba is also lower than that of DbaYKY (223 pN), which indicates that the presence and the number of Dba and lysine play an important role in adhesion. The primary amine group within Dba and lysine play a synergetic role with catechol to promote the adhesion strength to hydrogels. The observed importance of amine groups in our study is consistent to the conclusion in the literature that introduction of amine groups, such as lysine, to DOPA-containing peptides can enhance the adhesion strength. We prepared peptide KYK and found that its median rupture force to hydrogels (~172 pN, Fig. 3e) has no significant statistic difference from YKY (~164 pN), indicating both K and Y are important for adhesion. KYK showed lower adhesion than did DbaYKY, indicating the importance of two catechol units within DbaYKY.

All these show that two amine groups (from Dba and lysine) and two catechol groups (from two DOPA units) are important for the peptide DbaYKY to have strong adhesion to hydrogels, as inspired by the 1:1 DOPA:Lysine component in the adhesive protein Pc-1 from sandcastle worm. We also synthesized peptide KKYY and found its adhesion strength (~203 pN, Fig. 3f) is comparable to that of DbaYKY (no significant difference, Supplementary Table 2), which indicates that strict order within DbaYKY is not important for the adhesion to hydrogels, consistent with the observation in adhesive polymers[68]. The order of rupture force in our study is DbaYKY≈KKYY > YKY ≈ KYK > YY. The results imply that when

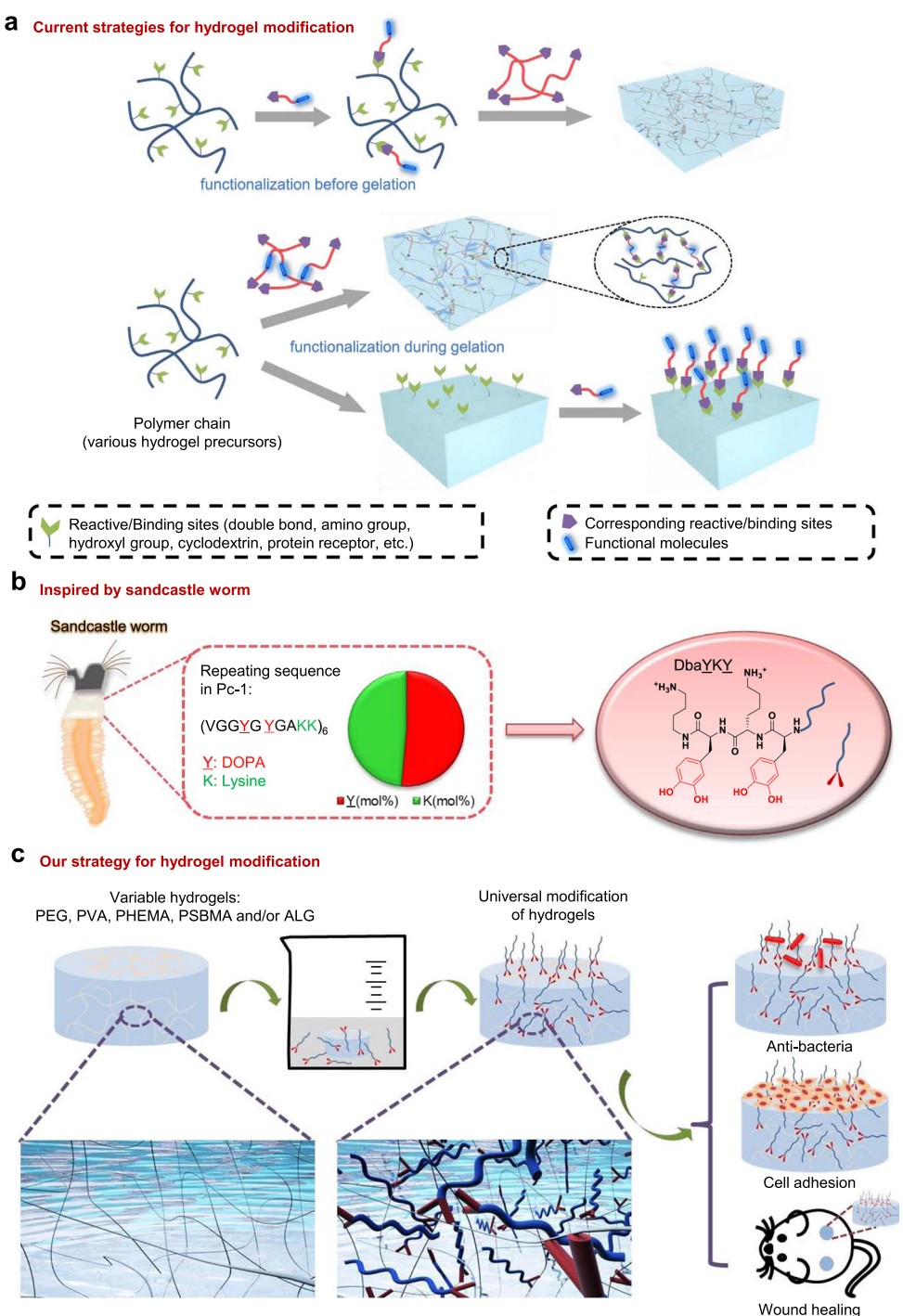

**Fig. 1 Current strategies and sandcastle worm-inspired strategy for hydrogel modification. a** Current strategies for hydrogel modification are hydrogel type and functional group specific. **b** Design of the DbaYKY adhesive tripeptide inspired by sandcastle worm that secretes adhesive Pc-1 protein with repeating sequence of 1:1 molar ratio DOPA:lysine. **c** Direct modification of variable classes of hydrogels using DbaYKY terminated molecules to obtain diverse functions and applications.

the sequence length and the number of K and Y increase, the overall adhesion force will increase, which is also consistent with the observation that modest increase in peptide length, from KY to (KY)$_3$, increases adhesion strength to TiO$_2$[42].

**One-step modification and characterization of hydrogels**. To demonstrate the simple and efficient one-step modification of hydrogel via the adhesive short peptide DbaYKY, we immersed PEG hydrogels into a solution of DbaYKY-terminated polymer

(Pol-1, Pol-3, and Pol-7) (Fig. 4a), and obtained the desired hydrogels with modification of functional polymers after rigorous washing, as confirmed by X-ray photoelectron spectroscopy (XPS) and attenuated total reflectance-fourier transform infrared spectroscopy (ATR-FTIR) analysis (Fig. 4b–e). Compared with the bare PEG hydrogel, polymer modified PEG hydrogels showed significantly increased intensity of the N1s peak; Pol-1 and Pol-3 modified PEG hydrogels showed increased intensity of the C1s peak and a new Br3d peak; Pol-7 modified PEG hydrogel showed

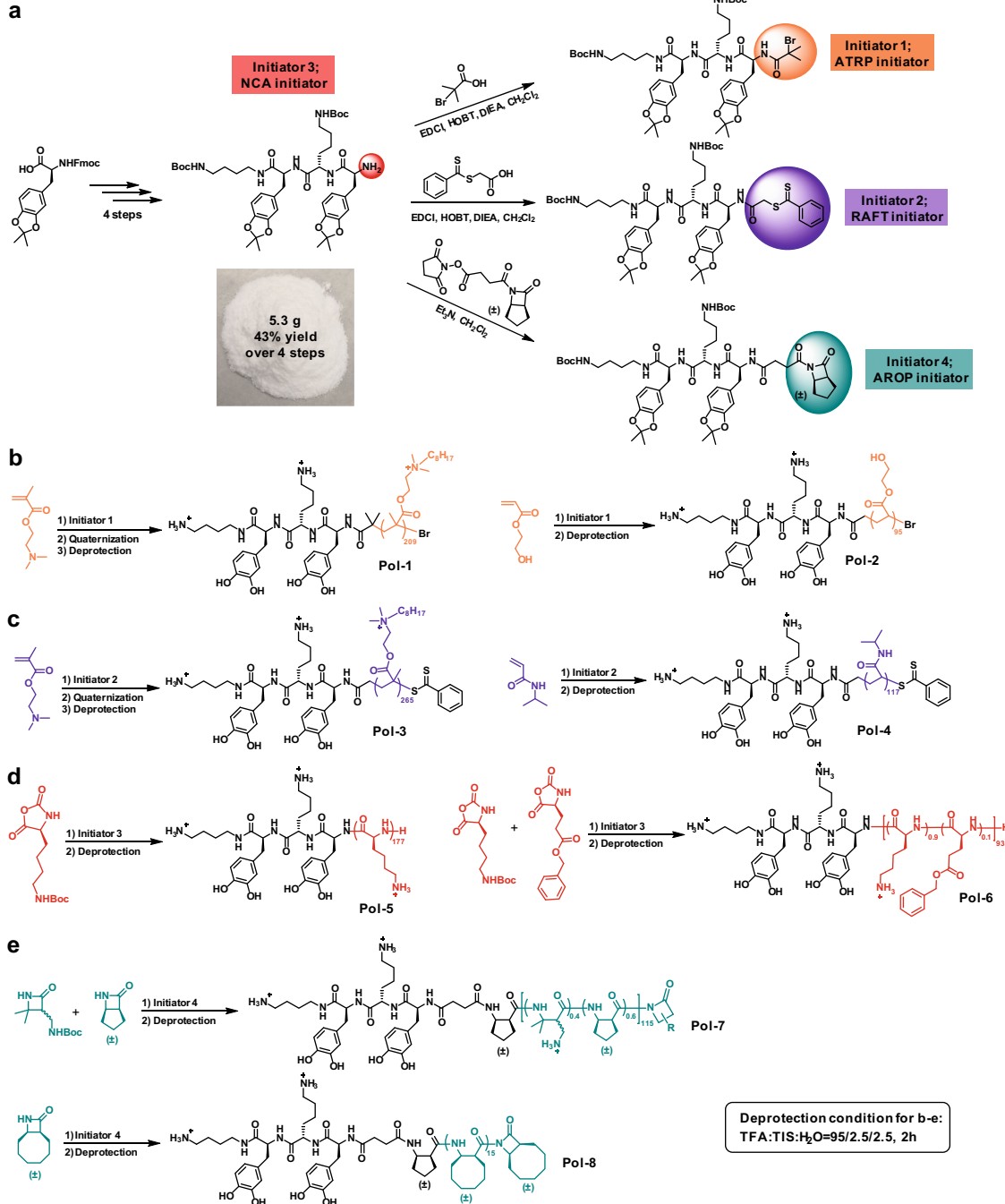

**Fig. 2 General synthesis of DbaYKY terminated initiators and application for variable polymerizations. a** Synthesis of DbaYKY terminated initiators for variable polymerizations including α-amino acid N-carboxyanhydrides (NCA) polymerization, atom transfer radical polymerization (ATRP), reversible addition-fragmentation chain transfer (RAFT) polymerization and anionic ring-opening polymerization of β-lactams (AROP). Demonstration of initiating variable classes of polymerizations: **b** ATRP polymerization, **c** RAFT polymerization, **d** NCA polymerization, **e** AROP polymerization. TFA trifluoroacetic acid, TIS triisopropylsilane.

new peaks of F1s and Cl2p (from ion exchange by Tris buffer during hydrogel modification).

ATR-FTIR characterization also confirmed the successful modification of bare PEG hydrogels with functional polymers, Pol-1, Pol-3, and Pol-7 (Fig. 4b–d). Compared with the bare PEG hydrogel, Pol-1 and Pol-3 modified PEG hydrogels showed a new peak at 3350 cm$^{-1}$ (O–H stretching) and peaks at 2925 cm$^{-1}$ and 2855 cm$^{-1}$ (both C–H stretching); Pol-7 modified PEG hydrogels showed new peaks at 3380 cm$^{-1}$ and 3250 cm$^{-1}$ (N–H stretching), and a new peak at 1650 cm$^{-1}$ (C=O stretching). In the control

group, after bare PEG hydrogels were treated with polymers without the adhesive short peptide DbaYKY (Pol-9, Pol-10, and Pol-11) (Supplementary Fig. 18), polymers were not obviously tethered to the hydrogels as confirmed by the XPS or ATR-FTIR characterization, such as the lack of characteristic Br3d or Cl2p peaks in the XPS spectra. This observation confirmed that these functional polymers were tethered to the bare PEG hydrogels via polymers' terminal adhesive short peptide DbaYKY as we proposed, rather than a chain entanglement of polymers. The successful modification of Pol-1 and Pol-3 to the PVA hydrogels were also

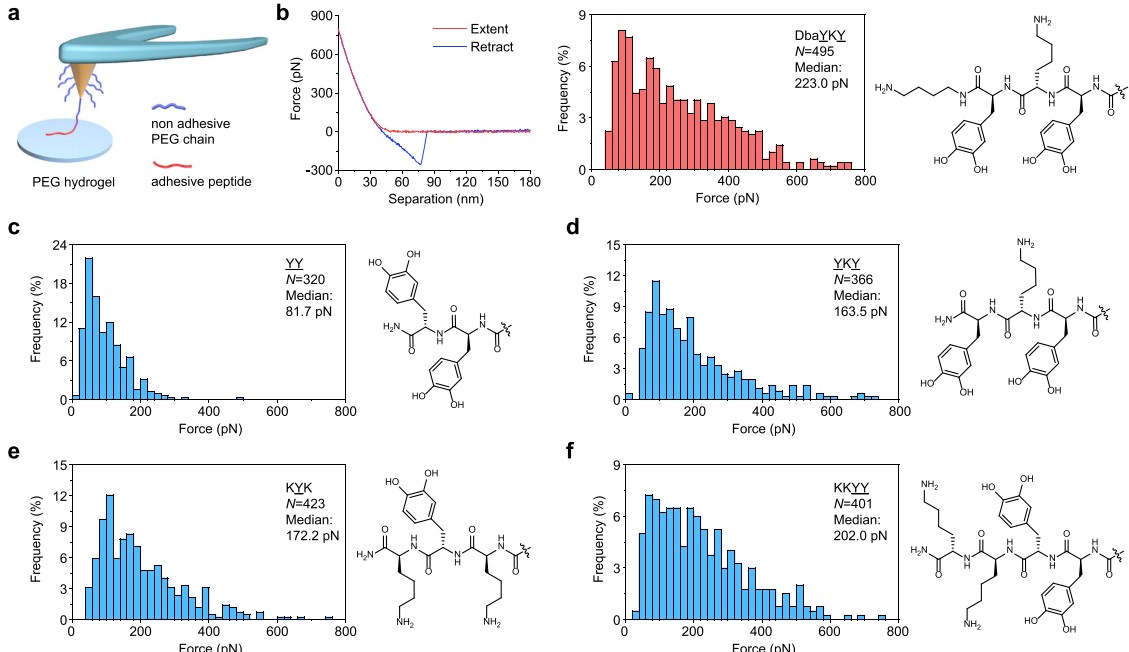

**Fig. 3 SMFS results for the interaction of DbaYKY, YY, YKY, KYK, and KKYY with PEG hydrogels. a** A schematic diagram for the SMFS experiments to measure the interaction strength of the adhesion groups with the PEG hydrogels. **b** Representative F–D curves and rupture force distribution are shown for DbaYKY modified tips. **c–f** Rupture force distribution and chemical structures for YY (**c**), YKY (**d**), KYK (**e**), and KKYY (**f**). *N* values represent the total number of measurements used to plot the histograms.

confirmed by the new peaks at 2925 cm$^{-1}$ and 2855 cm$^{-1}$ (C–H stretching) and the peak at 1725 cm$^{-1}$ (C═O stretching) (Supplementary Fig. 19).

Using Kaiser test to detect amino-containing polymers within PHEMA hydrogels that were modified with Pol-7 (with adhesive DbaYKY) or Pol-11 (without adhesive DbaYKY), we found that only Pol-7 modified PHEMA hydrogel changed to dark purple color; whereas, the Pol-11 "modified" PHEMA hydrogel and the bare PHEMA hydrogel still keep colorless (Fig. 4f). All these studies indicated successful functional polymer modification to hydrogel via the terminal adhesive DbaYKY group.

**Quantitation and analysis of molecule modification to hydrogels.** We tried to quantify the modification of polymers to hydrogels using Pol-7 as the model polymer because Pol-7 has many amine groups for quantification using the fluorescamine method[69] with the aid of the dry weight ratio of hydrogel (Supplementary Fig. 20), if the hydrogel can be dissolved in certain conditions, such as PVA (dissolvable in hot water) and ALG (dissolvable in sodium citrate solution). The freeze-dried hydrogel was dissolved as 1 wt% solution for fluorescence detection, from which the concentration of functional molecule was obtained using a calibration curve (Supplementary Fig. 21 and Supplementary Fig. 22a). Using the dry weight of hydrogel for calculation, the modification amount of Pol-7 to PVA hydrogels and ALG hydrogels was ~0.0515 % and ~0.203% (w/w, weight of Pol-7 bound to hydrogel/dry weight of initial hydrogel), respectively (Supplementary Fig. 22b). Using the wet weight of hydrogel for calculation, the modification amount of Pol-7 to PVA hydrogels and ALG hydrogels was ~0.003% and ~0.017% (w/w, weight of Pol-7 bound to hydrogel/wet weight of initial hydrogel), respectively (Supplementary Fig. 22c). The efficiency of Pol-7 modified to PVA hydrogels and ALG hydrogels were ~0.360% and ~1.97% (w/w, Pol-7 bound to hydrogel/Pol-7 used initially), respectively (Supplementary Fig. 22d). A higher amount of polymer was

modified to the ALG hydrogel than the PVA hydrogel likely because the ALG hydrogel has a large amount of negatively charged carboxyl groups, which will electrostatically attract Pol-7 that has multiple amine groups. However, other hydrogels in this study are not soluble and therefore are not suitable for quantification using the fluorescamine method.

To study the modification amount of adhesion molecules to all hydrogels, we used fluorescent rhodamine-conjugated DbaYKY via an OEG8 linker, namely DbaYKY-OEG8-Rh (Supplementary Fig. 23), which emits fluorescence in the green laser under acidic conditions[70]. After modification, the fluorescent intensity of each hydrogel was collected to quantify the modification amount of polymers to hydrogels using the calibration curve (Supplementary Fig. 24). To analyze efficiency of DbaYKY-OEG8-Rh modification to PEG hydrogels, a gradient concentration of DbaYKY-OEG8-Rh were used to obtain incrementally increased amount of modified molecules to the hydrogel with the increase of DbaYKY-OEG8-Rh concentration, from 0.0625 to 0.25 mg/mL, which indicates that the extent of functionalization can be tailored through the concentration of the modifier solution (Supplementary Fig. 25a–b). Using 0.5 mg/mL of DbaYKY-OEG8-Rh for modification, and using the dry weight of hydrogel for calculation, the modification amount of DbaYKY-OEG8-Rh to PEG hydrogel, PHEMA hydrogel, PSBMA hydrogel, PVA hydrogel, and ALG hydrogel was ~0.051%, ~0.104%, ~0.046%, ~0.020%, and ~0.083% (w/w, weight of DbaYKY-OEG8-Rh bound to hydrogel/dry weight of initial hydrogel), respectively (Supplementary Fig. 25c). Using the wet weight of hydrogel for calculation, the modification amount of DbaYKY-OEG8-Rh to PEG hydrogel, PHEMA hydrogel, PSBMA hydrogel, PVA hydrogel, and ALG hydrogel was ~0.004%, ~0.058%, ~0.022%, ~0.001%, and ~0.007% (w/w, weight of DbaYKY-OEG8-Rh bound to hydrogel/wet weight of initial hydrogel), respectively (Supplementary Fig. 25d). The efficiency of DbaYKY-OEG8-Rh modified to PEG hydrogel, PHEMA hydrogel, PSBMA hydrogel,

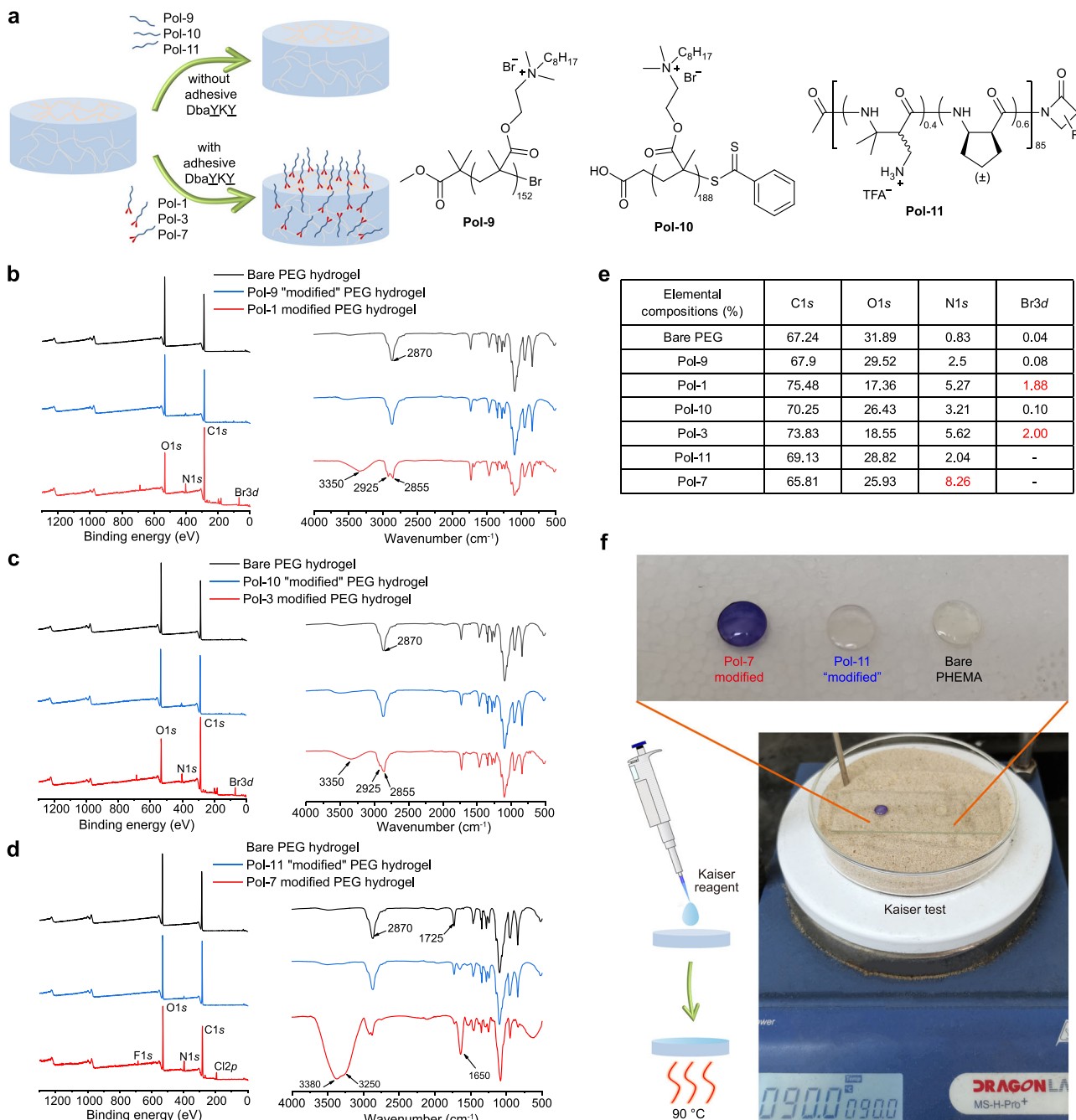

**Fig. 4 Characterization on one-step modification of hydrogels. a** A schematic diagram for direct modification of bare hydrogels with DbaYKY terminated polymers (Pol-1, Pol-3, and Pol-7) or with corresponding polymers (Pol-9, Pol-10, and Pol-11) that have no terminal adhesive peptide. **b** XPS and ATR-FTIR characterization on Pol-1 (with adhesive DbaYKY) and Pol-9 (without adhesive DbaYKY) modified PEG hydrogels, using bare PEG hydrogel for comparison. **c** XPS and ATR-FTIR characterization of Pol-3 (with adhesive DbaYKY) and Pol-10 (without adhesive DbaYKY) modified PEG hydrogels. **d** XPS and ATR-FTIR characterization of Pol-7 (with adhesive DbaYKY) and Pol-11 (without adhesive DbaYKY) modified PEG hydrogels. **e** Elemental compositions of hydrogels determined from XPS characterization. **f** Examination on PHEMA hydrogels modified with Pol-7 (with adhesive DbaYKY) or Pol-11 (without adhesive DbaYKY) using Kaiser test.

PVA hydrogel, and ALG hydrogel was ~3.92%, ~54.6%, ~20.6%, ~1.10%, and ~6.44% (w/w, DbaYKY-OEG8-Rh bound to hydrogel/DbaYKY-OEG8-Rh used initially), respectively (Supplementary Fig. 25e). The observed difference in the efficiency of functionalization on different hydrogels may come from many aspects, including the large difference in water content of different hydrogels, the difference in crosslinking methods (chemical crosslinking for PEG, PHEMA, and PSBMA, and physical crosslinking for PVA and ALG), and the difference in charge (zwitterionic for PSBMA, negatively charged for ALG, and uncharged for others). Using the DbaYKY-OEG8-Rh model, we also analyzed the stability of the modification under shaking in PBS at room temperature for 1 day, 3 days, and 7 days, or incubation with proteinase K at 37 °C for 2 h, 1 day, 3 days, and 7 days. Little difference in fluorescence intensity of the hydrogels was observed after either treatment for up to 7 days (Supplementary Fig. 26), indicating that the hydrogel modification has favorable stability.

To analyze the possible structure changes of DbaYKY adhesion moiety during hydrogel modification, we synthesized an adhesive peptide DbaYKY-Ac as a model molecule (Supplementary Fig. 27), and monitored the change of DbaYKY-Ac in the hydrogel modification condition (in a pH 8.5 Tris buffer) for up to 24 h. HPLC results indicated a gradually weakened peak of DbaYKY-Ac itself, and the appearance of some new peaks (Supplementary Fig. 28). UV–Vis results showed a new peak at around 360 nm, which may come from the change of catechol groups, such as Michael addition between amine and the catechol group (Supplementary Fig. 29). MALDI results further confirmed that both intramolecular and intermolecular Michael addition reaction exist in the DbaYKY-Ac solution for hydrogel modification (Supplementary Fig. 30). It's noteworthy that no Schiff base intermediate was observed from MALDI analysis.

**Application of one-step hydrogel modification**. This simple and efficient one-step modification of hydrogel facilitated the functionalization of bare hydrogels for variable application. Antimicrobial property can be obtained via modifying bare hydrogels with antimicrobial polymers that have proven to be effective antimicrobial agents[71]. We modified bare PEG and PVA hydrogels with Pol-1 and Pol-3 and examined the functionalized hydrogels for their antibacterial properties derived from the incorporated polymers, using *Escherichia coli* (*E. coli*) and *Staphylococcus aureus* (*S. aureus*) as the representative Gram-negative and Gram-positive bacteria, respectively (Fig. 5a). For both PVA and PEG hydrogels, Pol-1 and Pol-3 modification enabled more than 99% killing of *E. coli* and *S. aureus* (Fig. 5b–c). After hydrogels were incubated with the bacteria solution for 1 day, 3 days, and 7 days respectively, we found that all bacteria were killed. We then increase the challenge on the antibacterial activity of hydrogel surfaces and add bacteria again to these hydrogels. After hydrogels were incubated with the bacteria solution for another 2.5 h, all Pol-1 or Pol-3 modified PEG or PVA hydrogels showed >95% killing efficiency, with most cases showing >99% killing (Fig. 5d). This study also confirmed the durability of the functionality, which echoes the results of the stability test.

Encouraged by above demonstration, we continued to examine the wide applicability of the adhesive tripeptide DbaYKY by modifying DbaYKY terminated cell-adhesive polymer, Pol-7[72–74], to five different classes of hydrogels (PEG, PHEMA, PVA, PSBMA, ALG) without having to incorporate extra reactive groups. ATR-FTIR characterization confirmed the successful modification of Pol-7 to PHEMA, PVA hydrogels by the peaks at 1650 cm⁻¹ (C = O stretching) as well as the successful modification of Pol-7 to PEG, PSBMA and ALG hydrogels by the peaks at 3380 cm⁻¹ and 3250 cm⁻¹ (O–H stretching and N–H stretching, respectively) and the sharp peak at 1650 cm⁻¹ (C = O stretching) (Supplementary Fig. 19). Polymer modification to variable hydrogels was also confirmed using the Kaiser reagent to detect the amine groups within polymer chains (Supplementary Fig. 31). Bare hydrogels were used as the negative control, and TCPS petri dishes were used as the positive control (Supplementary Fig. 32). All five classes of hydrogels showed excellent antifouling effect when incubated with fibroblast cells, which is a desired property of hydrogels in preventing nonspecific cell adhesion; in sharp contrast, Pol-7 modified hydrogels strongly promoted cell adhesion and spreading (Fig. 6a).

To study the long-term cell adhesion on polymer-functionalized hydrogels, we tested fibroblast cell adhesion on Pol-7 modified PEG hydrogels for 1 day, 3 days, and 7 days, using bare PEG hydrogels as the control. We performed immunostaining of DAPI (blue), actin filaments (green), and vinculin (red)

after fibroblast cells culture for 1 day, 3 days, and 7 days. The bare PEG hydrogel does not support fibroblast adhesion; in sharp contrast, fibroblast cells adhere well to the Pol-7 modified hydrogel and proliferate to have gradually increased cell density from day 1 to day 7. It's noteworthy that cells adhere well to the Pol-7 modified hydrogel after 7 days and keep healthy morphology (Fig. 6b). The 7 days cell adhesion result confirms the durability of the functionality, which echoes the result of the stability test. To confirm the observed cell adhesion to polymer-modified hydrogel is not from polymer entanglement, we compared cell adhesion performance on PHEMA hydrogels after incubation with Pol-7 and Pol-11, respectively. We observed prominent cell adhesion to Pol-7 (with adhesive DbaYKY) modified PHEMA hydrogel; in sharp contrast, Pol-11 (without adhesive DbaYKY) "modified" PHEMA hydrogel cannot support cell adhesion, which underpinned the adhesive function of the DbaYKY tripeptide (Fig. 6c).

To increase the permeability of polymers into hydrogels, we utilized the freeze-drying treatment for hydrogels before polymer modification because the formation of ice crystals in the freezing process will cause partial collapse of the hydrogel structure, partially destroying the three-dimensional network structure of hydrogels and producing larger pores[75–77]. When a Pol-7 modified hydrogel was cut into two parts from the middle both inside faces showed excellent cell adhesion, which indicated an efficient functionalization of hydrogels for both the surface and the interior by a simple one-step modification via the DbaYKY adhesive tripeptide (Fig. 6d). In order to visualize the permeability of the hydrogel in normal and lyophilized hydrogels, we used rhodamine fluorescent molecule DbaYKY-OEG8-Rh to functionalize PHEMA hydrogels that are processed with or without the freeze-drying step. The confocal images of hydrogel surface and hydrogel cross section showed that DbaYKY-OEG8-Rh can only functionalize the surface of PHEMA hydrogel, and cannot penetrate the PHEMA hydrogel structure, because the cross-section confocal image has no fluorescence. In contrast, both surface and cross section of freeze-dried PHEMA hydrogels showed strong fluorescence, indicating that DbaYKY-OEG8-Rh can penetrate the interior of the freeze-dried PHEMA hydrogel (Supplementary Fig. 33).

Cell adhesion to hydrogels is critical for implants of wound repair[78,79]. We then examined how the DbaYKY bearing Pol-7 easily functionalize bare PVA hydrogels and promote cell adhesion and wound healing using a rat full-thickness wound model. We found that Pol-7 modification promoted wound healing significantly and the Pol-7 modified PVA hydrogels resulted in almost healed wound after treatment for 12 days, which again underpinned the functionalization and application of the easy and efficient one-step modification of hydrogels via the DbaYKY adhesive tripeptide (Fig. 7a–c). In contrast, the bare PVA hydrogel treated group and the blank control group (the control without hydrogel treatment) still had obvious unhealed wound after 12 days. Histological evaluation on wound tissue, using hematoxylin and eosin (H&E) staining, showed that Pol-7 modified PVA hydrogel resulted in a complete recover of epidermis; whereas, the bare PVA hydrogel treated group and the blank control group resulted in incomplete wound healing (Fig. 7a–c).

## Discussion
Hereby we report a simple one-step hydrogel functionalization strategy using a sandcastle worm-inspired cell adhesive DbaYKY tripeptide. The functionalization method is simple and could be widely used if with the availability of specific functionalization molecules, as our demonstration in both solution-phase and

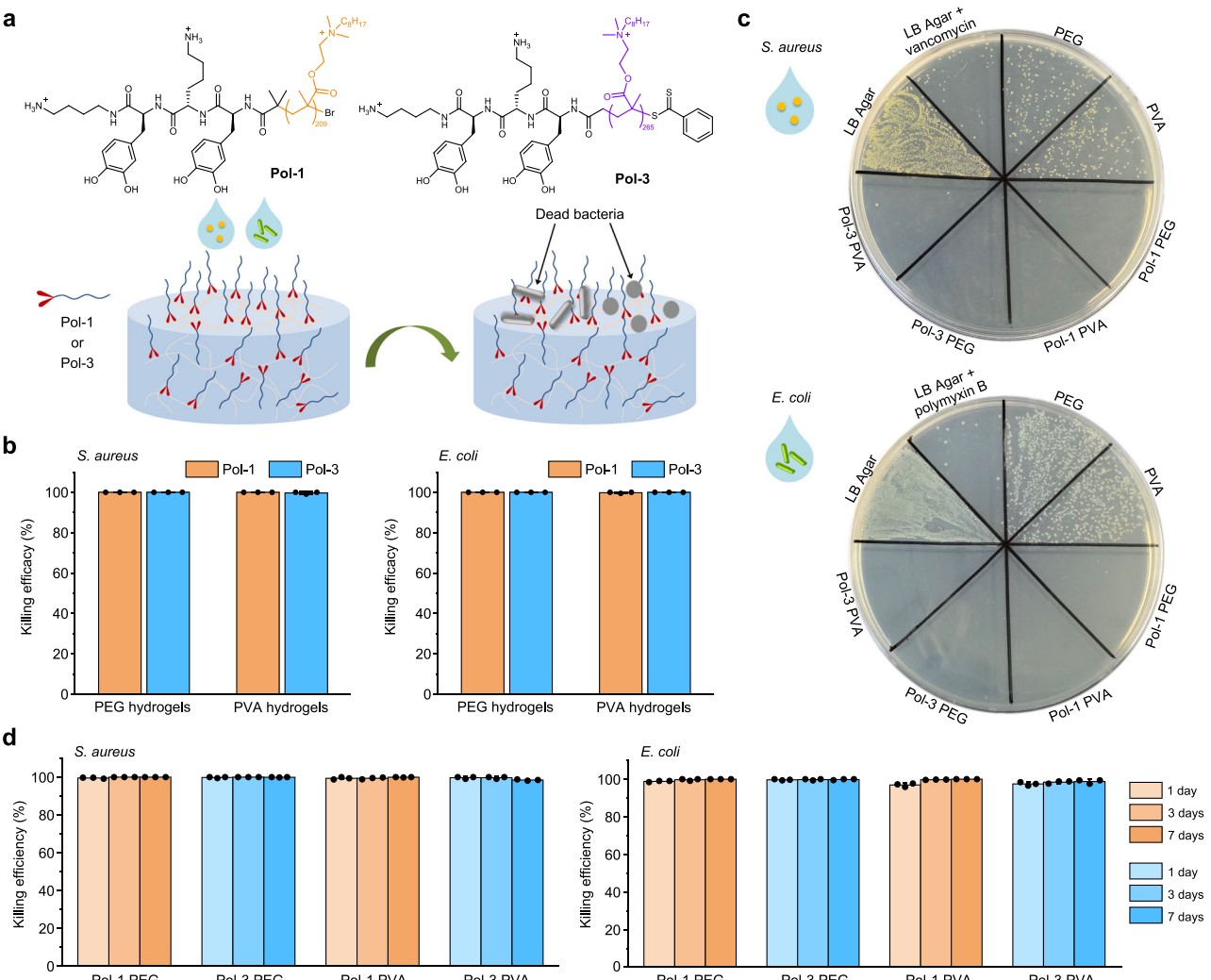

**Fig. 5 One-step modification of hydrogels to have antibacterial property. a** A schematic diagram for antibacterial test of hydrogels from direct modification with DbaYKY terminated antibacterial polymers, Pol-1 and Pol-3. Antibacterial tests were performed by incubating 40 μL bacterial suspension ($2 \times 10^6$ CFU/mL) on top of the hydrogels for 2.5 h. **b** Bacterial killing efficacy of Pol-1 and Pol-3 functionalized hydrogels against *E. coli* and *S. aureus*. n = 3, mean values ± s.d. **c** Images of *E. coli* and *S. aureus* colonies on the LB agar plates correlating to surviving bacteria on the surface of bare hydrogel and functionalized hydrogels. LB agar and antibiotic-loaded LB agar (1 mg/mL vancomycin for *S. aures* and 1 mg/mL polymyxin B for *E. coli*) were used as the negative control and the positive control, respectively. **d** Durability test on the bacterial killing efficacy of Pol-1 and Pol-3 functionalized hydrogels against *E. coli* and *S. aureus*. n = 3, mean values ± s.d. The antibacterial tests were performed using hydrogels that were initially incubated with bacterial solution for 1 day, 3 days, and 7 days.

solid-phase synthesis. The amine-terminated DbaYKY tripeptide (DbaYKY-NH₂) could be easily linked to commercial molecules such as antibacterial agents and cell adhesive RGD peptide. This terminal-functionalized tripeptide can initiate diverse types of polymerizations successfully to prepare functional polymers that have the adhesive tripeptide as the terminal group and directly modify hydrogels. The hydrogel modification strategy has controllable functionalization degree, and the functionalized hydrogels can both be stable in PBS and proteinase K for up to 7 days to acquire biological functions, such as maintaining antibacterial and cell adhesion functions, for at least 7 days, and promoting wound repair. This hydrogel modification strategy provides a convenient tool for the easy and direct modification of wet hydrogels with diverse applications.

Our study indicates that the amine and catechol groups within the adhesive peptide can form cross-linking at alkaline pH to obtain high density of functionalization, consistent to the report in literature[80]. Promoting cell adhesion and obtaining

antibacterial functions of hydrogels often require sufficient density of functional molecules. Therefore, we chose to use the pH 8.5 condition to have high modification density. If a research requires a monolayer modification, pH 6.0 condition can be used for modification to minimize the catechol-derived crosslinking and multilayer modification[52,81]. We hypothesize that the functionalization mechanism of the DbaYKY terminated polymers to hydrogels is attributed to multiple interactions including hydrogen bonding, cation–π stacking, and charge interactions. The catechol groups within DbaYKY can utilize two neighboring hydroxyl groups as the donors/acceptors to form hydrogen bonding to hydrogels. The positively charged amine groups can form cation–π interactions with the aromatic rings within DOPA. Therefore, the presence and the number K and Y are important for adhesion. The order of rupture force in our study is DbaY-KY≈KKYY > YKY ≈ KYK > YY, which implies that when the sequence length and the number of K and Y increase, the overall adhesion force and the functionalization degree will increase. In

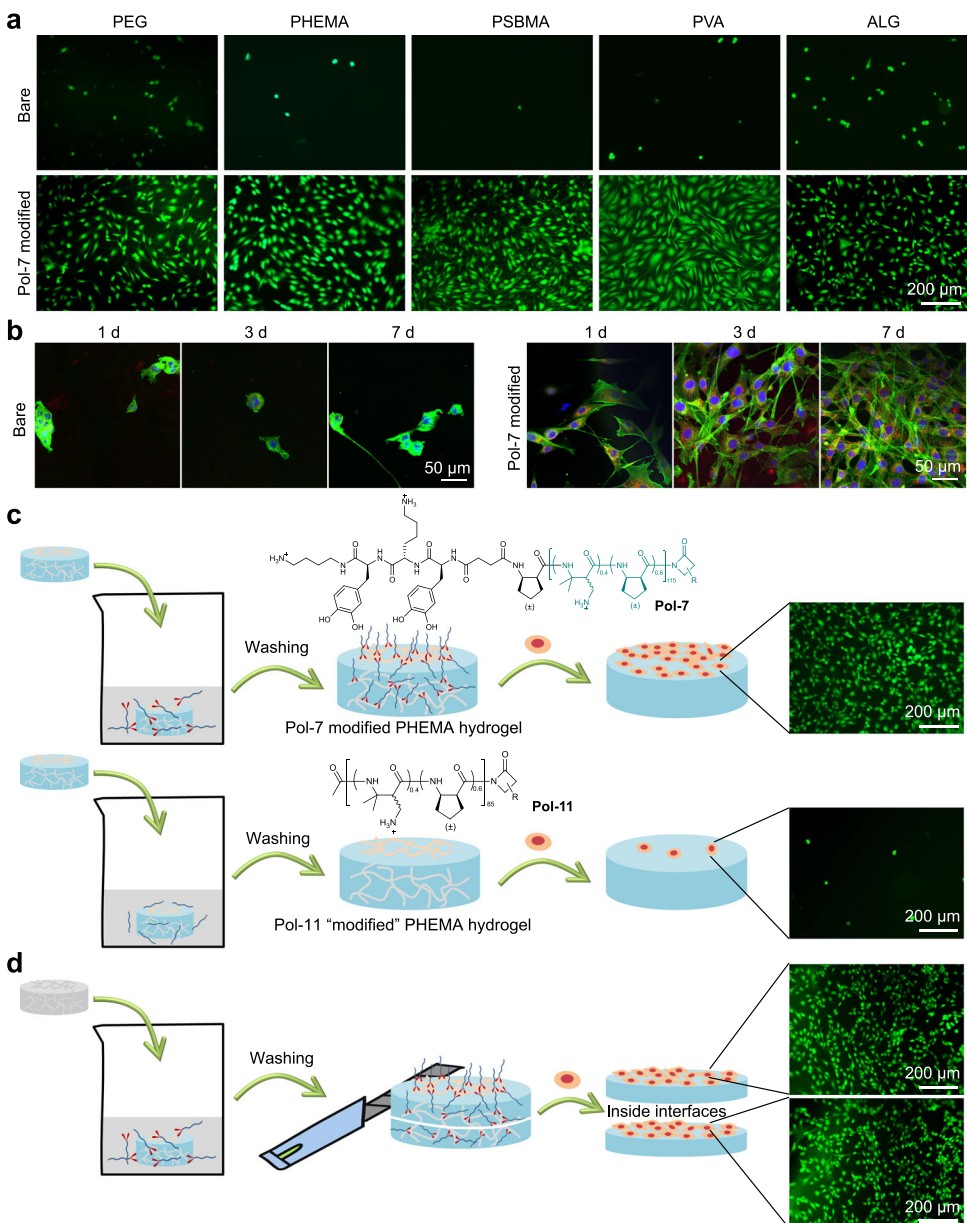

**Fig. 6 One-step modification of hydrogels to have cell adhesive function. a** Fluorescent microscopy images of 3T3 fibroblast cell adhesion to the bare hydrogels and functionalized hydrogels (PHEMA, PEG, PSBMA, PVA, and ALG) with Dba<u>YKY</u> terminated Pol-7 after cell seeding for 1 day. **b** Fluorescence confocal microscope images (green, actin; red, vinculin; blue, nucleus) of NIH 3T3 fibroblast cells on bare PEG hydrogels and Pol-7 modified PEG hydrogels after cell seeding for 1 day, 3 days, and 7 days. Scale bar: 50 μm. **c** Demonstration of cell adhesion on Pol-7 (with adhesive Dba<u>YKY</u>) modified PHEMA hydrogel and Pol-11 (without adhesive Dba<u>YKY</u>) "modified" PHEMA hydrogel, with polymer modification starting from swelling hydrogels. **d** Demonstration on direct modification of Pol-7 to both the outside and inside of the freeze-dried PHEMA hydrogels, as shown for the inside interfaces by cutting the PHEMA hydrogels from the middle. Cells were treated with LIVE/DEAD staining using calcein AM (green) and ethidium homodimer-1 (red), followed by imaging under a fluorescence microscope. Scale bar: 200 μm.

addition, the Dba<u>YKY</u> terminated polymers have positively charged amine groups, therefore, can interact electrostatically with hydrogels bearing negatively charged groups, such as PSBMA and ALG.

Through the mechanism study of polymer adhesion to hydrogels, it can be inferred that the modification amount and modification density will gradually increase over time. A good functionalization method hopes that the modification time should not be too long. We used 24 h as the modification time, which is also commonly used in the field of catecholic chemistry for surface modification, and is generally accepted by many researchers. We found that functionalized hydrogels with a high amount of

modification can be realized at this time window, and the functions of antibacterial and tissue engineering can be realized by using this protocol. During the modification process, we can quantify the efficiency of the functionalization that is the ratio of the functionalizing molecules, which has been linked to the surface, to the overall functionalizing molecules initially used. Moreover, it is possible to change the functionalization degree of hydrogel, for example, the modification amount of Dba<u>YKY</u>-OEG8-Rh to dry weight PEG hydrogel from ~0.0058% to ~0.0262%, with the increase of Dba<u>YKY</u>-OEG8-Rh concentration initially used, from 0.0625 to 1 mg/mL. For different molecules modified to different hydrogels, the functionalization degree and

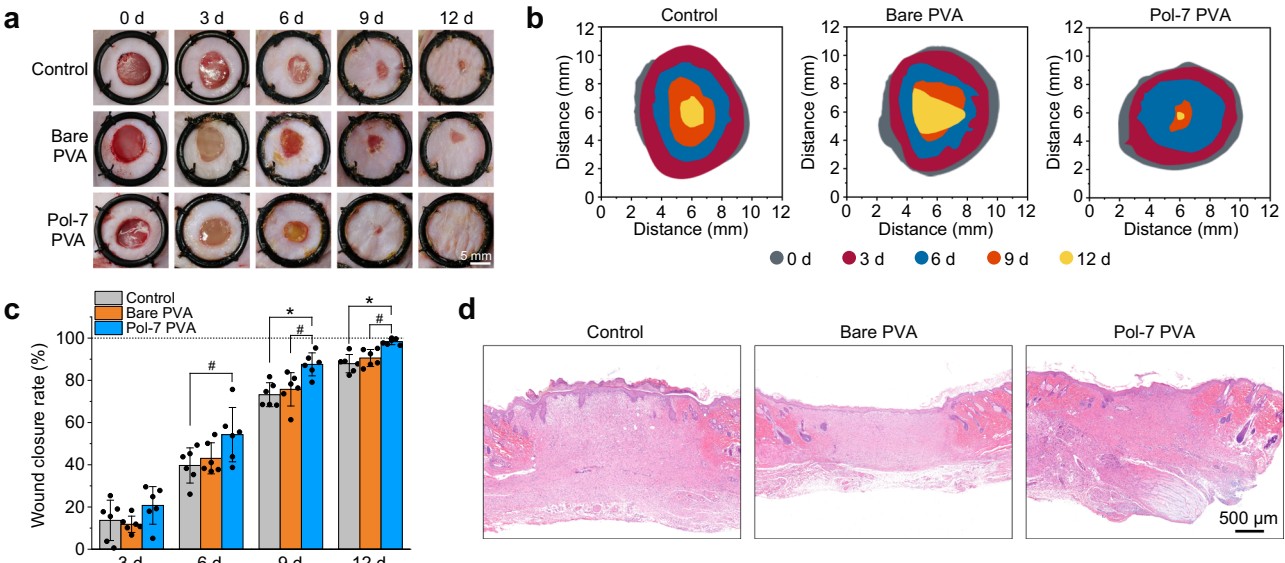

**Fig. 7 One-step modification of hydrogels to promote wound healing. a** Photographs of wounds on day 0, 3, 6, 9, and 12 after treatment with bare PVA hydrogels and Pol-7 modified PVA hydrogels. Scale bar: 5 mm. **b** Schematic diagram of wound healing process from day 0 to day 12 after treatment with bare PVA hydrogels and Pol-7 modified PVA hydrogels. **c** Wound closure rate after treatment with bare PVA hydrogels or Pol-7 modified PVA hydrogels. $n = 6$, mean values ± s.d. Statistical analysis: one-way ANOVA with Tukey's post-test, #$P < 0.05$, *$P < 0.01$. **d** H&E staining of wound tissue on day 12 after treated with bare PVA hydrogels or Pol-7 modified PVA hydrogels. Scale bar: 500 μm. Wounds without hydrogel treatment were used as the blank control in all above studies (**a–d**).

the efficiency of the functionalization are generally different. In our study, we found that the modification amount of DbaYKY-OEG8-Rh to five different hydrogels were from ~0.020% to ~0.104% respect to dry hydrogels, and from ~0.001% to ~0.058% respect to wet hydrogels. For specific applications in the future, the functionalization can be optimized by changing the functionalization conditions such as the concentration of molecules and pH value during functionalization.

This strategy is proposed as a surface functionalization of the exposed hydrogel surface. The functionalization typically cannot penetrate the hydrogel network, even though, bulk functionalization of hydrogel structure is possible by using lyophilized hydrogels. Hydrogels with large pore size for efficient polymer diffusion into hydrogels can be used directly for functionalization with the reported method here; polymers with small pore size can be pretreated, such as freeze-drying, to obtain enlarged pore size that enables efficient polymer diffusion into hydrogels and functionalization. Therefore, bulk functionalization of hydrogel structure is also possible by using lyophilized hydrogels. This adhesive peptide strategy can be used to meet the requirement for bulk hydrogel functionalization with diverse applications such as regulating cell migration into hydrogels in tissue engineering and cell encapsulation.

## Methods

**Synthesis**. Information about all reagents is present in the Supplementary Information. Synthetic procedure of all compounds and polymers are described in Supplementary Information. NMR spectra of compounds and polymers are showed in Supplementary Figs. 34–80. GPC trace of polymers are showed in Supplementary Figs. 81–91.

**Hydrogels preparation**. To demonstrate that our direct modification strategy is compatible with diverse types of hydrogels, we tested five different types of hydrogels, including PEG hydrogel (chemical crosslinking, without functional groups), PHEMA hydrogel (chemical crosslinking, with hydroxyl functional groups), PSBMA hydrogel (chemical crosslinking in the back bone and ion pair interactions between zwitterions), PVA hydrogel (physical crosslinking based on the amphiphilic structure), and ALG hydrogel (physical crosslinking based on ion chelation). These hydrogels can represent the commonly used crosslinking types of

hydrogels for the purpose of our study. All hydrogels were prepared and shaped in 4.5, 6 or 8 mm diameter, with 1 mm thickness.

Poly(ethylene glycol) diacrylate ($M_n = 2000$, PEGDA) was dissolved in Milli-Q water at a concentration of 20 wt% followed by addition of a photoinitiator 1-[4-(2-hydroxyethoxy)-phenyl]-2-hydroxy-2-methyl-1-propane-1-one (I2959, 0.1 wt%, pre-dissolved in EtOH at a 10 wt% stock solution). The mixed solution was placed in the mold and exposed to ultraviolet light (365 nm, 500 mW/cm$^2$) for 5 min to form a hydrogel disk.

PVA hydrogels were obtained from repeated freezing and thawing. 10 wt% PVA solution (dissolved at 90 °C) was frozen at −18 °C for 16 h, and then thawed at room temperature for 8 h. The hydrogels were formed after 3 cycles of freezing/thawing.

2-hydroxyethyl methacrylate (HEMA) monomer (50 wt%), cross-linker N, N'-methylenebisacrylamide (MBAA, 0.8 wt%) and photoinitiator I2959 (0.1 wt%) were dissolved in the mixed solvent (ethylene glycol/EtOH/H$_2$O = 1.5:1:1.5). The above mixed solution was placed in the mold and exposed to ultraviolet light (365 nm, 500 mW/cm$^2$) for 12 min to form a hydrogel disk.

The zwitterionic sulfobetaine methacrylate (SBMA) monomer (45 wt%), the cross-linker MBAA (1 wt%), and photoinitiator (I2959, 0.1 wt%) were dissolved in Milli-Q water. The above solution was placed in the mold exposed to ultraviolet light (365 nm, 500 mW/cm$^2$) for 3 min to form a hydrogel disk.

The 5 wt% ALG, (viscosity: 200–500 mPa.s) solution was prepared in Milli-Q water, then mixed with excess CaCl$_2$ solution (10 wt%) overnight to obtained ALG hydrogels.

All hydrogels were immersed in Milli-Q water and the water was replaced for five times every 12 h to remove free chemicals within hydrogels.

**Hydrogel modification**. Polymers with an adhesive dibutylamine-DOPA-lysine-DOPA tripeptide (DbaYKY), described in Fig. 2 and Supplementary Figs. 6–13, were prepared at a concentration of 4.0 mg/mL (0.093–0.27 mM). Water soluble polymers were dissolved in 100 mM Tris (pH = 8.5); polymers insoluble in water were dissolved in the mixed solution (Tris/MeOH = 4:6). For hydrogel modified by a polymer, the hydrogel (6 mm in diameter) was immersed into the polymer solution (60 μL) in a 2 mL tube at room temperature for 24 h. The modified hydrogel was washed with Milli-Q water for three times and then was immersed in water that was changed with fresh water for five times during 24 h.

In order to examine the efficacy of polymer modification inside the hydrogel, PHEMA hydrogel was freeze-dried using a Labconco® FreeZone Plus 4.5 liter cascade benchtop freeze dry system and then immersed into the Pol-7 solution for 24 h. The modified hydrogel was washed with Milli-Q water for three times and then was immersed in water that was changed with fresh water for five times during 24 h. The unmodified polymers were completely removed, as the last washing water showed negative result in the Kaiser test. The Kaiser test was conducted by followed the general method widely used in solid phase synthesis. The Kaiser reagent is a mixed solution of 2 μL EtOH containing 50 mg/mL

ninhydrin, 2 μL phenol, and 2 μL pyridine. The solution of amino group-containing compounds will turn to dark blue after heating. 2 μL last washing water of the hydrogel in duplicates was mixed with 6 μL of the Kaiser reagent and then was incubated at 90–100 °C for 1 min to show the color change.

**AFM cantilever modification**. The preparation process of AFM cantilever modification was shown in Supplementary Fig. 15. MLCT silicon nitride cantilevers were treated with UV-ozone for 25 min for surface activation. Then the cantilevers were immersed in a 0.2% (v/v) (3-aminopropyl)triethoxysilane (APTES)/toluene solution for 2 h, followed by rinsing with toluene, EtOH sequentially, and drying with a soft stream of $N_2$ to remove the unreacted APTES. The treated cantilevers were annealed at 100 °C for 1 h to give the amine-functionalized cantilevers. These cantilevers were immersed in a solution of 1:10 mixture of NHS-PEG-SH (5000 Da) and NHS-PEG-OMe (2000 Da) at a total concentration of 1 mg/mL in DMSO, containing 0.2% $Et_3N$, for 2 h. This ratio was used to control the binding density of functional PEG and to reduce multiple interactions in the force spectroscopy measurements. The cantilevers were then rinsed with DMSO, Milli-Q water sequentially, and incubated in a 1 mg/mL solution of maleimide-modified adhesive groups (DbaYKY-OEG8-Mal, YY-OEG8-Mal, YKY-OEG8-Mal, KKYY-OEG8-Mal, or KYK-OEG8-Mal) in degassed PBS for 1 h. The modified cantilevers were then rinsed with Milli-Q water and then dried under a soft stream of $N_2$.

**AFM-based force spectroscopy measurements**. AFM force spectroscopy measurements were carried out using Nanoscope VIIIa microscope system (Bruker, USA) at the contact mode. Silicon nitride MLCT A cantilevers of typical spring constant of 25–140 pN/nm were used for all experiments. PEG hydrogels were immersed in a 100 mM Tris buffer (pH 8.5, degassed with $N_2$) for 24 h before measurement. In a typical force measurement, one drop of above Tris buffer was dripped to the surface of a hydrogel and then the cantilever was controlled to approach to the substrate at a constant speed of 1000 nm/s and held at the surface for 2 s to allow for the interaction between adhesive group and the hydrogel. The cantilever was then retracted at the same speed. The F–D curves were recorded using the NanoScope Analysis software. This experiment was repeated twice at different times, and the two results were showed in Supplementary Table 3.

**Modified hydrogels characterization**. X-ray photoelectron spectroscopy (XPS) was conducted on a Thermo Scientific™ K-Alpha™+ spectrometer (with Avantage software) equipped with a monochromatic Al Kα X-ray source (1486.6 eV) operating at 100 W. After modification, hydrogels were freeze-dried. The external surface of each freeze-dried hydrogel was analyzed and all peaks were calibrated with C1s peak binding energy at 284.8 eV for adventitious carbon. The experimental peaks were fitted with Avantage software. Fourier transform infrared (FTIR) spectra were recorded on a Thermo Electron Nicolet 6700 FTIR spectrophotometer by using attenusated total reflectance (ATR). After modification, hydrogels were freeze-dried. The external surface of each freeze-dried hydrogel was analyzed. The hydrogels in duplicates were either incubated with 50 μL Kaiser reagent (Fig. 4f) or immersed into 200 μL Kaiser reagent (Supplementary Fig. 31), and then was incubated at 90–100 °C for 1 min to show the color change. Images of surface and z-axis cross section of DbaYKY-OEG8-Rh functionalized PHEMA hydrogels, with or without the freeze-drying treatment, were collected on a confocal microscope (Leica TCS SP8) with Leica Application Suite X software.

**Quantitative analysis**
*Fluorescamine method.* PVA or ALG hydrogels with 6 mm in diameter were modified by Pol-7 using aforementioned protocol. After modification and washing steps, hydrogels were freeze-dried, pulverized, and weighted. Dried PVA hydrogel fragment was gradually dissolved into Milli-Q water at a mass fraction of 2 wt% at 95 °C for 1 h, and diluted with an equal volume of PB buffer to get a 1 wt% PVA solution in 0.2 M Phosphate buffer (PB, pH = 8.0). Dried ALG hydrogel fragment was gradually dissolved into a 5% sodium citrate solution at a mass fraction of 1 wt% at room temperature under shaking conditions. An aliquot of 90 μL above solutions was added to each well in a 96-well black plate. Pol-7 at concentrations of 0–32 μg/mL in 0.2 M PB (pH = 8.0) and 5% sodium citrate were used to generate the calibration curves for PVA and ALG, respectively. An aliquot of 30 μL acetone (containing 60 μg/mL fluorescamine) was added to each well, and the fluorescent intensity in each well was measured on a SpectraMax M2 plate reader using $\lambda_{ex}$ and $\lambda_{em}$ at 388 nm and 425 nm, respectively. The fluorescent intensity of bare hydrogels without modification were measured and used as background. Dry weight ratio of hydrogel (Supplementary Fig. 20) was measured and calculated by the formula: weight of hydrogel after freeze-drying/weight of swollen hydrogel. The concentration of functional molecule ($C_f$) was obtained using a calibration curve (Supplementary Fig. 21). Hence, the actual weight of molecules bound to hydrogel = ($C_f$ × dry weight ratio of hydrogel/1%) × volume of hydrogel. The efficiency of the functionalization (%) = (($C_f$ × dry weight ratio of hydrogel/1%) × volume of hydrogel/weight of functionalizing molecules initially used) × 100. Quantitative analysis of Pol-7 on PVA and ALG hydrogels were conducted, including modification amount to dry weight hydrogel (%) that is calculated by the formula: (weight of molecules bound to hydrogel/dry weight of initial hydrogel) × 100, and

modification amount to wet weight hydrogel (%) that is calculated by the formula: (weight of molecules bound to hydrogel/wet weight of initial hydrogel) × 100.

*Rhodamine method.* All hydrogels (1 mm in thickness) were punched to 4.5 mm diameter. DbaYKY-OEG8-Rh with concentrations of 0.0625–1 mg/mL (0.032–0.52 mM) was used to functionalize hydrogels using aforementioned protocol in a dark environment. After modification and washing steps, the free water on the surface of the hydrogel was absorbed by the dust-free paper, and then each hydrogel was transferred to each well of a 96-well plate. To test the fluorescence intensity of hydrogels, 100 μL of 0.1 N HCl was added to each well, followed by incubation at room temperature for 1 h to allow HCl to penetrate into the hydrogel throughly. Under acidic conditions, the non-fluorescent rhodamine (Rh) spirolactams will transfer to the highly fluorescent xanthylium isomer of Rh[70]. The fluorescence intensity of each hydrogel was measured by a Typhoon TRIO variable mode imager with Typhoon Scanner Control software using an excitation wavelength ($\lambda_{ex}$) at 532 nm, an emission wavelength ($\lambda_{em}$) at 670 nm for PHEMA and PSBMA hydrogels and at 610 nm for other hydrogels. Concentrations of 2.5–100 μg/mL and 50–400 μg/mL of DbaYKY-OEG8-Rh in 0.1 N HCl were used to generate the calibration curves at $\lambda_{em}$ of 610 nm and 670 nm, respectively. The data were analyzed using the ImageQuant Tools software. All types of bare hydrogels without modification were used as controls, which showed negligible fluorescence values. The concentration of functional molecule ($C_r$) is the actual density of modified molecules to the hydrogel, and was obtained using a calibration curve (Supplementary Fig. 24). Therefore, the weight of molecules bound to hydrogel = $C_r$ × volume of hydrogel. The efficiency of the functionalization (%) = ($C_r$ × volume of hydrogel/weight of functionalizing molecules initially used) × 100. Quantitative analysis of DbaYKY-OEG8-Rh on various hydrogels were conducted, including modification amount to dry weight hydrogel (%) that is calculated by the formula: (weight of molecules bound to hydrogel/dry weight of initial hydrogel) × 100, and modification amount to wet weight hydrogel (%) that is calculated by the formula: (weight of molecules bound to hydrogel/wet weight of initial hydrogel) × 100.

**Analysis of possible adhesion molecule transformation**. DbaYKY-Ac (4.0 mg) was dissolved in 1 mL Tris buffer (100 mM, pH = 8.5), and then 20 μL of this solution was taken out and diluted to 0.1 mg/mL at 0, 1, 4, and 24 h, respectively. The diluent was analyzed by Ultraviolet–visible (UV–vis) spectra and reverse-phase HPLC (mobile phase using a mixture of MeCN and $H_2O$ in the presence of 0.1% TFA, with MeCN increasing incrementally from 5 to 70% over 20 min). At 24 h, the mixture was purified by Sep-pak C18 column, using water as the mobile phase to remove the Tris salt first and then using a mixed solvent of MeOH/water (75:25, v/v) to get DbaYKY-Ac derivatives. The later part was concentrated under reduced pressure to remove MeOH, and freeze-dried to give DbaYKY-Ac as a solid, which was analyzed by matrix-assisted laser desorption/ionization time-of-flight (MALDI-TOF) mass spectra.

**Bactericidal efficacy of polymer modified hydrogel**. *E. coli* 25922 and *S. aureus* 6538 were chosen in this assay. Bacteria cells were seeded in Luria-Bertani (LB) medium and cultured for 10 h at 37 °C with shaking at a speed of 150 rpm. 10 mL the bacterial suspension was centrifuged at 1698 *g* for 5 minutes to collect bacteria cells that were washed three times with PBS. The bacteria cell suspension was adjusted to a cell density of 2 × 10⁶ CFU/mL in PBS as the working suspension for this assay. Pol-1 and Pol-3 modified PEG hydrogel and PVA hydrogel with 6 mm in diameter were put in a 24-well plate and 40 μL bacterial working suspension was added onto the surfaces. LB agar and LB agar with antibotics (1 mg/mL vancomycin for *S.aures* and 1 mg/mL polymyxin B for *E.coli*) were used as the positive control and the negative control, respectively. Hydrogels without modification were used as the blank control. After the 24-well plate was placed at 37 °C for 2.5 h, 1920 μL of PBS was added into each well followed by sonication for 3 min and vortexed for 2 min to disperse bacteria in PBS. 30 μL of this bacteria suspension from each well was spread onto agar plates followed by overnight incubation at 37 °C for colony counting. Bacteria suspensions incubated with Pol-1 or Pol-3 were used to give the colony number $C_{polymer}$. Bacteria suspension incubated with unmodified hydrogels was used to give the colony number $C_{control}$. The killing efficacy of the polymer modified hydrogel surface was calculated using the following equation:

$$\text{killing efficacy}(\%) = \frac{C_{control} - C_{polymer}}{C_{control}} \times 100 \qquad (1)$$

For durability test on the bacterial killing efficacy, hydrogels pre-treated with bacteria cell suspension (40 μL, 2 × 10⁶ CFU/mL) were kept for 1 day, 3 days, and 7 days. Then the hydrogels were treated again with bacteria cell suspension, and the antibacterial efficacy was tested in the same way.

**Cell adhesion assay**. NIH 3T3 fibroblast cells were cultured in treated TCPS petri dish using Dulbecco's modified eagle medium (DMEM, containing 10% fetal bovine serum (FBS), 100 U/mL penicillin, 100 μg/mL streptomycin, and 2 mM L-glutamine) at 37 °C in a 5% $CO_2$ environment. Cells at 80–90% confluency were trypsinized, washed with PBS, and dispersed in DMEM to a final concentration of

$3 \times 10^5$ cell/mL. Before cell adhesion study, all hydrogels were sterilized by UV light irradiation for 30 min. An aliquot of 50 µL the cell suspension was added to the surfaces of Pol-7 modified hydrogels with 6 mm in diameter in a 24-well plate. After incubation at 37 °C for 2 h to allow cell attachment, 1 mL fresh medium was added to each well to immerse the hydrogel, followed by incubating the hydrogel at 37 °C for 24 h. All samples were removed from the medium and subjected to LIVE/DEAD staining to facilitate fluorescent imaging, using a 1 mL solution containing 2 µM calcein AM and 4 µM ethidium homodimer-1. The LIVE/DEAD stained cells were imaged using a fluorescence microscope. The Pol-7 modified cell adhesive PHEMA hydrogel was obtained via lyophilization treatment, which was then cut with a knife from the middle to expose the inside cross section. The same method was used for the cell adhesion assay on the surface and inside interface of Pol-7 modified PHEMA hydrogel that was prepared via lyophilization treatment.

PEG hydrogel modified with Pol-7 was used for long-term cell adhesion test, using bare PEG hydrogel as control. After 1 day, 3 days, and 7 days culture with NIH 3T3 fibroblast cells, medium was removed. The hydrogel was then washed with PBS and then cells on the hydrogel were fixed with 4% paraformaldehyde for 20 min. The hydrogel was washed with PBS thrice and incubated for 5 min with 0.4% triton-X in PBS. Subsequently, the hydrogel was washed with PBS thrice, followed by blocking with 3% bovine serum albumin (BSA) in PBS for 1 h at room temperature and then incubated with Alexa Fluor-555-conjugated anti-vinculin antibody (1:300 in PBS) overnight at 4 °C. The hydrogel was washed with PBS thrice and incubated with FITC-phalloidin (1:200 in PBS) for 2 h and 4′-6-diamidino-2-phenylindole (DAPI) (5 µg/mL) for 10 min. The slide was washed with PBS thrice and its fluorescent images were collected on a confocal microscope (Leica TCS SP8).

**In vivo wound healing in a full-thickness skin defect model**. The rat full-thickness wound model was conducted to evaluate the effect of Pol-7 modified PVA hydrogel on wound healing in vivo. All experiments were performed with the guidelines and approval of the Animal Care and Ethics Committee of East China University of Science and Technology. Three Pathogen-free Sprague-Dawley (SD) rats (female, 8 weeks old, ~200 g) were anesthetized by 1% pentobarbital sodium (50 mg/kg) via intraperitoneal injection. After removing the hair on the back of the rats thoroughly, six 8 mm diameter full thickness skin circular wounds were created in each rat using biopsy punch, with three wounds on each side of the back. 15 mm fixing rings were used to sew around each wound to avoid deformation of the wound caused by skin contraction. Total 18 wounds of three rats were divided into three groups ($n = 6$ for each group), including the control group (without hydrogel treatment), bare PVA hydrogel group, and Pol-7 modified PVA hydrogel group (Pol-7 PVA). Bare PVA hydrogels and Pol-7 modified PVA hydrogels with a diameter of 8 mm were placed directly to each wound, then all wounds were covered with 3 M Tegaderm Film and medical bandages. On day 0, 3, 6, 9, and 12, each wound was photographed and the wound area was calculated by Photoshop. The degree of wound healing was assessed using the wound closure rate from the following equation:

$$\text{Wound closure rate}(\%) = \frac{A_0 - A_t}{A_0} \times 100 \qquad (2)$$

where $A_0$ and $A_t$ represent the wound area at day 0 and certain time, respectively. At day 12, the rats were sacrificed and the wound skin tissues were removed and fixed in 4% paraformaldehyde at room temperature for overnight for paraffin section preparation and hematoxylin and eosin (H&E) staining. All staining images were scanned in Pannoramic 250/MIDI scanner equipped with the CaseViewer 2.0 software.

**Statistics and reproducibility**. Statistical analysis was performed with Origin software. Significance between two groups was determined by two-tailed $t$-test. One-way analysis of variance (ANOVA) with Tukey post-test for more than two variables was carried out. All results were expressed as mean values ± s.d. All micrograph assays were carried out at least three independent times with similar results.

**Reporting summary**. Further information on research design is available in the Nature Research Reporting Summary linked to this article.

## Data availability
Data that support the findings detailed in this study are available in the Supplementary Information and this article. Data is available from the corresponding author. The Source data underlying Figs. 3b–f, 4b, d, and 7c, Supplementary Figs. 20, 22b–d, 25b–e, and 26 are provided as a Source data file. Source data are provided with this paper.

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

## Acknowledgements

This research was supported by the National Natural Science Foundation of China for Innovative Research Groups (51621002), the National Natural Science Foundation of China (21774031, 31800801), Program of Shanghai Academic/Technology Research Leader (20XD1421400), Frontier Science Research Base of Optogenetic Techniques for Cell Metabolism grant 2021 Sci & Tech 03-28 (Shanghai Municipal Education Commission), Research program of State Key Laboratory of Bioreactor Engineering, and Fundamental Research Funds for the Central Universities (JKD01211520). The authors also thank Research Center of Analysis and Test of East China University of Science and Technology for the help on the characterization.

## Author contributions

R.L. directed the whole project. D.Z. and R.L. conceived the idea, proposed the strategy, designed the experiments, evaluated the data, and wrote the manuscript together. D.Z. and J.L. performed majority of the experiments. Q.C. and H.Z. participated the cell experiment. W.J. and J.X. participated the antimicrobial assay and animal experiment. Y.W. performed the AFM measurements. K.M participated solid-phase synthesis and hydrogel modification. C.S., M.C., J.W. and P.M. participated chemical synthesis. J.Z. participated figure preparation. W.Z. participated data analysis. F.Z. participated result discussion and trouble shooting. All authors proof read the manuscript.

## Competing interests

R.L., D.Z. and J.L are co-inventors on a patent application covering reported methods and application. All remaining authors declare no competing interests.

**Additional information**

