## [Peer Review File · Nature Communications]

Reviewers' comments:

Reviewer #1 (Remarks to the Author):

This manuscript from Zhang et al. generates polymers based on a dibutylamine-DOPA-lysine-DOPA peptide and applies them to reduce bacterial adhesion, increase mammalian cell-line adhesion and improve skin wound healing. The strengths of the work are the range of different polymers synthesized through different kinds of polymerization strategies. I considered that the authors gave sufficient information to reproduce the work. However, it should be noted that the authors do not demonstrate the benefit of the sandcastle-worm based sequence. The negative control polymers (Fig. 3a) lack any DOPA functionality. The use of DOPA or dopamine in proteins or synthetic polymers to improve adhesion has been widely demonstrated in the field. For example, it was reviewed back in 2017:

<https://pubs.rsc.org/en/content/articlehtml/2017/ra/c7ra06743g>

There have also been extensive investigations of the biological systems tested here, so there is no evidence to conclude that their polymers have enhanced performance over what has been explored elsewhere. Therefore, I suggest that the work does not show sufficient novelty to be appropriate for Nature Communications.

Minor points:

The use of "Universal" in the title is not adequately justified. The authors would need to explain how the sample of surfaces that are tested represent everything that could be encountered. Also, it is not to be encouraged for authors to claim how remarkable their own work is (in the abstract).

Explain DbayKY in the abstract.

"This tripeptide can initiate diverse types of polymerizations"

It is not the peptide that initiates polymerization but functional groups that can be attached to the peptide.

Main text does not refer to Supp Figures where compounds are characterized.

"design the adhesive peptide as dibutylamine-DOPA-lysine-DOPA (DbayKY), a short tripeptide"
Y is the standard symbol for tyrosine, not DOPA.

Fig. 3a has DbayKYK, when perhaps mean DbayKY

Fig. 5a legend: describe how seeing the cells

Reviewer #2 (Remarks to the Author):

The authors outline a universal strategy for obtaining functional hydrogels using a modular approach towards the functionalization of hydrogels. Authors utilize a tripeptide sequence consisting of DOPA and Lysine units as a terminal group for various types of polymers that possess various properties such as antibacterial effect through the presence of quaternary ammonium groups as side chains. Overall, the process of tailoring properties of the hydrogels using the presented approach appears to be simple and straightforward. While the approach seems to provide hydrogels with desired properties, the underlying science has not been investigated in a rigorous manner. Such shortcomings as explained below makes this report very superficial. The study should have been conducted in a manner that provides a more in-depth analysis of the process, as well as furnishes materials with fine-tuned tailorable properties, rather than broad overall attributes. As presented, the study appears superficial and lacks the rigor for publication in this journal.

Major Comments:

1. The argument that the present approach has superior attributes to the conventional approach where reactive groups are incorporated into hydrogels during their fabrication and are functionalized post-gelation is not correct. The conventional approach gives a level of control over the functionalization process. The extent of functionalization can be tailored. In the present approach, while authors demonstrate various modifications, no level of control over the extent of modification is demonstrated. Hence, as a result, there is also no level of control over the functional aspect. Such a level of control should be demonstrated.
2. No clear data has been provided to support the reason behind the observed modification. It is presented as if the sequence is leading to adhesion to the hydrogels. The reasoning provided is that a similar sequence exists in the sandcastle worm secreted adhesive protein. The sequence may have no importance, and either one or two DOPA containing peptide may lead to similar results. No proof of the importance of sequence is analyzed in this study.
3. The importance of lysine units is not proven. Again, the lack of experiments with control sequences where amine acids are randomized or absent does not provide any insight into the process.
4. In Figure 2, the depiction of the deprotection step of catechol groups is missing. It should be added.
5. If the argument is based on the adhesive nature of the tripeptide end group-containing polymers, force measurements on hydrogel coated substrates should be investigated.

Reviewer #3 (Remarks to the Author):

The manuscript describes a biomimetic strategy to easily functionalize hydrogel materials, using the well-known DOPA-lysine combination. Overall the manuscript could be interesting but it appears uncomplete as it lacks some relevant comments, and characterizations are sometimes incomplete.

1) First of all, in the introduction the authors should underline that there are many other papers in the literature which makes use of molecules terminally functionalized with DOPA or DOPA-lysine combinations for their grafting to surfaces. Such molecules can be adhesive peptides, antifouling molecules (PEG, peptoids) underlining that the novelty is the application of functionalization to hydrogels and the synthesis of a library of Pol with different properties to be used in the approach.

2) The authors underline that the functionalization process is simple and universal. This is true whether it is possible to have commercial Pol available for the functionalization of hydrogels, as their synthesis is not feasible for all the labs. I would consider to add some comments on this point: the functionalization method is simple and could be widely used but it needs the availability of specific functionalization molecules.

3) Additionally it would be interesting to have an explanation on the mechanism of functionalization on the different hydrogels: is there any hypothesis on the interactions between the different functionalization Pol molecules and the different hydrogels? Is it a chemical grafting or based on secondary interactions depending on the hydrogel type?

4) Is there any possibility that the functionalizing Pol may interact each other? Except for Pol 1 and Pol 3, the Pol solution pH is 8.5 and at that pH DOPA tends to be oxidized to DOPAquinone (anyway oxidation occurs even at neutral pH) and may react with amino groups of lysine. Did the authors evaluate such possibility by control experiments without using hydrogels? Indeed the control of pH is very important when using DOPA functionalities. For example in ref [Biofouling. 2008 ; 24(6): 439–448. doi:10.1080/08927010802331829] authors use a pH6 for a similar functionalization on Ti of a peptoid molecules through an anchoring part based on DOPA-lysine combinations.

5) Among missing key characterizations, the efficiency of the process and the functionalization

degree per hydrogels should be addressed. Additionally control experiments are often missing in the biological characterization part (see below for additional details).

Minor comments:

- 6) First sentence of the abstract: some hydrogels do not need any functionalization for applications in tissue engineering, for example gelatin based hydrogels. Please remove "after functionalization" as it depends on the chemistry of the hydrogels.
- 7) Introduction, line 31: should be "a large variety" instead of "varieties"
- 8) Introduction, line 35: The expression "background noise" is not well understandable. Please rephrase it.
- 9) Introduction, line 36: "to exert variable functions" would be better changed into "to exert a variety of functions such as.....[then adding examples]".
- 10) Introduction, line 37: "first before" should be changed into "before".
- 11) Introduction lines 39-41 are poorly understandable and should be rephrased.
- 12) Introduction line 44-45: the sentence overall appears poorly comprehensible especially when you add the part stating from "even though...". It is not clear how then part after "even though..." connects to the initial part of the sentence. Please check and rephrase it.
- 13) Introduction lines 46-48 are poorly comprehensible: Could you specify the type (composition) of wet mineral particles in the sandcastle worms as to give a better description of the natural model? Then the part of sentence after "in which the environment....", which compares the situation in the worm and your functionalization is not well expressed in my opinion. You may simply state that similarly you are using a biomimetic approach to functionalize hydrogels.
- 14) Introduction, line 52: please change into "simplified".
- 15) Legend of Figure 2, line 3: correct "polymerization"

Major Comments

16) In the results the functionalization was characterized by different physicochemical techniques: FTIR, XPS, (qualitative) Kaiser test.

I have concerns with characterizations performed: I believe the work is uncomplete as it misses:

- A) the efficiency of functionalization process. Authors use a certain amount of functionalization Pol per hydrogel type: which is the amount of Pol which functionalizes the different hydrogel types, respect to the one initially used? This should be calculated for all the hydrogel compositions tested.
- B) The degree of functionalization of each hydrogel type: expressed as the percentage of functionalizing Pol respect to the amount of hydrogel polymer
- C) Do you have an idea of the stability of the functionalization? Did you perform release tests of the functionalizing molecules as a function of time?
- D) The Kaiser tests was only performed qualitatively instead of quantifying the amino groups which could provide additional information.
- E) As you also mentioned hydrogel functionalization using freeze dried samples, could you please comment on the permeability of hydrogels to the Pol molecules? Do you expect there can be any constrains in the diffusion of such molecules in the bulk of the hydrogel material during the functionalization (and rehydration from freeze dried substrates), as a function of hydrogel concentration and type? This implies you should have idea of hydrogel permeability as a function of hydrogel concentration and type, for the different hydrogel types you used. I believe you should at least comment on this point considering the possible application of the technique for bulk functionalization.

17) For the biological results:

- A) Figure 4 concerning antibacterial properties: control experiments are missing (positive and negative control samples which are known to kill bacteria completely or to support bacteria exerting no bactericidal effect) so it is not possible to evaluate the bactericidal efficiency of the non-functionalized and functionalized hydrogels without the two controls (you cannot use the non-functionalized material as a control as we need substrates with known ability to inhibit or to support bacteria to be used as controls).
- B) Always in the legend of Figure 4, please remember the name of the test to evaluate bacterial killing and the time used for the evaluation.
- C) Figure 4c: there is a control supporting bacteria: what is it? Please indicate it in the legend and be sure to have inserted it in the exp. Part.
- D) Line 141: you state that hydrogels you used have no pre-organized reactive groups but indeed

they may have: ALG has -COOH and PVA has -OH... Maybe I did not correctly understand what you wanted to say. Could you please rephrase?

E) Figure 5: please in the legend write the test which has been carried out (calcein staining). Also for the in vitro cell tests a positive control (Petri Dish) is usually used and should be added.

F) Minor note at line 168: please delete "the" before "how the"

G) Concerning in vivo studies: again control experiments are missing (e.g. wound without treatment). Please add controls or introduce a reasonable explanation for their missing.

Additionally it is not clear how many mice were analyzed in the experimental part.

18) Experimental part:

A) Supplier of ALG and PEGDA are missing

B) In the universal hydrogel modification, the ratio between the used hydrogel material and Pol amount is missing as to allow the readers repeating the experiments. For example you describe Pol concentration but we do not know the Pol solution volume you used and the amount of hydrogel material immersed in that Pol Solution volume. This is important to allow reproduction of results.

C) Line 226: you state "until the unmodified polymer was completely removed" but how did you evaluate complete removal?

D) For XPS: please indicate if you analysed the external surface

E) Same comment as above for FTIR analysis

F) Kaiser test should be "quantitative" as it can be such. Why did you perform it as a qualitative test?

G) "Cell adhesion" tests should be changed into "Cell viability test" as you used LIVE/DEAD assay.

H) Lines 240-247: Please specify the procedure for sterilization before cell tests.

I) Lines 240-247: control experiments are not indicated (positive and negative controls)

J) Bactericidal tests: positive and negative controls should be inserted.

K) In vivo wound healing: number of mice is missing; control tests are missing.

Although we cannot offer to publish your paper in Nature Communications, the work may be appropriate for another journal in the Nature Research portfolio. If you wish to explore suitable journals and transfer your manuscript to a journal of your choice, please use our <https://mts-ncomms.nature.com/cgi-bin/main.plex?el=A3S3BmLJ4B2HwrB2X4A9ftd5fy6Lt8NK4aWZPsTQCnOQZ> manuscript transfer portal. If you transfer to Nature-branded journals or to the Communications journals, you will not have to re-supply manuscript metadata and files. This link can only be used once and remains active until used.

All Nature Research journals are editorially independent, and the decision to consider your manuscript will be taken by their own editorial staff. For more information, please see our http://www.nature.com/authors/author_resources/transfer_manuscripts.html?WT.mc_id=EMI_NPG_1511_AUTHORTRANSF&WT.ec_id=AUTHOR manuscript transfer FAQ page. Note that any decision to opt in to In Review at the original journal is not sent to the receiving journal on transfer. You can opt in to [In Review](https://www.nature.com/nature-research/for-authors/in-review) at receiving journals that support this service by choosing to modify your manuscript on transfer. In Review is available for primary research manuscript types only.

RESPONSE TO REVIEWERS

Reviewer #1 (Remarks to the Author):

This manuscript from Zhang et al. generates polymers based on a dibutylamine-DOPA-lysine-DOPA peptide and applies them to reduce bacterial adhesion, increase mammalian cell-line adhesion and improve skin wound healing. The strengths of the work are the range of different polymers synthesized through different kinds of polymerization strategies. I considered that the authors gave sufficient information to reproduce the work. However, it should be noted that the authors do not demonstrate the benefit of the sandcastle-worm based sequence. The negative control polymers (Fig. 3a) lack any DOPA functionality. The use of DOPA or dopamine in proteins or synthetic polymers to improve adhesion has been widely demonstrated in the field. For example, it was reviewed back in 2017:

<https://pubs.rsc.org/en/content/articlehtml/2017/ra/c7ra06743g>

There have also been extensive investigations of the biological systems tested here, so there is no evidence to conclude that their polymers have enhanced performance over what has been explored elsewhere. Therefore, I suggest that the work does not show sufficient novelty to be appropriate for Nature Communications.

Response: We thank the reviewer for the comments that inspire us to provide more support on the benefit of the sandcastle-worm based sequence. In our revision, we synthesized several DOPA-containing adhesive peptides to compare their binding affinity to hydrogels, using PEG hydrogel as a model. To analyze the benefit of the sandcastle-worm based sequence we synthesized peptides YKY and YY (Fig. 3c-d and Supplementary Scheme 15), which have one and two less primary amine groups respectively than does the DbayKY peptide (Supplementary Scheme 14). We also synthesized peptide KYK (Fig. 3e and Supplementary Scheme 15), with only one DOPA, to analyze the importance of DOPA in the adhesive tripeptide DbayKY. To analyze the result of randomized amino acids within the adhesive peptide sequence, we synthesized peptide KKYY (Fig. 3f and Supplementary Scheme 15). The obtained peptides were individually modified to atomic force microscope (AFM) cantilevers (Supplementary Fig. 1) to measure the adhesion strength of these peptides to PEG hydrogels, using AFM based single-molecule force spectroscopy (SMFS) approach. All SMFS results were added into Fig. 3 of our revised manuscript, as shown below.

Our results indicate that the adhesion strength of the peptide YY (~71 pN) without any amine group is significantly lower than that of DbayKY (~251 pN), and that the adhesion strength of YKY (~210 pN) with the removal of Dbay is also lower than that of DbayKY. These results indicate that the primary amine group within Dbay and lysine play a synergetic role with catechol to promote the adhesion strength to hydrogels. The observed importance of amine groups in our study is consistent to the conclusion in the literature that introduction of amine groups (such as lysine) to DOPA-containing peptides can enhance the adhesion strength (*Science* **2015**, 349, 628-632; *J. Am. Chem. Soc.* **2016**, 138, 9013-9016.). The adhesion strength of KYK (~203 pN) is also lower than that of DbayKY (~251 pN), indicating the importance of DOPA in the sequence. All these show that two amine groups (from Dbay and lysine) and two catechol groups (from two DOPA units) are important for the peptide DbayKY to have strong adhesion to hydrogels, as inspired by the 1:1 DOPA:Lysine component in the adhesive protein Pc-1 from sandcastle worm. We also found the adhesion strength of KKYY (~246 pN) is comparable to that of DbayKY (~251 pN), indicating that strict sequence of DOPA and lysine was not important for this adhesive peptide.

Fig. 3.

Supplementary Scheme 15.

Supplementary Scheme 14.

Supplementary Fig. 1.

Minor points:

The use of “Universal” in the title is not adequately justified. The authors would need to explain how the sample of surfaces that are tested represent everything that could be encountered.

Also, it is not to be encouraged for authors to claim how remarkable their own work is (in the abstract).

Response: We thank the reviewer for these comments and we have made changes accordingly. We removed the “Universal” in the title. We also modified the abstract, such as removing the words “universal” and “remarkable”. We followed the reviewer’s request and included an explanation into the Methods part of our revised manuscript that “To demonstrate that our direct modification strategy is compatible with diverse types of hydrogels, we tested five different types of hydrogels, including PEG hydrogel (chemical crosslinking, without functional groups), PHEMA hydrogel (chemical crosslinking, with hydroxyl functional groups), PSBMA hydrogel (chemical crosslinking in the back bone and ion pair interactions between zwitterions), PVA hydrogel (physical crosslinking based on the amphiphilic structure), and sodium alginate hydrogel (physical crosslinking based on ion chelation). These hydrogels can represent the commonly used crosslinking types of hydrogels for the purpose of our study.”

Explain DbayKY in the abstract.

“This tripeptide can initiate diverse types of polymerizations”

It is not the peptide that initiates polymerization but functional groups that can be attached to the peptide.

Response: We thank the reviewer for reminding us on this. We have added the explanation for the peptide as “dibutylamine-DOPA-lysine-DOPA tripeptide”. We also modified our original description “This tripeptide can initiate diverse types of polymerizations” to that “This tripeptide can be easily modified with various functional groups to initiate diverse types of polymerizations and to provide functional polymers with a terminal adhesive tripeptide...”.

Main text does not refer to Supp Figures where compounds are characterized.

Response: We thank the reviewer for pointing out this. In our revised manuscript, main text refers to Supplementary Figures where compounds are characterized.

“design the adhesive peptide as dibutylamine-DOPA-lysine-DOPA (Db \underline{a} YKY), a short tripeptide”
Y is the standard symbol for tyrosine, not DOPA.

Response: We thank the reviewer for pointing out this. In our revised manuscript, we used Y for DOPA as reported in literature and modified all “Db \underline{a} YKY” to “Db \underline{a} YKY”.

Fig. 3a has Db \underline{a} KYK, when perhaps mean Db \underline{a} YKY

Response: We thank the reviewer for pointing out this typo. We have changed Db \underline{a} KYK to Db \underline{a} YKY in Fig. 4a of our revised manuscript (Fig. 3a in our original manuscript).

Fig. 5a legend: describe how seeing the cells

Response: We thank the reviewer for this suggestion. In our revised manuscript, we added description in the legend of Fig. 6 (Fig. 5 in our original manuscript) that “Cells were treated with LIVE/DEAD staining using calcein AM (green) and ethidium homodimer-1 (red), followed by imaging under a fluorescence microscope.”.

Reviewer #2 (Remarks to the Author):

The authors outline a universal strategy for obtaining functional hydrogels using a modular approach towards the functionalization of hydrogels. Authors utilize a tripeptide sequence consisting of DOPA and Lysine units as a terminal group for various types of polymers that possess various properties such as antibacterial effect through the presence of quaternary ammonium groups as side chains. Overall, the process of tailoring properties of the hydrogels using the presented approach appears to be simple and straightforward. While the approach seems to provide hydrogels with desired properties, the underlying science has not been investigated in a rigorous manner. Such shortcomings as explained below makes this report very superficial. The study should have been conducted in a manner that provides a more in-depth analysis of the process, as well as furnishes materials with fine-tuned tailorable properties, rather than broad overall attributes. As presented, the study appears superficial and lacks the rigor for publication in this journal.

Response: We thank the reviewer for the comments that inspire us to do more in-depth study and analyze the result in a rigorous manner, as shown below to address specific questions.

Major Comments:

1. The argument that the present approach has superior attributes to the conventional approach where reactive groups are incorporated into hydrogels during their fabrication and are functionalized post-gelation is not correct. The conventional approach gives a level of control over the functionalization process. The extent of functionalization can be tailored. In the present approach, while authors demonstrate various modifications, no level of control over the extent of modification is demonstrated. Hence, as a result, there is also no level of control over the functional aspect. Such a level of control should be demonstrated.

Response: We thank the reviewer for the comment and the suggestion, which inspire us to explore and demonstrate the level of control. In our revision, we synthesized a fluorescent molecule **DbayKY-OEG8-Rh** (Rh represents rhodamine fluorophore) and modified it to PEG hydrogels using different concentration. The synthetic route of the fluorescent molecule and the quantification assay on fluorescent molecule-modified hydrogel were shown below (Supplementary Scheme 14, 17 and Supplementary Fig. 7a-b in our revised manuscript). The results showed that our method can control over the extent of modification by adjusting the concentration of the tripeptide for modification. We can obtain incrementally increased amount of modified molecules to the hydrogel with the increase of tripeptide concentration, from 0.0625 to 0.25 mg/mL.

Supplementary Scheme 14.

Supplementary Scheme 17

Supplementary Fig. 7a.

Supplementary Fig. 7b.

2. No clear data has been provided to support the reason behind the observed modification. It is presented as if the sequence is leading to adhesion to the hydrogels. The reasoning provided is that a similar sequence exists in the sandcastle worm secreted adhesive protein. The sequence may have no importance, and either one or two DOPA containing peptide may lead to similar results. No proof of the importance of sequence is analyzed in this study.

Response: We thank the reviewer for this question that inspired us to further explore the adhesive DbayKY peptide, especially the importance of the sequence and component in this adhesive peptide. In our revision, we synthesized several DOPA-containing adhesive peptides to compare their binding affinity to hydrogels, using PEG hydrogel as a model (Fig. 3 in our revised manuscript). To analyze the importance of sequence we synthesized peptides YKY and YY (Fig. 3c-d and Supplementary Scheme 15), which have one and two less primary amine groups respectively than does the DbayKY peptide (Supplementary Scheme 14). We also synthesized peptide KYK (Fig. 3e and Supplementary Scheme 15), with only one DOPA, to analyze the importance of DOPA in the adhesive tripeptide DbayKY. To analyze the result of randomized the amino acids within the adhesive peptide sequence, we synthesized peptide KKYY (Fig. 3f and Supplementary Scheme 15). The obtained peptides were individually modified to atomic force microscope (AFM) cantilevers (Supplementary Fig. 1) to measure the adhesion strength of these peptides to PEG hydrogels, using AFM based single-molecule force spectroscopy (SMFS) approach. The peptide synthesis and modification of AFM cantilever were shown below. All SMFS results were added into Fig. 3 of our revised manuscript, as shown below.

Our results indicate that the adhesion strength of the peptide YY (~71 pN) without any amino group is significantly lower than that of DbayKY (~251 pN), and that the adhesion strength of YKY (~210 pN) with the removal of Dbay (removal of one primary amine group) is also lower than that of DbayKY. These results indicate that the primary amine group within Dbay and lysine play an important synergetic role with catechols to promote the adhesion strength to hydrogels. The observed importance of amine groups in our study is consistent to the conclusion in the literature that introduction of amine groups (such as lysine) to DOPA-containing peptides can enhance the adhesion strength (*Science* **2015**, 349, 628-632; *J. Am. Chem. Soc.* **2016**, 138, 9013-9016.). The adhesion strength of KYK (~203 pN) is also lower than that of DbayKY (~251 pN), indicating the importance of DOPA in the sequence. All these show that two amine groups (from Dbay and

lysine) and two catechol groups (from two DOPA units) are important for the peptide DbaYKY to have strong adhesion to hydrogels, as inspired by the 1:1 DOPA:lysine component in the adhesive protein Pc-1 from sandcastle worm.

We also found the adhesion strength of KKYY (~246 pN) is comparable to that of DbaYKY (~251 pN), indicating that strict sequence of DOPA and lysine were not important for this adhesive peptide, as long as two DOPA and primary amine containing residues (Db or K) are present.

Fig. 3.

Supplementary Scheme 15.

Supplementary Scheme 14.

Supplementary Fig. 1.

3. The importance of lysine units is not proven. Again, the lack of experiments with control sequences where amine acids are randomized or absent does not provide any insight into the process.

Response: We thank the reviewer for the comment that inspires us to explore the importance of lysine and the sequence of amino acid. To analyze the importance of lysine unit (the primary amine group) we synthesized peptides YKY and YY, which have one and two less primary amine groups respectively than does the DbayKY peptide. We also synthesized peptide KKYY as a randomized control for comparison. The results showed that lysine units were important for the adhesion, and that the strict sequence of DOPA and lysine were not important for this adhesive peptide. A detailed explanation on these was provided in our response to the reviewer's question 2 above.

4. In Figure2, the depiction of the deprotection step of catechol groups is missing. It should be added.

Response: We thank the reviewer for reminding us on this. We have added the depiction of the deprotection step of catechol groups in Fig. 2 of our revised manuscript.

5. If the argument is based on the adhesive nature of the tripeptide end group-containing polymers, force measurements on hydrogel coated substrates should be investigated.

Response: We have followed this suggestion to do the force measurement on hydrogel, as described above in our response to the 2nd question from the reviewer. These studies indicate the importance of two amine groups (from Dbay and K) and two catechol groups (from two Y units) for the adhesive peptide DbayKY

to have strong adhesion to hydrogels, as inspired by the 1:1 DOPA:lysine component in the adhesive protein Pc-1 from sandcastle worm.

Reviewer #3 (Remarks to the Author):

The manuscript describes a biomimetic strategy to easily functionalize hydrogel materials, using the well-known DOPA-lysine combination. Overall the manuscript could be interesting but it appears uncomplete as it lacks some relevant comments, and characterizations are sometimes incomplete.

Response: We thank the reviewer for the comment and detailed suggestions below. By following these suggestions, we have strengthened our manuscript substantially.

1) First of all, in the introduction the authors should underline that there are many other papers in the literature which makes use of molecules terminally functionalized with DOPA or DOPA-lysine combinations for their grafting to surfaces. Such molecules can be adhesive peptides, antifouling molecules (PEG, peptoids) underlining that the novelty is the application of functionalization to hydrogels and the synthesis of a library of Pols with different properties to be used in the approach.

Response: We thank the reviewer for this suggestion. We have added a description into the introduction part of our revised manuscript that “Many of these studies use molecules that are terminally functionalized with 3,4-dihydroxy-L-phenylalanine (DOPA) or DOPA-lysine combinations to realize surfaces modification. The function of these molecules can be cell adhesive, antimicrobial and antifouling.”. Relevant references (new ref. 42-56) were also included into our revised manuscript.

2) The authors underline that the functionalization process is simple and universal. This is true whether it is possible to have commercial Pol available for the functionalization of hydrogels, as their synthesis is not feasible for all the labs. I would consider to add some comments on this point: the functionalization method is simple and could be widely used but it needs the availability of specific functionalization molecules.

Response: We thank the reviewer for the comment. To modify hydrogels with functional polymers, normally the functional polymers are synthesized in the lab according to the purpose of specific application. Therefore, we demonstrate in our study that the DbaYKY-NH_2 terminated molecules can be used as the initiators for several types of classical polymerization to prepare diverse functional polymers. All steps are simple chemical reactions that can be easily operated in the lab.

In response to the reviewer’s concern about modifying some commercialized functional polymers to hydrogels, we also did an extra demonstration in our revision. We demonstrated that DbaYKY-NH_2 can be easily prepared from solid-phase synthesis plus a simple coupling step (Supplementary Scheme 3). The obtained DbaYKY-NH_2 , can be easily attached to commercially available functional polymers via the terminal amine group within DbaYKY-NH_2 .

Nevertheless, we also take the reviewer’s suggestion and add a comment on this point into our revised manuscript that “The functionalization method is simple and could be widely used if with the availability of specific functionalization molecules, as our demonstration in both solution-phase and solid-phase synthesis.”

Supplementary Scheme 3.

3) Additionally it would be interesting to have an explanation on the mechanism of functionalization on the different hydrogels: is there any hypothesis on the interactions between the different functionalization Pol molecules and the different hydrogels? Is it a chemical grafting or based on secondary interactions depending on the hydrogel type?

Response: We hypothesize that the functionalization mechanism of the DbYKY terminated polymers to hydrogels is attributed to multiple interactions including hydrogen bonding, cation- π stacking and charge interactions (*J. Polym. Sci. Pol. Chem.* **2016**, 55(1): 9-33; *Angew. Chem. Int. Ed.* **2019**, 58, 696-714; *Chem. Soc. Rev.* **2020**, 49, 3605). For all polymers (Pol-1 to Pol-8) functionalized to five types of hydrogels as showed below, we hypothesize that hydrogen bonding between DbYKY terminated polymers to hydrogels are always existing because hydrogen bond is widely found in mussel-inspired adhesive molecules that leverage two neighboring hydroxyl groups of catechol moieties as the donors/acceptors (*Chem. Soc. Rev.* **2020**, 49, 3605). The ether in PEG hydrogel and the sulfonic acid in PSBMA hydrogel are hydrogen bonding acceptor; the hydroxy groups in PHEMA, PVA and ALG hydrogels are both hydrogen bonding donors and acceptors. The aromatic rings can also form cation- π interactions with positively charged ions, which is one of the strongest non-covalent interactions in water (*J. Polym. Sci. Pol. Chem.* **2016**, 55(1): 9-33). Therefore, the cation- π interaction is indispensable because of the presence of amine groups and catechol groups in the polymer chains. The DbYKY terminated polymers have positively charged amine groups, therefore, can interact electrostatically with hydrogels bearing negatively charged groups, such as PSBMA and ALG. A brief summary of these discussions was also included into our revised manuscript.

FIGURE 6 Schematic representation of peptide monomer (A) and dimer (B) adhering to mica substrates. The peptide dimer with a longer chain length is capable of bridging two mica surfaces and demonstrated enhanced adhesion.

(*J. Polym. Sci. Pol. Chem.* **2016**, *55*(1): 9-33)

4) Is there any possibility that the functionalizing Pol may interact each other? Except for Pol 1 and Pol 3, the Pol solution pH is 8.5 and at that pH DOPA tends to be oxidized to DOPA quinone (anyway oxidation occurs even at neutral pH) and may react with amino groups of lysine. Did the authors evaluate such possibility by control experiments without using hydrogels? Indeed the control of pH is very important when using DOPA functionalities. For example in ref [Biofouling. 2008 ; 24(6): 439–448. doi:10.1080/08927010802331829] authors use a pH6 for a similar functionalization on Ti of a peptoid molecules through an anchoring part based on DOPA-lysine combinations.

Response: We thank the reviewer for this question. We followed the suggestion and analyzed the possible change of the adhesive peptide over time without hydrogel. An adhesive peptide *Db*aYKY-Ac was synthesized as a model molecule (Supplementary Scheme 18), and then was dissolved in a pH 8.5 Tris buffer for 0-24 h to analyze the change of functional groups. We have added our observation into our revised manuscript that “HPLC results indicated a gradually weakened peak of *Db*aYKY-Ac itself, and the appearance of some new peaks (Supplementary Fig. 9). UV-vis results showed a new peak at around 360 nm, which may come from the change of catechol groups, such as Michael addition between amine and the catechol group (Supplementary Fig. 10). MALDI results further confirmed that both intramolecular and intermolecular Michael addition reaction exist in the *Db*aYKY-Ac solution for hydrogel modification (Supplementary Fig. 11). It’s noteworthy that no Schiff base intermediate was observed from MALDI analysis.”, as shown below.

Our study indicates that the amine and catechol groups within the adhesive peptide can form cross-linking at alkaline pH to obtain high density of functionalization, consistent to the report in literature (*Langmuir* **2012**, *28*(18): 7258-7266). Promoting cell adhesion and obtaining antibacterial functions of hydrogels often require sufficient density of functional molecules. Therefore, we chose to use the pH 8.5 condition to have high modification density. If a research requires a monolayer modification, it is recommended to use a pH 6.0 condition for modification to minimize the catechol-derived crosslinking and multilayer modification as described in precedent literatures (*Biofouling* **2008**, *24*(6): 439-448; *Langmuir* **2012**, *28*(4): 2288-2298.).

Supplementary Scheme 18.

Supplementary Fig. 9.

Supplementary Fig. 10.

Supplementary Fig. 11.

5) Among missing key characterizations, the efficiency of the process and the functionalization degree per hydrogels should be addressed. Additionally control experiments are often missing in the biological characterization part (see below for additional details).

Response: We thank the reviewer for the suggestion. To analysis efficiency of the process and the functionalization degree per hydrogel, we conducted quantitative analysis on hydrogel modification as shown below. Using Pol-7 as an example to quantify its modification to PVA and ALG hydrogels individually to quantify the amount of modified polymers by fluorescamine method (Supplementary Fig. 5a in our revised manuscript). We used Pol-7 as the model polymer for quantification because Pol-7 has many primary amine groups, which can be quantified using fluorescamine method with high sensitivity. The result showed that the amount of Pol-7 modified to PVA and ALG hydrogels was ~0.0515% and ~ 0.203% hydrogel polymer respectively, as suggested to analyze in the reviewer's question 16B below (Supplementary Fig. 5b in our revised manuscript). PVA hydrogel has no charge, while ALG hydrogel has a large amount of negatively charged carboxyl groups and may electrostatically attract more Pol-7.

Hydrogels (PEG, PHEMA and PSBMA hydrogel) formed via covalent bonding cannot be dissolved and thus are not suitable for quantification using above fluorescamine method. To analyze the efficiency of the process and the functionalization degree per hydrogels, we synthesized rhodamine (Rh)-modified adhesive molecule DbayKY-OEG8-Rh to functionalize hydrogels (PEG, PHEMA, PSBMA, PVA and ALG hydrogel) and quantified the efficiency of functionalization by measuring the fluorescence intensity as shown below (Supplementary Fig. 7a in our revised manuscript). First, we took PEG hydrogel as an example to analyze the efficiency of the modification process. An incrementally increased concentration of DbayKY-OEG8-Rh, from 0.0625 to 1.0 mg/mL, was used to modify the molecule to PEG hydrogels. The amount of modification increases with the increase of the adhesive molecule concentration and reaches a plateau when using the adhesive molecule at a concentration of 0.25 mg/mL (Supplementary Fig. 7b in our revised manuscript). Therefore, we used 0.5 mg/mL of DbayKY-OEG8-Rh for further modification efficiency study on five types of hydrogels (PEG, PHEMA, PSBMA, PVA and ALG hydrogel) within our study. The result showed that the amount of this molecule modified to hydrogels is ~0.0225% for PEG hydrogel, ~0.104% for PHEMA hydrogel, ~0.0464% for PSBMA hydrogel, ~0.0197% for PVA hydrogel, and ~0.0833% for ALG hydrogel (Supplementary Fig. 7c in our revised manuscript). The difference in the efficiency for functionalization on different hydrogels may come from multiple factors, including the difference in dry weight content of different hydrogels (such as ~10% PVA in PVA hydrogel and ~50% PHEMA in PHEMA hydrogel), the difference in crosslinking methods (chemical crosslinking for PEG, PHEMA and PSBMA, and physical crosslinking for PVA and ALG), the difference in charge (zwitterionic for PSBMA, negative charge for ALG, and uncharged for PEG, PHEMA and PVA hydrogels), and the difference in structure characteristics of these polymers to form hydrogels.

Supplementary Fig. 5.

Supplementary Fig. 7a.

Supplementary Fig. 7b.

Supplementary Fig. 7c.

Minor comments:

6) First sentence of the abstract: some hydrogels do not need any functionalization for applications in tissue engineering, for example gelatin based hydrogels. Please remove “after functionalization” as it depends on the chemistry of the hydrogels.

Response: We follow this suggestion and removed “after functionalization” from the text of our revised manuscript.

7) Introduction, line 31: should be “a large variety” instead of “varieties”

Response: We thank the reviewer for pointing out this. We have changed “a large varieties of ...” to “a large variety of ...” in our revision.

8) Introduction, line 35: The expression “background noise” is not well understandable. Please rephrase it.

Response: We thank the reviewer for pointing out this. We have changed “which provide low background noise favorable for biological recognition ...” to “which provide low fouling favorable for biological recognition ...”.

9) Introduction, line 36: “to exert variable functions” would be better changed into “to exert a variety of functions such as.....[then adding examples]”.

Response: We agree with the reviewer on this and have changed “to exert variable functions” to “exert a variety of functions such as...”.

10) Introduction, line 37: “first before” should be changed into “before”.

Response: We agree with the reviewer on this and have changed “first before” to “before” in our revision.

11) Introduction lines 39-41 are poorly understandable and should be rephrased.

Response: We thank the reviewer for pointing out this. We have changed “Nevertheless, current methods for functional hydrogel preparation are polymer and functional group specific, and require synthesis of functionalized or reactive group pre-organized polymers individually for each type of hydrogel, i.e. very less flexibility (Fig. 1a).” in our original manuscript to “Nevertheless, current methods to prepare functional hydrogels typically require incorporating reactive groups, such as azide, into the substrates and then tethering functional molecules to them via the reactive groups. However, for a substrate incorporated with a particular type of reactive group (such as azide), only functional molecules bearing specific reactive groups (such as alkynes) can be used to undergo the further functionalization step. Therefore, both the polymers pre-modified with specific reactive/binding sites and the functional molecules bearing the matching reactive/binding sites need to be synthesized to prepare a particular type of functional hydrogel, which highly limited the flexibility to develop functional hydrogels. (Fig. 1a).”

12) Introduction line 44-45: the sentence overall appears poorly comprehensible especially when you add the part stating from “even though...”. It is not clear how then part after “even though...” connects to the initial part of the sentence. Please check and rephrase it.

Response: We thank the reviewer for this suggestive comment. In our revision, we removed the part of “even though adhesive materials have been extensively studied in recent years especially the mussel-inspired adhesive materials”. We add new description about precedent literatures on DOPA-containing adhesive materials that “In recent years, adhesive materials have been extensively studied especially the mussel-inspired adhesive materials. Many of these studies use molecules that are terminally functionalized with 3,4-dihydroxy-L-phenylalanine (DOPA) or DOPA-lysine combinations to realize surfaces modification. The function of these molecules can be cell adhesive, antimicrobial and antifouling. Nevertheless, it is unknown if these types of adhesive materials can be used to modify inert hydrogels directly to obtain diverse functions.”

13) Introduction lines 46-48 are poorly comprehensible: Could you specify the type (composition) of wet mineral particles in the sandcastle worms as to give a better description of the natural model? Then the part of sentence after “in which the environment...”, which compares the situation in the worm and your functionalization is not well expressed in my opinion. You may simply state that similarly you are using a biomimetic approach to functionalize hydrogels.

Response: We thank the reviewer for the suggestions. The wet mineral particles are composed of sand,

shell, etc. (*J. Biol. Chem.* **2005**, 280(52): 42938-42944.) We included this information into our revised manuscript.

We also took the reviewer's suggestion and changed the part of sentence after "in which the environment is similar to that in modifying functional molecules to polymer chains within hydrogels under a wet environment." to "Similarly we are using a biomimetic approach to functionalize hydrogels."

14) Introduction, line 52: please change into "simplified".

Response: We have changed "simplify" to "simplified" in our revised manuscript.

15) Legend of Figure 2, line 3: correct "polymerization"

Response: We thank the reviewer for pointing out this typo. We have corrected our text to "polymerization".

Major Comments

16) In the results the functionalization was characterized by different physicochemical techniques: FTIR, XPS, (qualitative) Kaiser test.

I have concerns with characterizations performed: I believe the work is incomplete as it misses:

A) the efficiency of functionalization process. Authors use a certain amount of functionalization Pol per hydrogel type: which is the amount of Pol which functionalizes the different hydrogel types, respect to the one initially used? This should be calculated for all the hydrogel compositions tested.

Response: We thank the reviewer for the suggestion. We have conducted the quantitative analysis of hydrogel modification. We described the quantitative methods and results in details in our response to the 5th question of the reviewer above.

B) The degree of functionalization of each hydrogel type: expressed as the percentage of functionalizing Pol respect to the amount of hydrogel polymer

Response: We thank the reviewer for this question. We have added the quantitative test of each hydrogel. The experimental details have been provided in our response to the 5th question from the reviewer. Using Pol-7 as an example to quantify its modification to PVA and ALG hydrogels, we dissolved the Pol-7 modified PVA and ALG hydrogels individually to quantify the amount of modified polymers by fluorescamine method (Supplementary Fig. 5a in our revised manuscript). We used Pol-7 as the model polymer for quantification because Pol-7 has many primary amine groups, which can be quantified using fluorescamine method with high sensitivity. The result showed that the weight percentage of functionalizing Pol-7 respect to the amount of hydrogel polymer is ~0.0515% and ~ 0.203%, respectively, for PVA and ALG hydrogels (Supplementary Fig. 5b in our revised manuscript).

Supplementary Fig. 5.

Hydrogels (PEG, PHEMA and PSBMA hydrogel) formed via covalent bonding cannot be dissolved and thus are not suitable for quantification using above fluorescamine method. To analyze the efficiency of the process and the functionalization degree per hydrogels, we synthesized rhodamine (Rh)-modified adhesive molecule DbaYKY-OEG8-Rh to functionalize hydrogels (PEG, PHEMA, PSBMA, PVA and ALG hydrogel) and quantified the efficiency of functionalization by measuring the fluorescence intensity as shown below (Supplementary Fig. 7a). We used 0.5 mg/mL of DbaYKY-OEG8-Rh for modification efficiency study on five types of hydrogels (PEG, PHEMA, PSBMA, PVA and ALG hydrogel) within our study. The result showed that the weight percentage of functionalizing DbaYKY-OEG8-Rh respect to the amount of hydrogel polymer is ~0.0225%, ~0.104%, ~0.0464%, ~0.0197% and ~0.0833%, respectively for PEG, PHEMA, PSBMA, PVA and ALG hydrogels (Supplementary Fig. 7c in our revised manuscript). It's noteworthy that the modifying ratios between pol-7 and DbaYKY-OEG8-Rh are very close for PVA and ALG hydrogels, with tethered Pol-7/DbaYKY-OEG8-Rh being 2.61 and 2.44, respectively, for PVA and ALG hydrogels. This result indicates that we can deduce the rough percentage of functionalizing polymers respect to the amount of hydrogel polymer, by using the adhesion quantification result of DbaYKY-OEG8-Rh.

Supplementary Fig. 7a.

Supplementary Fig. 7c.

The difference in the efficiency for functionalization on different hydrogels may come from multiple factors, including the difference in dry weight content of different hydrogels (such as ~5.9% PVA in PVA hydrogel and ~55.9% PHEMA in PHEMA hydrogel), the difference in crosslinking methods (chemical crosslinking for PEG, PHEMA and PSBMA, and physical crosslinking for PVA and ALG), the difference in charge (zwitterionic for PSBMA, negative charge for ALG, and uncharged for PEG, PHEMA and PVA hydrogels), and the difference in structure characteristics of these polymers to form hydrogels.

C) Do you have an idea of the stability of the functionalization? Did you perform release tests of the functionalizing molecules as a function of time?

Response: We thank the reviewer for this question. We modified the rhodamine-tethered fluorescent molecule DbaYKY-OEG8-Rh to PEG hydrogels, and did stability tests by monitoring the fluorescent intensity.

We tested the stability of functionalization under two conditions: 1) treating functionalized hydrogels with proteinase K for 2 h at 37 °C; 2) immersing functionalized hydrogels in PBS under shaking for 24 h. The result showed that the functionalization was stable at both two conditions, as shown below (Supplementary Fig. 7d in our revised manuscript) in our revised manuscript). We also checked possible release of the functionalized molecules from hydrogels but found no detectable DbaYKY-OEG8-Rh molecules, which indicates negligible release of the functionalizing molecules.

Supplementary Fig. 7d.

D) The Kaiser tests was only performed qualitatively instead of quantifying the amino groups which could provide additional information.

Response: We agree with the reviewer that quantitative test of functionalized molecule was important. We have attempted to quantify the amine groups in the hydrogels using Kaiser test. However, it is very difficult to obtain reliable calibration curve for quantification using the Kaiser test. Therefore, we conducted quantifying analysis using more sensitive and reliable methods, as described in our response to the reviewer's question 16B above.

E) As you also mentioned hydrogel functionalization using freeze dried samples, could you please comment on the permeability of hydrogels to the Pol molecules? Do you expect there can be any constrains in the diffusion of such molecules in the bulk of the hydrogel material during the functionalization (and rehydration from freeze dried substrates), as a function of hydrogel concentration and type? This implies you should have idea of hydrogel permeability as a function of hydrogel concentration and type, for the different hydrogel types you used. I believe you should at least comment on this point considering the possible application of the technique for bulk functionalization.

Response: We thank the reviewer for this question and comment. In fact, the pore size of the hydrogels in our study is too small for polymers, such as Pol-7, to diffuse efficiently if without freeze-drying treatment. For example, hydrogels prepared from PEG 2K, 4K, and 8K diacrylates were impermeable for proteins with a size equal to or larger than myoglobin because the pore size of these hydrogels is only 1.4-3.4 nm (*Biomaterials* **1998**, 19, 1287-1294. See the table shown below for data). We also tried Pol-7 to modify the PHEMA hydrogel that was not subjected to freeze-drying treatment, and as we can predict we cannot achieve functionalization inside the hydrogel.

To increase the permeability of polymers into hydrogels, we utilized the freeze-drying treatment for hydrogels before polymer modification because the formation of ice crystals in the freezing process will cause partial collapse of the hydrogel structure, partially destroying the three-dimensional network structure of hydrogels and producing larger pores (*Biomaterials* **1999**, 20, 1339-1344; *Biomaterials* **2002**, 23, 1205-1212). For example, the pore size (280 μ m) of a hydrogel after freeze-drying is several orders of magnitude greater

than the mesh size of the polymer network (90 nm) (*Acta Biomaterialia* **2019**, 94 195–203). In short, during the functionalization process the diffusion of polymers to hydrogels is constrained for hydrogels without freeze-drying treatment; however, polymer diffusion to hydrogels is not constrained for hydrogels after freeze-drying treatment.

As suggested by the reviewer, we added a comment on this into our revised manuscript that “Hydrogels with large pore size for efficient polymer diffusion into hydrogels can be used directly for functionalization with the reported method here; polymers with small pore size can be pretreated, such as freeze-drying, to obtain enlarged pore size that enables efficient polymer diffusion into hydrogels and functionalization. Therefore, this adhesive peptide strategy can be used to meet the requirement for bulk hydrogel functionalization with diverse applications such as regulating cell migration into hydrogels in tissue engineering and cell encapsulation.”

Table 2

The effect of PEG diacrylate molecular weight and concentration on the diffusion of biological molecules

	Diffusion Coefficient (cm ² s ⁻¹) mean ± SD, n ≥ 4				
	Vitamin B ₁₂	Myoglobin	Ovalbumin	Albumin	IgG
10% PEG 2K diacrylate	1.4 × 10 ⁻⁶ ± 1.8 × 10 ⁻⁷	BLD	BLD	BLD	BLD
20% PEG 2K diacrylate	7.6 × 10 ⁻⁷ ± 1.2 × 10 ⁻⁷	BLD	BLD	BLD	BLD
30% PEG 2K diacrylate	5.3 × 10 ⁻⁷ ± 1.8 × 10 ⁻⁷	BLD	BLD	BLD	BLD
10% PEG 4K diacrylate	1.6 × 10 ⁻⁶ ± 1.7 × 10 ⁻⁷	BLD	BLD	BLD	BLD
20% PEG 4K diacrylate	8.8 × 10 ⁻⁷ ± 8.7 × 10 ⁻⁸	BLD	BLD	BLD	BLD
30% PEG 4K diacrylate	5.6 × 10 ⁻⁷ ± 7.8 × 10 ⁻⁸	BLD	BLD	BLD	BLD
10% PEG 8K diacrylate	1.9 × 10 ⁻⁶ ± 1.6 × 10 ⁻⁷	BLD	BLD	BLD	BLD
20% PEG 8K diacrylate	9.9 × 10 ⁻⁷ ± 7.4 × 10 ⁻⁸	BLD	BLD	BLD	BLD
30% PEG 8K diacrylate	4.8 × 10 ⁻⁷ ± 2.4 × 10 ⁻⁷	BLD	BLD	BLD	BLD
10% PEG 20K diacrylate	2.3 × 10 ⁻⁶ ± 4.4 × 10 ⁻⁷	6.4 × 10 ⁻⁸ ± 2.6 × 10 ⁻⁸	BLD	BLD	BLD
20% PEG 20K diacrylate	1.3 × 10 ⁻⁶ ± 3.4 × 10 ⁻⁷	2.2 × 10 ⁻⁸ ± 3.1 × 10 ⁻⁸	BLD	BLD	BLD
30% PEG 20K diacrylate	7.3 × 10 ⁻⁷ ± 1.1 × 10 ⁻⁷	5.6 × 10 ⁻⁸ ± 8.3 × 10 ⁻⁸	BLD	BLD	BLD
0.22 μm PVF filter	5.6 × 10 ⁻⁶ ± 2.0 × 10 ⁻⁶	2.0 × 10 ⁻⁶ ± 4.4 × 10 ⁻⁷	–	1.1 × 10 ⁻⁶ ± 1.5 × 10 ⁻⁷	7.8 × 10 ⁻⁷ ± 7.6 × 10 ⁻⁸
Literature values	3.94 × 10 ⁻⁶	1.56 × 10 ⁻⁶	7.76 × 10 ⁻⁷	9.06 × 10 ⁻⁷	6.25 × 10 ⁻⁷

Note: BLD, below the limit of detectability, 5 × 10⁻¹⁰ cm² s⁻¹, of the method employed. Literature values are for diffusion in water.

Table 3

The effect of PEG diacrylate molecular weight and concentration on the network structure of PEG diacrylate hydrogels

	Water content	M _c (g mol ⁻¹)	Mesh size (Å)
10% PEG 2K diacrylate	84% ± 4%	150 ± 85	14 ± 6
20% PEG 2K diacrylate	84% ± 3%	277 ± 91	19 ± 5
30% PEG 2K diacrylate	83% ± 2%	337 ± 53	21 ± 3
10% PEG 4K diacrylate	87% ± 2%	302 ± 121	22 ± 5
20% PEG 4K diacrylate	83% ± 2%	355 ± 98	22 ± 4
30% PEG 4K diacrylate	84% ± 4%	530 ± 235	27 ± 8
10% PEG 8K diacrylate	90% ± 1%	538 ± 119	31 ± 4
20% PEG 8K diacrylate	87% ± 2%	738 ± 185	34 ± 6
30% PEG 8K diacrylate	82% ± 2%	494 ± 92	25 ± 3
10% PEG 20K diacrylate	93% ± 2%	1961 ± 1566	70 ± 38
20% PEG 20K diacrylate	91% ± 1%	1666 ± 413	58 ± 10
30% PEG 20K diacrylate	87% ± 3%	1138 ± 389	42 ± 10

17) For the biological results:

A) Figure 4 concerning antibacterial properties: control experiments are missing (positive and negative control samples which are known to kill bacteria completely or to support bacteria exerting no bactericidal effect) so it is not possible to evaluate the bactericidal efficiency of the non-functionalized and functionalized hydrogels without the two controls (you cannot use the non-functionalized material as a control as we need substrates

with known ability to inhibit or to support bacteria to be used as controls).

Response: We thank the reviewer for this suggestion. We have added LB agar as the negative control that is known to support bacterial growth, and antibiotic-loaded LB agar (1 mg/mL vancomycin for *S. aureus* and 1 mg/mL polymyxin B for *E. coli*) as the positive control that is known to kill bacteria. The updated data with controls were included in new Fig. 5 (Fig. 4 in our original manuscript) of our revised manuscript, as shown below. With these controls, we are confident to draw the conclusion that we obtained potent antibacterial property after functionalizing hydrogels with DbayKY-terminated antibacterial polymers.

Fig.5.

B) Always in the legend of Figure 4, please remember the name of the test to evaluate bacterial killing and the time used for the evaluation.

Response: We thank the reviewer for reminding us on this. In our revision we have added the name “antibacterial test of hydrogels” into the legend of new Fig. 5 (Fig. 4 in our original manuscript) as shown below. We also described the details of the test that “Antibacterial tests were performed by incubating 40 μ L bacterial suspension (2×10^6 CFU/mL) on top of the hydrogels for 2.5 h. n = 3.”

Fig. 5 One-step modification of hydrogels with antibacterial properties. a A schematic diagram for antibacterial test of hydrogels from direct modification with DbayKY terminated antibacterial polymers, Pol-1 and Pol-3. Antibacterial tests were performed by incubating 40 μ L bacterial suspension (2×10^6 CFU/mL) on top of the hydrogels for 2.5 h. n = 3. b. Bacterial killing efficacy of Pol-1 and Pol-3 functionalized hydrogels against *E. coli* and *S. aureus*. c Images of *E. coli* and *S. aureus* colonies on the LB agar plates correlating to surviving bacteria on the surface of bare hydrogel and functionalized hydrogels. LB agar and antibiotic-loaded LB agar (1 mg/mL vancomycin for *S. aureus* and 1 mg/mL polymyxin B for *E. coli*) were used as the negative control and the positive control respectively.

C) Figure 4c: there is a control supporting bacteria: what is it? Please indicate it in the legend and be sure to have inserted it in the exp. Part.

Response: We thank the reviewer for pointing out this. According to the reviewer’s suggestion above, in

our revision we used LB agar as the negative control that is known to support bacterial growth, and antibiotic-loaded LB agar (1 mg/mL vancomycin for *S. aureus* and 1 mg/mL polymyxin B for *E. coli*) as the positive control that is known to kill bacteria. Related information has been updated into the legend and exp. Part (the Methods part) of our revised manuscript.

Fig. 5 One-step modification of hydrogels with antibacterial properties. **a** A schematic diagram for antibacterial test of hydrogels from direct modification with DbayKY terminated antibacterial polymers, Pol-1 and Pol-3. Antibacterial tests were performed by incubating 40 μ L bacterial suspension (2×10^6 CFU/mL) on top of the hydrogels for 2.5 h. n = 3. **b**. Bacterial killing efficacy of Pol-1 and Pol-3 functionalized hydrogels against *E. coli* and *S. aureus*. **c** Images of *E. coli* and *S. aureus* colonies on the LB agar plates correlating to surviving bacteria on the surface of bare hydrogel and functionalized hydrogels. LB agar and antibiotic-loaded LB agar (1 mg/mL vancomycin for *S. aureus* and 1 mg/mL polymyxin B for *E. coli*) were used as the negative control and the positive control respectively. \leftarrow

D) Line 141: you state that hydrogels you used have no pre-organized reactive groups but indeed they may have: ALG has -COOH and PVA has -OH... Maybe I did not correctly understood what you wanted to say. Could you please rephrase?

Response: We thank the reviewer for pointing out this confusing expression in our original manuscript. We want to express that hydrogels do not need to have specific reactive groups for functionalization using DbayKY-terminated molecules. For example, we can functionalize PEG hydrogel directly using DbayKY-terminated molecules, without having to introduce reactive groups to PEG hydrogel first. This strategy can be used to functionalize a wide variety of hydrogels including ALG and PVA. So, we rephrase this part as “to five different classes of hydrogels (PEG, PHEMA, PVA, PSBMA, ALG) without having to incorporate extra reactive groups” in our revision.

E) Figure 5: please in the legend write the test which has been carried out (calcein staining). Also for the *in vitro* cell tests a positive control (Petri Dish) is usually used and should be added.

Response: We thank the reviewer for this suggestion. We have added information about the test which has been carried out in the legend of Fig. 6 of our revised manuscript (the Fig.5 of our original manuscript) that “Cells were treated with LIVE/DEAD staining using calcein AM (green) and ethidium homodimer-1 (red).”

Fig. 6 One-step modification of hydrogels with cell adhesive functions. **a** Fluorescent microscopy images of 3T3 fibroblast cell adhesion to the bare hydrogels and functionalized hydrogels (PHEMA, PEG, PSBMA, PVA, and ALG) with DbayKY terminated Pol-7 after cell seeding for 1 day. **b** Demonstration of cell adhesion on Pol-7 (with adhesive DbayKY) modified PHEMA hydrogel and Pol-11 (without adhesive DbayKY) “modified” PHEMA hydrogel, with polymer modification starting from swelling hydrogels. **c** Demonstration on direct modification of Pol-7 to both the outside and inside of the freeze-dried PHEMA hydrogels, as shown for the inside interfaces by cutting the PHEMA hydrogels from the middle. Cells were treated with LIVE/DEAD staining using calcein AM (green) and ethidium homodimer-1 (red), followed by imaging under a fluorescence microscope. Scale bar: 200 μ m. \leftarrow

We also included a positive control (Petri Dish) for the *in vitro* cell tests. The result was included in the Supplementary Fig. 12, as shown below.

Supplementary Fig. 12.

F) Minor note at line 168: please delete “the” before “how the”

Response: We have deleted “the” before “how the”. The sentence was revised to “We then examined how the DbayKY bearing Pol-7...”.

G) Concerning in vivo studies: again control experiments are missing (e.g. wound without treatment). Please add controls or introduce a reasonable explanation for their missing. Additionally it is not clear how many mice were analyzed in the experimental part.

Response: We thank the reviewer for this suggestion. We repeated the *in vivo* studies with control groups (wound without treatment) as suggested by the reviewer. Six mice were analyzed for each group. We have updated the new *in vivo* data and experimental details into our revised manuscript. The result, as shown below, is consistent with our previous *in vivo* result that Pol-7 functionalized PVA hydrogel promotes wound healing.

We updated our observation in the wound healing study to our revised manuscript that “We found that Pol-7 modification promoted wound healing significantly and the Pol-7 modified PVA hydrogels resulted in almost healed wound after treatment for 10 12 days, which again underpinned the functionalization and application of the easy and efficient one-step modification of hydrogels via the DbayKY adhesive tripeptide (Fig. 7a-c). In contrast, the bare PVA hydrogel treated group and the blank control group (the control without hydrogel treatment) still had obvious unhealed wound after 12 days. Histological evaluation on wound tissue, using hematoxylin and eosin (H&E) staining, showed that Pol-7 modified PVA hydrogel resulted in a complete recover of epidermis; whereas, the bare PVA hydrogel treated group and the blank control group resulted in incomplete wound healing (Fig. 7a-c).”

Fig. 7.

18) Experimental part:

A) Supplier of ALG and PEGDA are missing

Response: We thank the reviewer for reminding us on this. We have added the related information to the Materials part in the supplementary information.

B) In the universal hydrogel modification, the ratio between the used hydrogel material and Pol amount is missing as to allow the readers repeating the experiments. For example you describe Pol concentration but we do not know the Pol solution volume you used and the amount of hydrogel material immersed in that Pol Solution volume. This is important to allow reproduction of results.

Response: We thank the reviewer for pointing out this. We have added related information to the Methods part in the main text of our revised manuscript that “For hydrogel modified by a polymer, the hydrogel (6 mm in diameter) was immersed into the polymer solution (60 μ L) in a 2 mL tube...”.

C) Line 226: you state “until the unmodified polymer was completely removed” but how did you evaluate complete removal?

Response: We thank the reviewer for this question. We confirmed that the unmodified polymer, Pol-7, was completely removed as the last washing water of the hydrogel showed negative result in Kaiser test. We also added the description in the Methods part of our revised manuscript.

D) For XPS: please indicate if you analysed the external surface

Response: We thank the reviewer for reminding us on this. We have added the related information that the XPS characterized the external surface of hydrogel in the Methods part of our revised manuscript.

E) Same comment as above for FTIR analysis

Response: The FTIR analysis was used to characterize the external surface of hydrogel. We have added this information to the Methods part of our revised manuscript.

F) Kaiser test should be “quantitative” as it can be such. Why did you perform it as a qualitative test?

Response: We thank the reviewer for this question. We have attempted to quantify the amine groups in the hydrogels using Kaiser test. However, we found very difficult to obtain reliable calibration curve for quantification. Therefore, we conducted quantifying analysis using more sensitive and reliable methods, rhodamine (Rh) method and fluorecamine method, which have been described in details in our response to the reviewer’s question 16D above.

G) “Cell adhesion” tests should be changed into “Cell viability test” as you used LIVE/DEAD assay.

Response: We thank the reviewer to point out this. Our purpose is to study the cell adhesion function of the modified hydrogels. We used LIVE/DEAD assay kit just to facilitate the fluorescent imaging of surface adhered cells, rather than evaluating the viability. To avoid misunderstanding, we changed the text to that “... for cell adhesion study using LIVE/DEAD staining to facilitate fluorescent imaging.” in our revised manuscript.

H) Lines 240-247: Please specify the procedure for sterilization before cell tests.

Response: We thank the reviewer for reminding us on this and we have added the procedure to our revised manuscript that “Before cell adhesion test, all hydrogels were sterilized by UV light irradiation for 30 min.”

I) Lines 240-247: control experiments are not indicated (positive and negative controls)

Response: We thank the reviewer for pointing out this. We have added related information about cell adhesion study into our revised manuscript that “Bare hydrogels were used as the negative control, and TCPS petri dish were used as the positive control.”

J) Bactericidal tests: positive and negative controls should be inserted.

Response: We thank the reviewer for pointing out this. We have added related information to our revised manuscript that “LB agar and LB agar with antibiotics (1 mg/mL vancomycin for *S. aureus* and 1 mg/mL

polymyxin B for *E. coli*) were used as the positive control and the negative control, respectively. Hydrogels without modification were used as the blank control.”

K) In vivo wound healing: number of mice is missing; control tests are missing.

Response: We thank the reviewer for pointing out this. We have added related information into our revision that “Total 18 wounds of three rats were divided into three groups (n = 6 for each group), including the control group (without hydrogel treatment), bare PVA hydrogel group and Pol-7 modified PVA hydrogel group (Pol-7 PVA).”

We hope that the revised manuscript will prove to be acceptable for publication in *Nature Communications*.

Thank you for your assistance.

Sincerely,

Runhui Liu

Professor of Chemistry and Biomaterials

REVIEWER COMMENTS

Reviewer #1 (Remarks to the Author):

In the previous version of the paper, there was not sufficient comparison to polymers with DOPA in a different environment. Therefore it was not clear that there was anything distinctive about the polymers that were made. In the new version the authors have performed comparison to a range of variants of the DOPA-Lys-DOPA peptide, particularly in the context of AFM analysis of the interaction to a PEG surface. These data suggest some change in adhesion strength according to the presence of lysine and the dibutylamine. I think it is important to clarify this data:

1. spell out whether the average referred to is mean or median
2. perform a suitable statistical test on whether the results are significantly different.

In this kind of analysis, median is usually more appropriate than mean, to reduce the impact of outliers.

It is very hard to perform this sort of AFM reproducibly because of the variable number of polymer chains interacting with the surface. Those performing force spectroscopy to understand protein adhesion now routinely include controls such as DNA strands or I27 domains as fingerprints to validate that single polymer chains are being analyzed. Traces indicating the extension of multiple protein chains are then discarded from the analysis. It would not be helpful to conclude that a polymer interacts stronger with a surface, if the data may reflect only that the polymer ended up attaching at a higher density on the cantilever.

My other comments have been suitably addressed.

Reviewer #2 (Remarks to the Author):

In the revised manuscript the authors have added several experiments that were lacking in the original manuscript. This has improved the overall quality of the manuscript, whereas the original draft had several basic shortcomings. One of the main comments was the role played by the individual amino acids and the DOPA units in the anchoring fragment. Force measurements using AFM have been used to show an increased adhesion when the amino acids are present, along with DOPA units, and that the sequence is not important. Other sets of experiments involve demonstration of the extent of functionalization, which was also lacking in the original version. Although these experiments did not add any novelty they made the work more complete. While these corrections have improved the manuscript, unfortunately, the underlying scientific reason behind this adhesiveness is still not at all clear. PEG-based hydrogels that are inherently anti-biofouling in this study show quite a bit of affinity for the adhesive construct. It is hard to rule out if interactions between various adhesive anchors lead to such attachment. Furthermore, any level of oxidation of DOPA units will lead to many side reactions, which even in minimal amounts will lead to the entanglement of polymers within hydrogels. Thus, it is not clear how this work is so different from other approaches where DOPA-based polymers have been synthesized and used for various surface modifications. Also, in most of those studies, the reason behind the affinity of the anchoring groups for the underlying substrate is more clear than as presented here. It does not seem likely that this study will bring any new directions in the area of functional hydrogels. In light of limited novelty, and lack of clarity of the underlying adhesion phenomenon, publication of manuscript is not recommended.

Reviewer #3 (Remarks to the Author):

The paper has been improved following reviewers' suggestions, however some remarks remain and a few parts are still suboptimal.

Concerning the need for further tests:

1) Stability tests should be performed at least up to 7 days (e.g. 1 day, 3 days, 7 days)
2) Although I did not comment on that previously (as I had many other basic points to arise such as the lack of controls), cell tests should be performed up to 7 days to confirm durability of the functionality (e.g. 1 day, 3 days, 7 days). It would be also interesting to give immunostaining of DAPI, actin filaments and focal adhesion proteins (e.g. vinculin) at the same time-points in addition to live/dead assay.

3) Similarly antibacterial tests should be performed up to 7 days (1, 3, 7 days).

Without such more in-depth functional characterisation, the potentialities of the functional approach are not evident for short-term applications. Of course longer in vitro cell and antibacterial experiments could further improve the manuscript.

Additionally the following parameters should be included and discussed:

1) Efficiency of functionalization: (peptide bound to hydrogel/binding peptide used initially) x 100
2) Degree of functionalisation: (peptide bound to hydrogel/wet weight of initial hydrogel) x 100 and (binding peptide bound to hydrogel/dry weight of initial hydrogel) x 100 - the latter to account for differences among hydrogel concentrations.
3) Comments on hydrogel permeability present in the answer to reviewers, allowing to understand the inability of molecules linked to peptides to penetrate the hydrogel structure. As you are able to stain the binding peptide, I think you may also visualize its penetration in the hydrogel structure, e.g. by confocal microscopy analysis. This could help in understand any penetration as a function of hydrogel types.

Abstract

The abstract in the current form does not well recapitulate the manuscript content and I propose the following suggestions:

Lines 18-19: Functionalisation of hydrogels typically occurs before gelation (by the preparation of functional molecules) or during gelation. Please rephrase accordingly.

Lines 21-23: should be "wet hydrogels using molecules provided with adhesive etc..." as this is what you are proposing in the paper.

Lines 24-27: Line 24, after "tripeptide" you may start a new sentence "Such functional molecules enable direct modification of wet hydrogels...etc". Line 27: should be "...wet hydrogels to provide them with diverse...".

Other details could be provided in the abstract (e.g. range of functionalisation degree which can be obtained, stability of the proposed functionalisation, penetration of functionalization within the hydrogel structure, etc).

Introduction

Line 34: should be "hydrophilic polymers form highly hydrated three-dimensional networks and many of them provide low fouling properties, hence require chemical etc". Indeed, some hydrophilic polymers forming hydrogels are not inert. For example, you also listed chitosan which is a well-known bioactive polymer. Additionally, "inert" refers to the lack of any interaction with the biological environment, while physical crosslinked hydrogels can be gradually dissolved in biological fluids in vivo. An inert material does not undergo any physicochemical change in a biological environment. So I would suggest to pay attention to the use of "inert" in the paper.

Lines 36-39: Figure 1a could be cited.

Line 38: as a second option, I would consider "modification during gelation" rather than post-gelation and proper references should be considered.

Line 43: after "functionalization step" a reference is needed.

Line 45: I would change in " which limits the flexibility etc"

Line 51: should be "to realize surface modifications"

Line 59: should be "designed"

Results

Line 103: which is the meaning of F-X curve?

Lines 99-112: Here initially authors compare the adhesion force of a dipeptide YY, tripeptide YKY

and their sequence DbA-YKY however they do not comment on the possible effect of sequence length on adhesion force. No other comments are present explaining the differences found. In lines 107-108, they state that K and DbA are important for adhesion of their DbA-YKY sequence. Then they tested KYK and showed slightly lower adhesion to PEG than YKY (why?) and much lower than for DbA-YKY (why?). On the other hand KKYY has an adhesion strength which appears close to that of DbA-YKY (246 vs 251 pN) and here they state that the sequence in DbA-YKY is not important for the adhesion. After this paragraph, I am confused: initially the authors state that lysine and DbA are important for adhesion (line 104), then it seems the sequence is not important. What is it important for the adhesion? Although in lines 148-154 there is a general explanation, here you should specify the reasons for the different adhesion strengths to PEG substrate. On the other hand, I found very informative your answer to Reviewer 1: please try to give here similar information to those provided in the answer to Reviewer 1. Importantly, it appears that you use the term "sequence" which can be misunderstood while I think you could substitute it with comments on the "order" or "distribution" of amino acids in the peptide sequence you test. Please also consider that the number of peptides also affect interactions.

Figure 3: please correct the letters a-g (c has been shifted and g is missing). In the legend please shift the comment on N values at the end of the legend.

Line 170: is the % modification expressed respect to polymer forming the hydrogel after hydrogel drying or respect to the weight of wet hydrogel? I would suggest to present both data, considering the differences in concentration (water content). However, in the experimental part you should clearly report the formula for calculation of the % of modification. Please detail in the exp. Part.

Line 189: I would suggest to express the functionalization also respect to dry weight of hydrogel (after drying hydrogels) = [amount of functionalizing molecules (e.g. weight) in final hydrogel/weight of initial polymer hydrogel (in dry state)] x 100

Lines 192-196: stability tests should be performed at least for 7 days at different time points to assess stability of the functionalization.

In the main manuscript, discussion would deserve improvement and rearrangement, by including comments on the following points (in some cases, they are already present but need better explanation and integration each other)

1) Please motivate the choice of pH for functionalization and pH effect (together with chosen functionalization time) on the self-polymerisation of the peptide and on functionalization efficiency. You introduced such comments in the answer to reviewers letter but they are missing in the main manuscript.

2) Please stress on the aim of the approach, being the synthesis of DbA-DOPA-Lys-DOPA-terminated molecules for the preparation of functionalization polymers, which can then be exploited for wet hydrogel functionalization according to a versatile approach. In doing that, please underline the adhesion mechanisms by the binding peptide and that this could change the functionalization degree, with examples on samples you tested. Also please underline how it is possible to change the functionalization degree of hydrogel and within which range it can be done. Please refer the functionalization degree respect both to the wet hydrogel and the dried hydrogel in order to properly comment the results based on material chemistry, hydrogel polymer amount, etc.

Please clearly state that the functionalization typically cannot penetrate the hydrogel network, but it is proposed as a surface functionalization of the exposed hydrogel surface. Also please underline that bulk functionalization of hydrogel structure is possible by using lyophilized hydrogels.

3) You should also comment on the functionalization stability as a function of incubation time (up to at least 7 days), please. I appreciate the addition of stability studies up to 24 h and to 2 h (with enzyme), however these times are very brief: what does it happen later? You can also simply refer to stability in PBS as a function of time (at least 7 days), please.

4) Additionally, you could comment on the efficiency of the functionalization (I have asked it in my previous comments but maybe I was not clear). In your work, you use a certain functionalizing molecule amount: a part binds to the hydrogel surface. Is it possible to measure (the ratio between the functionalizing molecule which has been linked to the surface to the overall functionalizing molecule initially used) x 100 (efficiency of functionalisation) ? In case please, add it also in the experimental part, mentioning the formula.

5) Please comment on prospective exploitation of your functionalizing strategy and add recommendations/suggestions for its optimisation on different hydrogel materials. In the answer to reviewers you introduced the synthesis of a simple binding peptides which could be easily linked

to commercial molecules: please comment also on the use of this binding peptide

6) The antimicrobial and cell adhesive functionality it evaluated only at one timepoint (24 h). As it is not clear whether the functionalization is stable and preserves its biological function as a function of time, I would recommend to add 2 additional timepoints up to 7 days for antimicrobial and cell adhesion tests (in parallel to the additional timepoints in stability studies)

Methods

Lines 287-293: please be sure the codes have been previously defined

Line 317: Please introduce what you mean here as polymer. You are not introducing your Pol materials before explaining functionalization of hydrogels with them. Please refer to Supplementary as to introduce them.

Line 324: indicate the type of freeze drier

Line 327: Kaiser test should be explained here or in supplementary material. Proper reference to Supplementary material should be present.

Line 328: cell adhesion studies are not explained here. I suggest to describe them later in the specific par. on in vitro cell tests, saying there that you also tested these types of samples with cells.

Line 332: should be "shown" not showed

AFM force spectroscopy: indicate how many samples you tested for each Pol for statistical purposes and which data you reported in the paper.

XPS analysis: was the sample dried before the analysis? If yes, please indicate it

FITR analysis: was the sample dried before the analysis? If yes, please indicate it

Line 360: Kaiser test should be better explained. It has already been introduced in the text so it should be explained at the first introduction in the exp.

Line 366: "were modified to hydrogel" is wrong expression

Lines 379-381: it is not clear how you referred data to hydrogel material. Please indicate the equation you used to give the modification % and state clearly whether you made reference to dried hydrogel. If not, please also add calculation respect to dried hydrogel

Lines 385-386 and 389: What is the PB buffer?

Line 395: indicate the equation used to define the modification amount. Is it expressed respect to dried hydrogel?

Line 410-415: this is not an experimental part, this is a result and should be integrated into the main manuscript.

Line 419: should be "were"

Line 433: parenthesis is not needed

Line 438: should be "were"

Cell adhesion: please notice that you also introduced cell adhesion tests before. Hence you could describe the cell analysis on cut hydrogels here, too.

In vivo trials: indicate the microscope used for the analysis

Please control that all the reagent characteristics (supplier and other key properties) are present in the Supplementary materials or in the Experimental part of the main paper. Importantly, guide the reader in the Exp part for reference to the Supplementary material for additional details. This is missing.

Supplementary materials

Some minor mistakes:

Lines 74-76, 88-89, 124-125, 147-149, 199-200, 211-213 248-250, 260-262, 271-272, 417-419, 476-478, 487-488, 506-508: please rephrase as that the subject is reported before the verb (and not the opposite as in the current form)

Supplementary Figure 12 Legend: Please indicate the culture time (24 h)

Supplementary Figure 5 and 6, (a): please remove "diagram" as it is a "schematic representation"

Line 462: please correct into "are conducted"

Line 444: please correct "after coupling complete", e.g. into "after completion of the coupling reaction"

Finally I have noticed that the reference to Supplementary material is "random", i.e. it does not

follow the order in which Supplementary materials are presented. This should be checked and adjusted.

Point-by-point response to the reviewers' comments

Reviewer #1 (Remarks to the Author):

In the previous version of the paper, there was not sufficient comparison to polymers with DOPA in a different environment. Therefore it was not clear that there was anything distinctive about the polymers that were made. In the new version the authors have performed comparison to a range of variants of the DOPA-Lys-DOPA peptide, particularly in the context of AFM analysis of the interaction to a PEG surface. These data suggest some change in adhesion strength according to the presence of lysine and the dibutylamine. I think it is important to clarify this data:

1. spell out whether the average referred to is mean or median

Response: We thank the reviewer for pointing out this. In our previous revision the average referred to mean. In this revision, we use median instead of mean to evaluate the average force, as also suggested by the reviewer in the question below. Using the median to evaluate the average force has also been reported in the precedent literature (*Angew. Chem. Int. Ed.* **2020**, 59, 16616). Therefore, in this revision we use the median to describe the results of the median values of rupture force for DbayKY, YY, YKY, KYK and KKYY (Supplementary Table 2 in our revised manuscript, as shown below). The conclusion based on this force study is not changed after we use median to evaluate the average force.

Supplementary Table 2. Rupture force of the adhesive moieties measured using SMFS.

Adhesive moiety	DbayKY	YY	YKY	KYK	KKYY
Median (pN)	223.0	81.7	163.5	172.2	202.8
Mean (pN)	251.9	97.2	210.0	203.3	246.0
P [#]	-	<0.001	<0.001	<0.001	0.63

[#]Significant difference analysis between DbayKY and other adhesive moieties in the table was determined by two-tailed t-test.

2. perform a suitable statistical test on whether the results are significantly different.

Response: We thank the reviewer for this kind suggestion. Analysis of the significant difference between adhesive moieties (YY, YKY, KYK or KKYY) and DbayKY was determined by two-tailed t-test (Supplementary Table 2 in our revised manuscript, as shown above). The adhesion strength of YY, YKY or KYK to PEG hydrogel was significantly lower than that of DbayKY, while the adhesion strength of KKYY is no significantly different from that of DbayKY. We added this analysis into our revision.

In this kind of analysis, median is usually more appropriate than mean, to reduce the impact of outliers.

Response: We agree with the reviewer's comment that median is more appropriate than mean in AFM force analysis, as is used in the precedent literature (*Angew. Chem. Int. Ed.* **2020**, 59, 16616). In this revision, we use the median, instead of the mean, to evaluate the force, as described above in our response to the first question from the reviewer.

It is very hard to perform this sort of AFM reproducibly because of the variable number of polymer chains interacting with the surface. Those performing force spectroscopy to understand protein adhesion now routinely include controls such as DNA strands or I27 domains as fingerprints to validate that single polymer chains are being analyzed. Traces indicating the extension of multiple protein chains are then discarded from the analysis. It would not be helpful to conclude that a polymer interacts stronger with a surface, if the data may reflect only that the polymer ended up attaching at a higher density on the cantilever.

Response: We thank the reviewer for the comment that “*It is very hard to perform this sort of AFM reproducibly because of the variable number of polymer chains interacting with the surface. Those performing force spectroscopy to understand protein adhesion now routinely include controls such as DNA strands or I27 domains as fingerprints to validate that single polymer chains are being analyzed.*” We agree with the reviewer that researchers need to pay attention to this. It is said in the precedent literature (*Front. Mol. Biosci.* **2020**, 7, 85) that “A drawback of AFM-SMFS on receptor-ligand interactions is that valid single-molecule interactions are difficult to discriminate from non-specific interactions or multiple interactions occurring in parallel, and these fingerprint domains have been used to screen for single receptor-ligand complex unbinding events from large datasets.” In our study, our research object is not a specific receptor-ligand binding, such as the antibody-antigen and biotin-streptavidin interactions. We only test the direct rupture force between the small adhesion moiety (such as DbaYKY) and the hydrogel. This force itself is not from a receptor-ligand-type bond, so it is not involved in protein folding/unfolding rates or folding intermediate states in polyproteins. In addition, in our data analysis, traces indicating the extension of multiple adhesive chains are already discarded as the reviewer’s comment. The data with multiple rupture events in the F-D curve are filtered out, as a similar method in precedent literature (*Nat. Commun.* **2020**, 11, 3895). Therefore, using SMFS approach to study the rupture force of catechol moieties to the surface, fingerprint domains are not necessarily used in published studies (*Angew. Chem. Int. Ed.* **2020**, 59, 16616; *Nat. Commun.* **2020**, 11, 3895; *Angew. Chem. Int. Ed.* **2019**, 58, 1077. *Nanoscale* **2016**, 8, 15309. *PNAS*. **2006**, 103, 12999).

We agree with the reviewer’s comment that “*It would not be helpful to conclude that a polymer interacts stronger with a surface, if the data may reflect only that the polymer ended up attaching at a higher density on the cantilever.*” We have considered this when designing the experiment to avoid variable number of polymer chains that interact with the surface by utilizing two solutions, as described below.

1) These cantilevers were immersed in a solution of 1:10 mixture of NHS-PEG-SH (5000 Da) and NHS-PEG-OMe (2000 Da). Only less than 10% PEG chains, bearing terminal thiol group, can further react with maleimide-terminated adhesion moieties. This 1:10 ratio of NHS-PEG-SH vs. NHS-PEG-OMe was used to control the binding density of functional PEG and to reduce multiple interactions in the force spectroscopy measurements (*Nat. Commun.* **2020**, 11, 3895). We also added relevant discussion for choosing this ratio in our revision. The force measurement results show that most of the contacts are negligible because only non-adhesive PEG chains contact the hydrogel (Supplementary Figure 2a, as shown below), which greatly reduces the probability of contact between multiple adhesion moieties and the hydrogel.

2) In our data analysis, the expected single chain adhesion traces are analyzed (Figure 3b, as shown below); indeed, a small ratio of traces indicating the extension of multiple adhesive chains are already discarded (Supplementary Figure 2b, as shown below) as the reviewer’s comment that “*traces indicating the extension of multiple protein chains are then discarded from the analysis*”.

We also added relevant description into the Results part of our revision that “Rupture events that the DbaYKY is not in contact with PEG hydrogel and a few events that DbaYKY has multiple interactions with PEG hydrogel were discarded in the data analysis”.

Regarding the reviewer’s comment on measurement reproducibility, we found that some adhesion groups such as DbaYKY have a wide distribution of force against hydrogel because there are more than one catechol and amino groups on the molecule, leading to complex interaction forces when the molecules

contact the hydrogel, which is similar to some of the wide distribution adhesion force result in the study by Prof. Messersmith et al. (*Nat. Commun.* **2020**, *11*, 3895). Therefore, to address concerns on reproducibility, we tested a larger number of contact events (N value) to accurately measure the force distribution. In addition, the SMFS experiment was repeated twice at different time, and the results of the two experiments are comparable (Supplementary Table. 3 in our revised manuscript, as shown below).

Supplementary Figure 2. Representative discarded data of F-D curves that the DbayKY is not in contact with PEG hydrogel (a) and that multiple chains of DbayKY interact with PEG hydrogel (b).

Fig. 3b. Representative F-D curves and rupture force distribution are shown for DbayKY modified tips.

Supplementary Table 3. SMFS results of samples from two different batches.

		Rupture force (pN)				
		DbayKY	YY	YKY	KYK	KKYY
Median	Batch 1	223.0	81.7	163.5	172.2	202.8
	Batch 2	240.6	58.2	151.4	167.3	227.8
Mean	Batch 1	251.9	97.2	210.0	203.3	246.0
	Batch 2	260.2	70.8	171.9	192.1	272.8

My other comments have been suitably addressed.

Response: We thank the reviewer for these kind suggestions to help us improve the quality of our manuscript.

Reviewer #2 (Remarks to the Author):

In the revised manuscript the authors have added several experiments that were lacking in the original manuscript. This has improved the overall quality of the manuscript, whereas the original draft had several basic shortcomings. One of the main comments was the role played by the individual amino acids and the DOPA units in the anchoring fragment. Force measurements using AFM have been used to show an increased adhesion when the amino acids are present, along with DOPA units, and that the sequence is not important. Other sets of experiments involve demonstration of the extent of functionalization, which was also lacking in the original version. Although these experiments did not add any novelty they made the work more complete. While these corrections have improved the manuscript, unfortunately, the underlying scientific reason behind this adhesiveness is still not at all clear. PEG-based hydrogels that are inherently anti-biofouling in this study show quite a bit of affinity for the adhesive construct. It is hard to rule out if interactions between various adhesive anchors lead to such attachment. Furthermore, any level of oxidation of DOPA units will lead to many side reactions, which even in minimal amounts will lead to the entanglement of polymers within hydrogels. Thus, it is not clear how this work is so different from other approaches where DOPA-based polymers have been synthesized and used for various surface modifications. Also, in most of those studies, the reason behind the affinity of the anchoring groups for the underlying substrate is more clear than as presented here. It does not seem likely that this study will bring any new directions in the area of functional hydrogels. In light of limited novelty, and lack of clarity of the underlying adhesion phenomenon, publication of manuscript is not recommended.

Response: We thank the reviewer for the recognition of our previous revision and these comments, which inspired us to further improve our manuscript, as also suggested by other reviewers, and strength the novelty of our work. Hydrogels are extensively used in biomaterials and tissue engineering. In most of the cases, functionalization of the hydrogels, such as incorporation of functional peptides, is highly required. Functionalization of hydrogels, such as PEG hydrogels, can be very difficult and tedious via multiple steps. To address this long lasting challenge, our study focus on the **easy functionalization of diverse wet hydrogels through a simple one-step strategy**. We found surprisingly that the simple one-step functionalization of diverse hydrogels, including PEG hydrogels, can be achieved by using an adhesive DbYKY tripeptide. One of the superior advantages of such a short adhesive tripeptide lies in its' easy solution synthesis in a large amount, and easy conjugation to various molecules and polymers.

About the adhesion mechanism, the DbYKY terminated polymers to hydrogels can be attributed to multiple interactions including hydrogen bonding, cation- π stacking and charge interactions. The catechol groups within DbYKY can utilize two neighboring hydroxyl groups as the donors/acceptors to form hydrogen bonding to hydrogels. The positively charged amine groups can form cation- π interactions with the aromatic rings within DOPA. Therefore, the presence and the number K and Y are important for adhesion. The order of rupture force in our study is DbYKY \approx KKYY $>$ YKY \approx KYK $>$ YY, which implies that when the sequence length and the number of K and Y increase, the overall adhesion force and the functionalization degree will increase. In addition, the DbYKY terminated polymers have positively charged amine groups, therefore, can interact electrostatically with hydrogels bearing negatively charged groups, such as PSBMA and ALG. Beside the adhesion mechanistic study, the contribution of our work lies in a simple one-step hydrogel functionalization of diverse hydrogels using an adhesive DbYKY tripeptide that can be synthesized in solution easily for large quantity.

We agree with the reviewer's comment that PEG-based hydrogels are inherently anti-biofouling because of the hydration effect. However, hydration effect of PEG doesn't necessarily prevent catechol from interacting with PEG chains. In addition, the purpose of our hydrogel functionalization strategy is to modify the hydrogels and enable biological functions, rather than restricting to single molecular chain modification to hydrogels. So we do not need to rule out if interactions between various adhesive anchors lead to such attachment. The direct interaction of DbYKY tripeptide to PEG hydrogel and the probable interaction

between various adhesive anchors together realize the functionalization of the hydrogel. Polymers without adhesion groups cannot obviously entangle with the hydrogel, as shown in the ATR-FTIR result (Fig. 4 in our revised manuscript). Because the functional polymers only have a single adhesive tripeptide at one of the chain terminal, the interaction between various adhesive anchors, if happens as speculated by the reviewer, only lead to an increased polymer length, but not entanglement as just explained above.

In short, the reviewer's comment inspired us to strength the novelty and contribution of this work in two aspects: 1) easy functionalization of diverse wet hydrogels through a simple one-step strategy, using an adhesive DbYKY tripeptide. 2) easy solution synthesis in a large amount, and easy conjugation of the adhesive tripeptide to various molecules and polymers. Hydrogels are widely studied and used in recent years, and biomedical hydrogels need to be endowed with antibacterial or tissue engineering functions when used in various applications. Our study provides a one-step simple strategy for functionalization of hydrogels and expands the study and application of functional hydrogels. We added relevant discussion into our revised manuscript.

Reviewer #3 (Remarks to the Author):

The paper has been improved following reviewer's suggestions, however some remarks remain and a few parts are still suboptimal.

Response: We thank the reviewer for the comment and detailed questions/suggestions below to help us improve our manuscript.

Concerning the need for further tests:

1) Stability tests should be performed at least up to 7 days (e.g. 1 day, 3 days, 7 days)

Response: We thank the reviewer for this suggestion. We have added the stability tests of functionalized hydrogels in PBS or proteinase K for up to 7 days. The results show that the functionalized hydrogels can be stable either in PBS or proteinase K for 1 day, 3 days and 7 days (Supplementary Figure 9 of our revised manuscript, as shown below).

In our revised manuscript, we modified the relevant description to "Using the DbYKY-OEG8-Rh model, we also analyzed the stability of the modification under shaking in PBS at room temperature for 1 day, 3 days, and 7 days, or incubation with proteinase K at 37 °C for 2 hours, 1 day, 3 days, and 7 days. Little difference in fluorescence intensity of the hydrogels was observed after either treatment for up to 7 days (Supplementary Fig. 9), indicating that the hydrogel modification has favorable stability."

Supplementary Figure 9. Stability of DbYKY-OEG8-Rh modified PEG hydrogels in PBS under staking for 1 day, 3 days and 7 days or in protease K (PK) solution at 37 °C for 2 h, 1 day, 3 days and 7 days. n = 6, mean values \pm s.d. Statistical analysis: two-tailed t-test. ns: not significant.

2) Although I did not comment on that previously (as I had many other basic points to arise such as the lack of controls), cell tests should be performed up to 7 days to confirm durability of the functionality (e.g. 1 day, 3 days, 7 days). It would be also interesting to give immunostaining of DAPI, actin filaments and focal adhesion proteins (e.g. vinculin) at the same time-points in addition to live/dead assay.

Response: We thank the reviewer for this suggestion. We did experiments of cell adhesion on Pol-7 modified PEG hydrogels, using bare PEG hydrogels as the control. As requested by the reviewer, we performed immunostaining of DAPI (blue), actin filaments (green) and vinculin (red) after fibroblast cells culture for 1 day, 3 days and 7 days. The bare PEG hydrogel does not support fibroblast adhesion; in sharp contrast, fibroblast cells adhere well to the Pol-7 modified hydrogel and proliferate to have gradually increased cell density from day 1 to day 7. It's noteworthy that cell adhere well to the Pol-7 modified

hydrogel after 7 days and keep healthy morphology (Fig. 6b in our revised manuscript, as shown below). The 7 days cell adhesion result confirms the durability of the functionality, which echoes the result of the stability test in our response to the first question above from the third reviewer.

Fig. 6b. Fluorescence confocal microscope images (green, actin; red, vinculin; blue, nucleus) of NIH 3T3 fibroblast cells on bare PEG hydrogels and Pol-7 modified PEG hydrogels after cell seeding for 1 day, 3 days and 7 days. Scale bar: 50 μm .

3) Similarly antibacterial tests should be performed up to 7 days (1, 3, 7 days).

Without such more in-depth functional characterisation, the potentialities of the functional approach are not evident for short-term applications. Of course longer in vitro cell and antibacterial experiments could further improve the manuscript.

Response: We thank the reviewer for this suggestion. In the original manuscript, we showed that Pol-1 or Pol-3 modified PEG or PVA hydrogels have strong antibacterial activity. So, we followed the suggestion from the reviewer and did extra tests on the antibacterial activity of polymer-modified hydrogel surfaces. After hydrogels were incubated with the bacteria solution for 1 day, 3 days and 7 days, respectively, we found that all bacteria were killed. We then increased the challenge on the antibacterial activity of hydrogel surfaces and added bacteria again to these hydrogels. After hydrogels were incubated with the bacteria solution for another 2.5 hours, all Pol-1 or Pol-3 modified PEG or PVA hydrogels showed >95% killing efficiency, with most cases showing >99% killing (Fig. 5d in our revised manuscript, as shown below). This study also confirmed the durability of the functionality, which echoes the results of the stability test and cell adhesion test mentioned above.

Fig. 5d. Durability test on the bacterial killing efficacy of Pol-1 and Pol-3 functionalized hydrogels against *E. coli* and *S. aureus* ($n = 3$). The antibacterial tests were performed using hydrogels that were initially incubated with bacterial solution for 1 day, 3 days and 7 days.

Additionally the following parameters should be included and discussed:

1) Efficiency of functionalization: (peptide bound to hydrogel/binding peptide used initially) x 100

Response: We thank the reviewer for this suggestion. We calculated the efficiency of functionalization in our revision. The efficiency of Pol-7 modified to PVA hydrogels and ALG hydrogels were ~0.360% and

~1.97% (w/w, Pol-7 bound to hydrogel/Pol-7 used initially), respectively (Supplementary Fig. 6d in our revised manuscript, as shown below). The efficiency of DbayKY-OEG8-Rh modified to PEG hydrogel, PHEMA hydrogel, PSBMA hydrogel, PVA hydrogel and ALG hydrogel was ~3.92%, ~54.6%, ~20.6%, ~1.10% and ~6.44% (w/w, DbayKY-OEG8-Rh bound to hydrogel/DbayKY-OEG8-Rh used initially), respectively (Supplementary Fig. 8e in our revised manuscript, as shown below).

We also included discussions in our revised manuscript. The observed difference in the efficiency of functionalization on different hydrogels may come from many aspects, including the large difference in water content of different hydrogels, the difference in crosslinking methods (chemical crosslinking for PEG, PHEMA and PSBMA, and physical crosslinking for PVA and ALG), and the difference in charge (zwitterionic for PSBMA, negatively charged for ALG, and uncharged for others. For example, a higher amount of polymer was modified to the ALG hydrogel than the PVA hydrogel likely because the ALG hydrogel has a large amount of negatively charged carboxyl groups, which will electrostatically attract Pol-7 that has multiple amine groups.

Supplementary Figure 6d. Efficiency of functionalization (%) calculated by the formula: (peptide bound to hydrogel/binding peptide used initially) × 100

Supplementary Figure 8e. Efficiency of functionalization (%) calculated by the formula: (peptide bound to hydrogel/binding peptide used initially) × 100

2) Degree of functionalisation: (peptide bound to hydrogel/wet weight of initial hydrogel)x100 and (binding peptide bound to hydrogel/dry weight of initial hydrogel)x100 - the latter to account for differences among hydrogel concentrations.

Response: We thank the reviewer for this suggestion. We calculated the degree of functionalization including (peptide bound to hydrogel/wet weight of initial hydrogel) \times 100 and (peptide bound to hydrogel/dry weight of initial hydrogel) \times 100. **Modification amount to dry weight hydrogel (%)** is calculated by the formula: (weight of molecules bound to hydrogel/dry weight of initial hydrogel) \times 100. **Modification amount to wet weight hydrogel (%)** is calculated by the formula: (weight of molecules bound to hydrogel/wet weight of initial hydrogel) \times 100.

Using the dry weight of hydrogel for calculation, the modification amount of Pol-7 to PVA hydrogels and ALG hydrogels was \sim 0.0515 % and \sim 0.203% (w/w, weight of Pol-7 bound to hydrogel/dry weight of initial hydrogel), respectively (Supplementary Fig. 6b, as shown below). Using the wet weight of hydrogel for calculation, the modification amount of Pol-7 to PVA hydrogels and ALG hydrogels was \sim 0.003% and \sim 0.017% (w/w, weight of Pol-7 bound to hydrogel/wet weight of initial hydrogel), respectively (Supplementary Fig. 6c, as shown below).

Using the dry weight of hydrogel for calculation, the modification amount of DbaYKY-OEG8-Rh to PEG hydrogel, PHEMA hydrogel, PSBMA hydrogel, PVA hydrogel and ALG hydrogel was \sim 0.051%, \sim 0.104%, \sim 0.046%, \sim 0.020% and \sim 0.083% (w/w, weight of DbaYKY-OEG8-Rh bound to hydrogel/dry weight of initial hydrogel), respectively (Supplementary Fig. 8c, as shown below). Using the wet weight of hydrogel for calculation, the modification amount of DbaYKY-OEG8-Rh to PEG hydrogel, PHEMA hydrogel, PSBMA hydrogel, PVA hydrogel and ALG hydrogel was \sim 0.004%, \sim 0.058%, \sim 0.022%, \sim 0.001% and \sim 0.007% (w/w, weight of DbaYKY-OEG8-Rh bound to hydrogel/wet weight of initial hydrogel), respectively (Supplementary Fig. 8d, as shown below).

Supplementary Figure 6b-c. Quantitative analysis of Pol-7 on PVA and ALG hydrogels, including modification amount to dry weight hydrogel (%) that is calculated by the formula: (weight of molecules bound to hydrogel/dry weight of initial hydrogel) \times 100 (b), modification amount to wet weight hydrogel (%) that is calculated by the formula: (weight of molecules bound to hydrogel/wet weight of initial hydrogel) \times 100 (c).

Supplementary Figure 8c-d. (c-d) Quantitative analysis of DbayKY-OEG8-Rh (0.5 mg/mL) on various hydrogels, including modification amount to dry weight hydrogel (%) that is calculated by the formula: (weight of molecules bound to hydrogel/dry weight of initial hydrogel) \times 100 (c), and modification amount to wet weight hydrogel (%) that is calculated by the formula: (weight of molecules bound to hydrogel/wet weight of initial hydrogel) \times 100 (d).

3) Comments on hydrogel permeability present in the answer to reviewers, allowing to understand the inability of molecules linked to peptides to penetrate the hydrogel structure. As you are able to stain the binding peptide, I think you may also visualize its penetration in the hydrogel structure, e.g. by confocal microscopy analysis. This could help in understand any penetration as a function of hydrogel types.

Response: We thank the reviewer for this suggestion. In order to visualize the permeability of the hydrogel in normal and lyophilized hydrogels, we used rhodamine (Rh)-linked fluorescent molecule DbayKY-OEG8-Rh to functionalize PHEMA hydrogels that are processed with or without the freeze-drying step. The confocal images of hydrogel surface and hydrogel cross section showed that DbayKY-OEG8-Rh can only functionalize the surface of PHEMA hydrogel, and cannot penetrate the PHEMA hydrogel structure, because the cross-section confocal image has no fluorescence. In contrast, both surface and cross section of freeze-dried PHEMA hydrogels showed strong fluorescence, indicating that DbayKY-OEG8-Rh can penetrate the interior of the freeze-dried PHEMA hydrogel (Supplementary Figure 15 in our revised manuscript, as shown below). We also added relevant discussion in our revised manuscript.

Supplementary Figure 15. Confocal images of surface and z-axis cross section of DbayKY-OEG8-Rh functionalized PHEMA hydrogels that are processed with or without the freeze-drying step.

Abstract

The abstract in the current form does not well recapitulate the manuscript content and I propose the following suggestions:

Lines 18-19: Functionalisation of hydrogels typically occurs before gelation (by the preparation of functional molecules) or during gelation. Please rephrase accordingly.

Response: We thank the reviewer for pointing out this. We have changed "...incorporation of functional molecules either during gelation or post during gelation..." to "...incorporation of functional molecules either before gelation or during gelation..." in our revision.

Lines 21-23: should be "wet hydrogels using molecules provided with adhesive etc..." as this is what you are proposing in the paper.

Response: We agree with the reviewer on this. We have changed this sentence to "...wet hydrogels using molecules provided with an adhesive dibutylamine-DOPA-lysine-DOPA tripeptide" in our revision.

Lines 24-27: Line 24, after "tripeptide" you may start a new sentence "Such functional molecules enable direct modification of wet hydrogels...etc". Line 27: should be "...wet hydrogels to provide them with diverse...".

Response: We thank the reviewer for these kind suggestions. After "tripeptide" we have added a new sentence that "Such functional molecules enable direct modification of wet hydrogels...". We also followed the suggestion and changed "...wet hydrogels with diverse..." to "...wet hydrogels to provide them with diverse..."

Other details could be provided in the abstract (e.g. range of functionalisation degree which can be obtained, stability of the proposed functionalisation, penetration of functionalization within the hydrogel structure, etc).

Response: We thank the reviewer for the kind suggestion. Due to the total limit of the abstract within 150 words, we added brief details into the abstract that "The strategy has a tunable functionalization degree and a stable attachment of functional molecules."

Introduction

Line 34: should be "hydrophilic polymers form highly hydrated three-dimensional networks and many of them provide low fouling properties, hence require chemical etc". Indeed, some hydrophilic polymers forming hydrogels are not inert. For example, you also listed chitosan which is a well-known bioactive polymer. Additionally, "inert" refers to the lack of any interaction with the biological environment, while physical crosslinked hydrogels can be gradually dissolved in biological fluids in vivo. An inert material does not undergo any physicochemical change in a biological environment. So I would suggest to pay attention to the use of "inert" in the paper.

Response: We thank the reviewer for the kind suggestion. We have changed "Hydrophilic polymers provide highly hydrated and inert three-dimensional networks, which provide low fouling favorable for biological recognition, however, require..." to "Hydrophilic polymers form highly hydrated three-dimensional networks and many of them provide low fouling properties, hence require chemical..." as

suggested by the reviewer.

We also agree with the reviewers' comments on the use of word "inert". We have removed "inert" from our revised manuscript.

Lines 36-39: Figure 1a could be cited.

Response: We have cited Fig. 1a in this sentence.

Line 38: as a second option, I would consider "modification during gelation" rather than post-gelation and proper references should be considered.

Response: We thank the reviewer for the suggestion. We have changed "post gelation modification of functional molecules onto pre-organized reactive sites along polymer chains within hydrogels" to "modification of functional molecules or the pre-organized reactive sites during gelation", with appropriate references.

Line 43: after "functionalization step" a reference is needed.

Response: We thank the reviewer for the suggestion and we have cited a relevant reference after "functionalization step" in our revision.

Line 45: I would change in " which limits the flexibility etc"

Response: We take the suggestion and changed "...which highly limited the flexibility..." to "...which limits the flexibility..." in our revision.

Line 51: should be "to realize surface modifications"

Response: We thank the reviewer to point out this. We have changed "...to realize surfaces modification..." to "...to realize surface modifications..." in our revision.

Line 59: should be "designed"

Response: We thank the reviewer for pointing out this. We have corrected the tense to "designed" in our revision.

Results

Line 103: which is the meaning of F-X curve?

Response: We thank the reviewer for this question. F-X curve means force-extension curve (*Nat. Commun.* **2020**, *11*, 3895.). More literature used force-distance (F-D) curve, which exhibits the characteristic

point of separation of the tip from the surface and single-molecule adhesion events (*Angew. Chem. Int. Ed.* **2020**, *59*, 16616; *Nanoscale* **2016**, *8*, 15309; *PNAS* **2006**, *103*, 12999). The F-D curve is more suitable for our experiments, so we changed “F-X curve” to “F-D curve” in our revised manuscript.

Lines 99-112: Here initially authors compare the adhesion force of a dipeptide YY, tripeptide YKY and their sequence DbayKY however they do not comment on the possible effect of sequence length on adhesion force. No other comments are present explaining the differences found. In lines 107-108, they state that K and Dbay are important for adhesion of their DbayKY sequence. Then they tested KYK and showed slightly lower adhesion to PEG than YKY (why?) and much lower than for DbayKY (why?). On the other hand KKYY has an adhesion strength which appears close to that of DbayKY (246 vs 251 pN) and here they state that the sequence in DbayKY is not important for the adhesion. After this paragraph, I am confused: initially the authors state that lysine and Dbay are important for adhesion (line 104), then it seems the sequence is not important. What is it important for the adhesion? Although in lines 148-154 there is a general explanation, here you should specify the reasons for the different adhesion strengths to PEG substrate. On the other hand, I found very informative your answer to Reviewer 1: please try to give here similar information to those provided in the answer to Reviewer 1. Importantly, it appears that you use the term “sequence” which can be misunderstood while I think you could substitute it with comments on the “order” or “distribution” of amino acids in the peptide sequence you test. Please also consider that the number of peptides also affect interactions.

Response: We thank the reviewer for these important suggestions. The order of rupture force in our study is $\text{DbayKY} \approx \text{KKYY} > \text{YKY} \approx \text{KYK} > \text{YY}$. The results imply that when the sequence length and the number of K and Y increase, the overall adhesion force will increase, which is also consistent with the observation that the modest increase in peptide length, from KY to (KY)₃, increases adhesion strength to TiO₂ (*Nat. Commun.* **2020**, *11*, 3895). When use median values for comparison, KYK showed slightly higher (but without significant statistic difference) adhesion to PEG than YKY, indicating both K and Y are important for adhesion. KYK showed lower adhesion than that of DbayKY, indicating the importance of two catechol units within DbayKY. The presence and the number of Dbay and lysine play an important role in adhesion, but when the component is similar, the strict order of amino group and catechol group is not important. To prevent ambiguity, we have changed “sequence” to “order” as suggested by the reviewer.

In our revised manuscript, based on the reviewer’s suggestions and our response to Reviewer 1 in our previous revision, we added relevant discussion into our revision this time, as shown below:

“We found that median values of rupture force of YY without any amine group (~82 pN, Fig. 3c) is significantly lower than that of DbayKY (223 pN), and that the rupture force of YKY (~164 pN, Fig. 3d) with the removal of Dbay is also lower than that of DbayKY (223 pN), which indicates that the presence and the number of Dbay and lysine play an important role in adhesion. The primary amine group within Dbay and lysine play a synergetic role with catechol to promote the adhesion strength to hydrogels. The observed importance of amine groups in our study is consistent to the conclusion in the literature that introduction of amine groups, such as lysine, to DOPA-containing peptides can enhance the adhesion strength. We prepared peptide KYK and found that its median rupture force to hydrogels (~172 pN, Fig. 3e) has no significant statistic difference from YKY (~164 pN), indicating both K and Y are important for adhesion. KYK showed lower adhesion than did DbayKY, indicating the importance of two catechol units within DbayKY.”

“All these show that two amine groups (from Dbay and lysine) and two catechol groups (from two DOPA units) are important for the peptide DbayKY to have strong adhesion to hydrogels, as inspired by the 1:1 DOPA:Lysine component in the adhesive protein Pc-1 from sandcastle worm. We also synthesized peptide KKYY and found its adhesion strength (~203 pN, Fig. 3f) is comparable to that of DbayKY (no significant difference), which indicates that strict order of units within DbayKY is not important for the adhesion to hydrogels, consistent with the observation in adhesive polymers. The order of rupture force in our study is $\text{DbayKY} \approx \text{KKYY} > \text{YKY} \approx \text{KYK} > \text{YY}$. The results imply that when the sequence length and the number of K

and Y increase, the overall adhesion force will increase, which is also consistent with the observation that modest increase in peptide length, from KY to (KY)₃, increases adhesion strength to TiO₂.”

Figure 3: please correct the letters a-g (c has been shifted and g is missing). In the legend please shift the comment on N values at the end of the legend.

Response: We thank the reviewer for pointing out the error. We have corrected the letters and shifted the comment on N values at the end of the legend in our revision.

Line 170: is the % modification expressed respect to polymer forming the hydrogel after hydrogel drying or respect to the weight of wet hydrogel? I would suggest to present both data, considering the differences in concentration (water content). However, in the experimental part you should clearly report the formula for calculation of the % of modification. Please detail in the exp. Part.

Response: We thank the reviewer for this question and suggestion. The % modification in our previous manuscript is expressed respect to polymer forming the hydrogel after hydrogel drying. According to the reviewer’s suggestion, we present both data of modification amount relative to dry weight hydrogel and wet weight hydrogel in our revised manuscript (Supplementary Figure 6 and Supplementary Figure 8, as shown below). In addition, we have provided the formula for calculation of the % of modification in the exp. Part (Methods part):

For fluorescamine method in analyzing soluble PVA and ALG hydrogels, the freeze-dried hydrogel was dissolved as 1 wt% solution for fluorescence detection, from which the concentration of functional molecule (C_f) was obtained using a calibration curve (Supplementary Figure 5). Hence, the actual **weight of molecules bound to hydrogel** = $(C_f \times \text{dry weight ratio of hydrogel}/1\%) \times \text{volume of hydrogel}$. The **efficiency of the functionalization (%)** = $((C_f \times \text{dry weight ratio of hydrogel}/1\%) \times \text{volume of hydrogel}/\text{weight of functionalizing molecules initially used}) \times 100$. For rhodamine method, hydrogels were directly used for fluorescence detection, the concentration of functional molecule (C_r) is the actual density of modified molecules to the hydrogel, and was obtained using a calibration curve (Supplementary Figure 7). Therefore, the **weight of molecules bound to hydrogel** = $C_r \times \text{volume of hydrogel}$. The **efficiency of the functionalization (%)** = $(C_r \times \text{volume of hydrogel}/\text{weight of functionalizing molecules initially used}) \times 100$.

Quantitative analysis of molecule modified to hydrogel were conducted, including **modification amount to dry weight hydrogel (%)** that is calculated by the formula: $(\text{weight of molecules bound to hydrogel}/\text{dry weight of initial hydrogel}) \times 100$, and **modification amount to wet weight hydrogel (%)** that is calculated by the formula: $(\text{weight of molecules bound to hydrogel}/\text{wet weight of initial hydrogel}) \times 100$.

Supplementary Figure 6b-c. Quantitative analysis of Pol-7 on PVA and ALG hydrogels, including modification amount to dry weight hydrogel (%) that is calculated by the formula: (weight of molecules bound to hydrogel/dry weight of initial hydrogel) \times 100 (**b**), modification amount to wet weight hydrogel (%) that is calculated by the formula: (weight of molecules bound to hydrogel/wet weight of initial hydrogel) \times 100 (**c**).

Supplementary Figure 8c-d. (c-d) Quantitative analysis of DbayKY-OEG8-Rh (0.5 mg/mL) on various hydrogels, including modification amount to dry weight hydrogel (%) that is calculated by the formula: (weight of molecules bound to hydrogel/dry weight of initial hydrogel) \times 100 (**c**), and modification amount to wet weight hydrogel (%) that is calculated by the formula: (weight of molecules bound to hydrogel/wet weight of initial hydrogel) \times 100 (**d**).

Line 189: I would suggest to express the functionalization also respect to dry weight of hydrogel (after drying hydrogels) = [amount of functionalizing molecules (e.g. weight) in final hydrogel/weight of initial polymer hydrogel (in dry state)] \times 100

Response: We thank the reviewer for this suggestion. We have added these data in the Supplementary Figure 6b and Supplementary Figure 8c of our revised manuscript. The calculation details have been described in details in our response to the reviewer's comment above (question above for Line 170).

Lines 192-196: stability tests should be performed at least for 7 days at different time points to assess stability of the functionalization.

Response: We thank the reviewer for this suggestion. We have added the stability tests of functionalized hydrogels in PBS and proteinase K for up to 7 days. The results showed that the functionalized hydrogels can both be stable in PBS and proteinase K for 1 day, 3 days and 7 days (Supplementary Figure 9 of our revised manuscript, as shown below).

Supplementary Figure 9. Stability test of DbayKY-OEG8-Rh modified PEG hydrogels under the treatment of PBS for 1 day, 3 days and 7 days or protease K (PK) at 37 °C for 2 h, 1 day, 3 days and 7 days. n = 6, mean values \pm s.d. Statistical analysis: two-tailed t-test. ns: not significant.

In the main manuscript, discussion would deserve improvement and rearrangement, by including comments on the following points (in some cases, they are already present but need better explanation and integration each other)

Response: We thank the reviewer for this suggestion and detailed comments below. In response to the reviewer's suggestions, we added a Discussion Part in our revised manuscript for better explanation and integration the comments of the following points.

1) Please motivate the choice of pH for functionalization and pH effect (together with chosen functionalization time) on the self-polymerisation of the peptide and on functionalization efficiency. You introduced such comments in the answer to reviewers letter but they are missing in the main manuscript.

Response: We thank the reviewer for pointing out this. We added relevant comments in the Discussion part of our revised manuscript as mentioned below:

1. Our study indicates that the amine and catechol groups within the adhesive peptide can form cross-linking at alkaline pH to obtain high density of functionalization, consistent to the report in the literature. Promoting cell adhesion and obtaining antibacterial functions of hydrogels often require sufficient density of functional molecules. Therefore, we chose to use the pH 8.5 condition to have high modification density. If a research requires a monolayer modification, pH 6.0 condition can be used for modification to minimize the catechol-derived crosslinking and multilayer modification.
2. Through the mechanism study of polymer adhesion to hydrogels, it can be inferred that the modification amount and modification density will gradually increase over time. A good functionalization method hopes that the modification time should not be too long. We used 24 h as the modification time, which is also commonly used in the field of catecholic chemistry for surface modification, and is generally accepted by many researchers. We found that functionalized

hydrogels with a high amount of modification can be realized at this time window, and the functions of antibacterial and tissue engineering can be realized by using this protocol.

2) Please stress on the aim of the approach, being the synthesis of DbA-DOPA-Lys-DOPA-terminated molecules for the preparation of functionalization polymers, which can then be exploited for wet hydrogel functionalization according to a versatile approach. In doing that, please underline the adhesion mechanisms by the binding peptide and that this could change the functionalization degree, with examples on samples you tested. Also please underline how it is possible to change the functionalization degree of hydrogel and within which range it can be done. Please refer the functionalization degree respect both to the wet hydrogel and the dried hydrogel in order to properly comment the results based on material chemistry, hydrogel polymer amount, etc. Please clearly state that the functionalization typically cannot penetrate the hydrogel network, but it is proposed as a surface functionalization of the exposed hydrogel surface. Also please underline that bulk functionalization of hydrogel structure is possible by using lyophilized hydrogels.

Response: We thank the reviewer for these suggestions. We added relevant comments to the Discussion part in our revision as shown below.

1. We stress the aim of our approach that “Hereby we report a simple one-step hydrogel functionalization strategy using a sandcastle worm-inspired cell adhesive DbAYKY tripeptide.”
2. About the adhesion mechanism, “We hypothesize that the functionalization mechanism of the DbAYKY terminated polymers to hydrogels is attributed to multiple interactions including hydrogen bonding, cation- π stacking and charge interactions. The catechol groups within DbAYKY can utilize two neighboring hydroxyl groups as the donors/acceptors to form hydrogen bonding to hydrogels. The positively charged amine groups can form cation- π interactions with the aromatic rings within DOPA. Therefore, the presence and the number K and Y are important for adhesion. The order of rupture force in our study is DbAYKY \approx KKYY $>$ YKY \approx KYK $>$ YY, which implies that when the sequence length and the number of K and Y increase, the overall adhesion force and the functionalization degree will increase. In addition, the DbAYKY terminated polymers have positively charged amine groups, therefore, can interact electrostatically with hydrogels bearing negatively charged groups, such as PSBMA and ALG. To have our discussion on mechanism more readable, we have moved the discussion on the adhesion mechanism from the Result part to the Discussion part in our revision.”
3. About the functionalization degree we added discussion that “Moreover, it is possible to change the functionalization degree of hydrogel, for example, the modification amount of DbAYKY-OEG8-Rh to dry weight PEG hydrogel from \sim 0.0058% to \sim 0.0262%, with the increase of DbAYKY-OEG8-Rh concentration initially used, from 0.0625 to 1 mg/mL. For different molecules modified to different hydrogels, the functionalization degree and the efficiency of the functionalization are generally different. In our study, we found that the modification amount of DbAYKY-OEG8-Rh to five different hydrogels were from \sim 0.020% to \sim 0.104% respect to dry hydrogels, and from \sim 0.001% to \sim 0.058% respect to wet hydrogels. For specific applications in the future, the functionalization can be optimized by changing the functionalization conditions such as the concentration of molecules and pH value during functionalization.”
4. We also have discussion on the hydrogel penetration and bulk functionalization that “This strategy is proposed as a surface functionalization of the exposed hydrogel surface. The functionalization typically cannot penetrate the hydrogel network, even though, bulk functionalization of hydrogel structure is possible by using lyophilized hydrogels. Hydrogels with large pore size for efficient polymer diffusion into hydrogels can be used directly for functionalization with the reported method

here; polymers with small pore size can be pretreated, such as freeze-drying, to obtain enlarged pore size that enables efficient polymer diffusion into hydrogels and functionalization. Therefore, bulk functionalization of hydrogel structure is also possible by using lyophilized hydrogels. This adhesive peptide strategy can be used to meet the requirement for bulk hydrogel functionalization with diverse applications such as regulating cell migration into hydrogels in tissue engineering and cell encapsulation.”

3) You should also comment on the functionalization stability as a function of incubation time (up to at least 7 days), please. I appreciate the addition of stability studies up to 24 h and to 2 h (with enzyme), however these times are very brief: what does it happen later? You can also simply refer to stability in PBS as a function of time (at least 7 days), please.

Response: We thank the reviewer for this suggestion. We have added the stability tests of functionalized hydrogels in PBS and proteinase K for up to 7 days. The results showed that the functionalized hydrogels can both be stable in PBS and proteinase K for 1 day, 3 days and 7 days (Supplementary Figure 9 in our revised manuscript, as shown above). We have also commented on the functionalization stability in the Discussion part that “The functionalized hydrogels can both be stable in PBS and proteinase K for up to 7 days...”

4) Additionally, you could comment on the efficiency of the functionalization (I have asked it in my previous comments but maybe I was not clear). In your work, you use a certain functionalizing molecule amount: a part binds to the hydrogel surface. Is it possible to measure (the ratio between the functionalizing molecule which has been linked to the surface to the overall functionalizing molecule initially used) $\times 100$ (efficiency of functionalisation) ? In case please, add it also in the experimental part, mentioning the formula.

Response: We thank the reviewer for this question. As discussed in our response to the reviewer’s related question above, it is possible to measure the efficiency of functionalization. The amount of functionalizing molecules that has been linked to the hydrogel can be calculated by fluorescamine method or rhodamine method, according to relevant calibration curve and the initial amount of functionalizing molecules.

For fluorescamine method in analyzing soluble PVA and ALG hydrogels, the freeze-dried hydrogel was dissolved as 1 wt% solution for fluorescence detection, from which the concentration of functional molecule (C_f) was obtained using a calibration curve (Supplementary Figure 5). Hence, the actual **weight of molecules bound to hydrogel** = ($C_f \times$ dry weight ratio of hydrogel / 1%) \times volume of hydrogel. The **efficiency of the functionalization (%)** = (($C_f \times$ dry weight ratio of hydrogel / 1%) \times volume of hydrogel / weight of functionalizing molecules initially used) $\times 100$. For rhodamine method, hydrogels were directly used for fluorescence detection, the concentration of functional molecule (C_r) is the actual density of modified molecules to the hydrogel, and was obtained using a calibration curve (Supplementary Figure 7). Therefore, the **weight of molecules bound to hydrogel** = $C_r \times$ volume of hydrogel. The **efficiency of the functionalization (%)** = ($C_r \times$ volume of hydrogel / weight of functionalizing molecules initially used) $\times 100$.

As we mentioned in our response to the reviewer’s similar comment above, we added these formulas in the Methods part of our revised manuscript. We also added relevant comments in the Discussion part that “During the modification process, we can quantify the efficiency of the functionalization that is the ratio of the functionalizing molecules, which has been linked to the surface, to the overall functionalizing molecules

initially used.”

5) Please comment on prospective exploitation of your functionalizing strategy and add recommendations/suggestions for its optimisation on different hydrogel materials. In the answer to reviewers you introduced the synthesis of a simple binding peptides which could be easily linked to commercial molecules: please comment also on the use of this binding peptide

Response: We thank the reviewer for these suggestions. We added relevant comments in the Discussion part of our revised manuscript:

1. About the prospective exploitation of our strategy, we comment that “...acquire biological functions such as maintaining antibacterial and cell adhesion functions for at least 7 days, and promoting wound repair. This hydrogel modification strategy provides a convenient tool for the easy and direct modification of wet hydrogels with diverse applications.” and “This adhesive peptide strategy can be used to meet the requirement for bulk hydrogel functionalization with diverse applications such as regulating cell migration into hydrogels in tissue engineering and cell encapsulation.”
2. We also comment on the strategy optimization that “For different molecules modified to different hydrogels, the functionalization degree and the efficiency of the functionalization are generally different...” and “For specific applications in the future, the functionalization can be optimized by changing the functionalization conditions such as the concentration of molecules and pH value during functionalization.”
3. We comment on the use of this binding peptide that “The functionalization method is simple and could be widely used if with the availability of specific functionalization molecules, as our demonstration in both solution-phase and solid-phase synthesis. The amine-terminated DbaYKY tripeptide (DbaYKY-NH₂) could be easily linked to commercial molecules such as antibacterial agents and cell adhesive RGD peptide.”

6) The antimicrobial and cell adhesive functionality it evaluated only at one timepoint (24 h). As it is not clear whether the functionalization is stable and preserves its biological function as a function of time, I would recommend to add 2 additional timepoints up to 7 days for antimicrobial and cell adhesion tests (in parallel to the additional timepoints in stability studies)

Response: We thank the reviewer for these suggestions. We have added antimicrobial and cell adhesion tests (for 1 day, 3 days and 7 days) in parallel to the additional timepoints in stability studies in our revised manuscript, as in our response to the reviewer's first three comments above. We also added relevant discussion in the Discussion part that “...acquire biological functions such as maintaining antibacterial and cell adhesion functions for at least 7 days, and promoting wound repair.”

Methods

Lines 287-293: please be sure the codes have been previously defined

Response: We thank the reviewer for reminding us on this. We checked those codes in this part and make sure that the full name of the code was provided for the first time showing in the manuscript.

Line 317: Please introduce what you mean here as polymer. You are not introducing your Pol materials before explaining functionalization of hydrogels with them. Please refer to Supplementary as to introduce them.

Response: We thank the reviewer for this kind suggestion. We have changed “All polymers were...” to “Polymers with an adhesive dibutylamine-DOPA-lysine-DOPA tripeptide (DbaYKY), described in Fig. 2 and Supplementary Scheme 6-13, were...”

Line 324: indicate the type of freeze drier

Response: We thank the reviewer for pointing out this. The freeze drier we used is the Labconco® FreeZone Plus 4.5 liter cascade benchtop freeze dry system. We added this information in the Method part as well as the Materials part in our revised Supplementary Information.

Line 327: Kaiser test should be explained here or in supplementary material. Proper reference to Supplementary material should be present.

Response: We thank the reviewer for reminding us on this. We have added a detailed description of Kaiser test in the Method part that “The Kaiser test was conducted by followed the general method widely used in solid phase synthesis. Kaiser reagent is a mixed solution of 2 μ L EtOH containing 50 mg/mL ninhydrin, 2 μ L phenol and 2 μ L pyridine. The solution of amino group-containing compounds will turn to dark blue after heating. 2 μ L last washing water of the hydrogel in duplicates was mixed with 6 μ L of the Kaiser reagent and heated to 90~100 °C for 1 min to show the color change.”

Line 328: cell adhesion studies are not explained here. I suggest to describe them later in the specific par. on in vitro cell tests, saying there that you also tested these types of samples with cells.

Response: We take this suggestion and move the explanation of cell adhesion studies to the Cell Adhesion assay part that “The Pol-7 modified cell adhesive PHEMA hydrogel was obtained via lyophilization treatment, which was then cut with a knife from the middle to expose the inside cross section. The same method was used for the cell adhesion assay on the surface and inside interface of Pol-7 modified PHEMA hydrogel that was prepared via lyophilization treatment.”

Line 332: should be “shown” not showed

Response: We thank the reviewer for pointing out this. We have corrected the tense.

AFM force spectroscopy: indicate how many samples you tested for each Pol for statistical purposes and which data you reported in the paper.

Response: We thank the reviewer for pointing out this. Due to the broader range of interfacial adhesive mechanisms of the adhesion moiety, the cantilever should be continuously approaching to hydrogel to obtain the rupture force distribution in the AFM-based force spectroscopy measurements (*Nat. Commun.* **2020**, *11*, 3895; *Angew. Chem. Int. Ed.* **2019**, *58*, 1077). The data of the undetected force were discarded, whin

agreement to the suggestion from the first reviewer. The remaining valid tip–surface binding events (*N* value) ranged from 320 to 495 were used for statistical analysis.

In order to verify the repeatability of the experiment, we prepared two samples for each adhesion moiety in the AFM-based force spectroscopy measurements. In an experiment, we prepared 5 modified tips with DbaYKY, YY, YKY, KYK and KKYY respectively. This experiment was repeated twice at different times to obtain results that are comparable and have the same conclusion. The results of two repeats were shown below:

Supplementary Table 3. SMFS results of two batches of samples.

		Rupture force (pN)				
		Db a YKY	YY	YKY	KYK	KKYY
Median	Batch 1	223.0	81.7	163.5	172.2	202.8
	Batch 2	240.6	58.2	151.4	167.3	227.8
Mean	Batch 1	251.9	97.2	210.0	203.3	246.0
	Batch 2	260.2	70.8	171.9	192.1	272.8

XPS analysis: was the sample dried before the analysis? If yes, please indicate it

Response: We thank the reviewer for this question. Hydrogels were freeze-dried before XPS analysis. We have changed the description on this part in our revision to “After modification, hydrogels were freeze-dried. The external surface of each freeze-dried hydrogel was analyzed and all peaks were calibrated...”

FTIR analysis: was the sample dried before the analysis? If yes, please indicate it

Response: We thank the reviewer for this question. Hydrogels were freeze-dried before FTIR analysis. We have changed the description on this part in our revision to “After modification, hydrogels were freeze-dried. The external surface of each freeze-dried hydrogel was analyzed.”

Line 360: Kaiser test should be better explained. It has already been introduced in the text so it should be explained at the first introduction in the exp.

Response: We thank the reviewer for this suggestion. We have explained the Kaiser test at the first introduction in the experimental part, the “Hydrogel modification” part, as suggested by the reviewer in previous comment. The Kaiser test described here is to detect the color change of the hydrogel to confirm that the hydrogel is successfully modified with pol-7. The previous Kaiser test description in the “Hydrogel modification” part is to detect the color change of the hydrogel washing solution, which confirms whether unmodified pol-7 was thoroughly removed from the hydrogel. The only difference between them is that the tested samples are different. In response to the reviewer’s comment, we provided a description on Kaiser test with more details in the “Modified hydrogels characterization” part of our revision that “The hydrogels in duplicates were either incubated with 50 μ L Kaiser reagent (Fig. 4f) or immersed into 200 μ L Kaiser reagent (Supplementary Figure 13), and then was incubated at 90~100 °C for 1 min to show the color change.”

Line 366: “were modified to hydrogel” is wrong expression

Response: We thank the reviewer for pointing out this. We have changed “...were modified to hydrogel...” to “...was used to functionalize hydrogels...”

Lines 379-381: it is not clear how you referred data to hydrogel material. Please indicate the equation you used to give the modification % and state clearly whether you made reference to dried hydrogel. If not, please also add calculation respect to dried hydrogel

Response: We thank the reviewer for this question. In previous revision, the modification % was respected to dried hydrogel. In our revised manuscript this time, we quantified both modification amount to dry and wet hydrogels, and described the quantitative process and formula in detail, as suggested by the reviewer in the comment above. Modification amount to dry weight hydrogel (%) is calculated by the formula: (weight of molecules bound to hydrogel / dry weight of initial hydrogel) \times 100, and modification amount to wet weight hydrogel (%) is calculated by the formula: (weight of molecules bound to hydrogel / wet weight of initial hydrogel) \times 100. A detailed description is added to the Quantitative analysis part in our revision, as is described in or response to the reviewer's previous comment on this question.

Lines 385-386 and 389: What is the PB buffer?

Response: We used phosphate buffer, but the expression of "PB buffer" is not suitable. So, we changed "PB buffer" to "Phosphate buffer (PB)" in our revised manuscript.

Line 395: indicate the equation used to define the modification amount. Is it expressed respect to dried hydrogel?

Response: We thank the reviewer for this suggestion. We quantified both modification amount to dry and wet hydrogels in this revision. We have added the equation in the Method part of our revised manuscript.

In "*Fluorescamine method*" part we added: "The concentration of functional molecule (C_f) was obtained using a calibration curve (Supplementary Figure 5). Hence, the actual **weight of molecules bound to hydrogel** = ($C_f \times$ dry weight ratio of hydrogel / 1%) \times volume of hydrogel. The **efficiency of the functionalization (%)** = (($C_f \times$ dry weight ratio of hydrogel / 1%) \times volume of hydrogel / weight of functionalizing molecules initially used) \times 100. Quantitative analysis of Pol-7 on PVA and ALG hydrogels were conducted, including **modification amount to dry weight hydrogel (%)** that is calculated by the formula: (weight of molecules bound to hydrogel / dry weight of initial hydrogel) \times 100, and **modification amount to wet weight hydrogel (%)** that is calculated by the formula: (weight of molecules bound to hydrogel / wet weight of initial hydrogel) \times 100.

In "*Rhodamine method*" part we added: "The concentration of functional molecule (C_r) is the actual density of modified molecules to the hydrogel, and was obtained using a calibration curve (Supplementary Figure 7). Therefore, the **weight of molecules bound to hydrogel** = $C_r \times$ volume of hydrogel. The **efficiency of the functionalization (%)** = ($C_r \times$ volume of hydrogel / weight of functionalizing molecules initially used) \times 100. Quantitative analysis of DbYKY-OEG8-Rh on various hydrogels were conducted, including **modification amount to dry weight hydrogel (%)** that is calculated by the formula: (weight of molecules bound to hydrogel / dry weight of initial hydrogel) \times 100, and **modification amount to wet weight hydrogel (%)** that is calculated by the formula: (weight of molecules bound to hydrogel / wet weight of initial hydrogel) \times 100."

Line 410-415: this is not an experimental part, this is a result and should be integrated into the main manuscript.

Response: We thank the reviewer for reminding us on this. We have moved this part to the main

manuscript.

Line 419: should be “were”

Response: We thank the reviewer for pointing out this. We have corrected “*E. coli* 25922 and *S. aureus* 6538 was chosen...” to “*E. coli* 25922 and *S. aureus* 6538 were chosen...” in our revised manuscript.

Line 433: parenthesis is not needed

Response: We thank the reviewer for pointing out this. We have deleted the parenthesis in our revised manuscript. The sentence was changed to “...calculated using the equation that killing efficacy (%) = $\frac{C_{control} - C_{polymer}}{C_{control}} \times 100$.”

Line 438: should be “were”

Response: We thank the reviewer for pointing out this. We have corrected “Cells at 80-90% confluency was trypsinized...” to “Cells at 80-90% confluency were trypsinized...” in our revised manuscript.

Cell adhesion: please notice that you also introduced cell adhesion tests before. Hence you could describe the cell analysis on cut hydrogels here, too.

Response: We thank the reviewer for this kind suggestion. We have added relevant description on the description of cell analysis to our revision that “The Pol-7 modified cell adhesive PHEMA hydrogel was obtained via lyophilization treatment, which was then cut with a knife from the middle to expose the inside cross section. The same method was used for the cell adhesion assay on the surface and inside interface of Pol-7 modified PHEMA hydrogel that was prepared via lyophilization treatment.” in our revised manuscript.

In vivo trials: indicate the microscope used for the analysis

Response: We thank the reviewer for pointing out this. All staining images were scanned in Panoramic 250/MIDI scanner equipped with the CaseViewer 2.0 software. We added this information into the last sentence of the “In vivo wound healing in a full-thickness skin defect model” part in our revised manuscript.

Please control that all the reagent characteristics (supplier and other key properties) are present in the Supplementary materials or in the Experimental part of the main paper. Importantly, guide the reader in the Exp part for reference to the Supplementary material for additional details. This is missing.

Response: We thank the reviewer for pointing out this. All the reagent characteristics (supplier and other key properties) are present in the “Materials” part in the Supplementary Information. To guide readers, we have added “All the reagent information is present in the Supplementary Information.” in the “Hydrogels preparation” part in our revised manuscript.

Supplementary materials

Some minor mistakes:

Lines 74-76, 88-89, 124-125, 147-149, 199-200, 211-213 248-250, 260-262, 271-272, 417-419, 476-478, 487-488, 506-508: please rephrase as that the subject is reported before the verb (and not the opposite as in the current form)

Response: We thank the reviewer for this suggestion. We have rephrased all sentences pointed out by the reviewer to have subject reported before the verb.

Supplementary Figure 12 Legend: Please indicate the culture time (24 h)

Response: We thank the reviewer for pointing out this. We have added the culture time (24 h) in the legend of Supplementary Figure 14 of our revised manuscript (Supplementary Figure 12 in our original manuscript).

Supplementary Figure 5 and 6, (a): please remove “diagram” as it is a “schematic representation”

Response: We thank the reviewer for reminding us on this. We have changed “schematic diagram” to “schematic representation” in the Supplementary Figure 6 and 8 of our revised Supplementary Information.

Line 462: please correct into “are conducted”

Response: We thank the reviewer for pointing out this. We have corrected this typo.

Line 444: please correct “after coupling complete”, e.g. into “after completion of the coupling reaction”

Response: We thank the reviewer for pointing out this. We have changed “After couplings complete ...” to “After completion of the couplings reaction...” in our revised Supplementary Information.

Finally I have noticed that the reference to Supplementary material is “random”, i.e. it does not follow the order in which Supplementary materials are presented. This should be checked and adjusted.

Response: We thank the reviewer for pointing out this. We follow the request and adjust the reference to Supplementary material in our revision.

REVIEWERS' COMMENTS

Reviewer #1 (Remarks to the Author):

The authors have made constructive steps to respond to my comments on analysis of the AFM data and this is now substantially improved.
There are so many figures in the SI and the AFM stats so closely inform the interpretation of Fig. 3 that I would be inclined to integrate this data in with Fig. 3.
However, I will leave that to the authors' discretion.
I have no further comments.

Reviewer #3 (Remarks to the Author):

authors have addressed all the many points arisen by myself, also integrating the manuscript with new experiments and discussion. From my side I am satisfied by the revised version.

Point-by-point response to the reviewers' comments

Reviewer #1 (Remarks to the Author):

The authors have made constructive steps to respond to my comments on analysis of the AFM data and this is now substantially improved. There are so many figures in the SI and the AFM stats so closely inform the interpretation of Fig. 3 that I would be inclined to integrate this data in with Fig. 3.

However, I will leave that to the authors' discretion.

I have no further comments.

Response: We thank the reviewer for this suggestive comment to help us improve our manuscript. We have followed this suggestion and added the important median value into Fig. 3 in our revised manuscript, as shown below.

Fig. 3 SMFS results for the interaction of DbayKYK, YY, YKY, KYK and KKYY with PEG hydrogels.

Reviewer #3 (Remarks to the Author):

authors have addressed all the many points arisen by myself, also integrating the manuscript with new experiments and discussion. From my side I am satisfied by the revised version.

Response: We thank the reviewer for the positive response and all valuable suggestions in previous comments to help us improve our manuscript.

We greatly appreciate all the reviewers' valuable comments. We hope that the revised manuscript will prove to be acceptable for publication in *Nature Communications*.

Sincerely,

Runhui Liu

Professor of Chemistry and Biomaterials